# Human tactile sensing and sensorimotor mechanism: from afferent tactile signals to efferent motor control

Yuyang Wei [1,2], Andrew G. Marshall [3], Francis P. McGlone[4], Adarsh Makdani [5], Yiming Zhu [2], Lingyun Yan[2], Lei Ren [2,6] ✉ & Guowu Wei [7] ✉

In tactile sensing, decoding the journey from afferent tactile signals to efferent motor commands is a significant challenge primarily due to the difficulty in capturing population-level afferent nerve signals during active touch. This study integrates a finite element hand model with a neural dynamic model by using microneurography data to predict neural responses based on contact biomechanics and membrane transduction dynamics. This research focuses specifically on tactile sensation and its direct translation into motor actions. Evaluations of muscle synergy during in-vivo experiments revealed transduction functions linking tactile signals and muscle activation. These functions suggest similar sensorimotor strategies for grasping influenced by object size and weight. The decoded transduction mechanism was validated by restoring human-like sensorimotor performance on a tendon-driven biomimetic hand. This research advances our understanding of translating tactile sensation into motor actions, offering valuable insights into prosthetic design, robotics, and the development of next-generation prosthetics with neuromorphic tactile feedback.

Cutaneous neural dynamics are integrated by the cuneate nucleus located at the brainstem to elicit primary perceptual and sensorimotor reactions before further processing in the thalamus and somatosensory cortex[1]. Understanding human tactile sensing and sensorimotor mechanisms is a long-term scientific challenge. To achieve this, population-level afferent tactile signals evoked during active touch should be studied under the resulting perception and efferent actions[2]. Additionally, understanding the transduction mechanism between afferent tactile signals and efferent motor control is crucial for a deeper insight into the human sensorimotor system, and has potential applications in prosthetics to enhance neural compatibility[3–8]. Previous studies have been conducted to observe the morphology and connectivity of cuneate neurons[9]. These neurons process and encode tactile information from hand mechanoreceptors, subsequently transmitting these signals to the somatosensory cortex for interpretation. Furthermore, techniques such as in-vivo microneurography and microsimulation have been developed to record afferent tactile signals and study the perceptual consequences of activating single sensory neurons, thus advancing our understanding of tactile coding mechanisms[10–14]. Post-synaptic neural signals[15–20] have been recorded to determine the flow and interaction of these sensorimotor signals within the peripheral and central nervous system. However, microneurography and macrostimulation are unsuitable for conducting repetitive[21] and invasive experiments on living subjects[12–14]. Moreover,

[1]Department of Engineering Science, University of Oxford, Oxford OX1 3PJ, UK. [2]Department of Mechanical, Aerospace and Civil Engineering, The University of Manchester, Manchester M13 9PL, UK. [3]Institute of Life Course and Medical Sciences, University of Liverpool, Liverpool L69 3BX, UK. [4]Department of Neuroscience and Biomedical Engineering, Aalto University, Otakaari 24, Helsinki, Finland. [5]School of Natural Sciences and Psychology, Liverpool John Moores University, Liverpool L3 5UX, UK. [6]Key Laboratory of Bionic Engineering, Ministry of Education, Jilin University, Jilin, China. [7]School of Science, Engineering and Environment, University of Salford, Manchester M5 4WT, UK. ✉e-mail: lren@jlu.edu.cn; g.wei@salford.ac.uk

the understanding of human tactile sensing and sensorimotor control mechanisms remain preliminary owing to the technically demanding measurements of the population-level afferent tactile signals during active touch. An effective method is required to gain access to afferent tactile signals during active exploration[22–26] and relate them with the resulting perception or efferent response. An integrated numerical method, based on the contact mechanics and neural dynamic model, was used to predict the group response of the cutaneous receptors; moreover, post-synaptic afferent signals were used to overcome the difficulties in this study.

In the meantime, researchers are attempting to restore these biological afferent tactile neural dynamics and the sensorimotor function on robotics toward the application of neuroprosthetic. Tactile sensors have been integrated with spiking neural models to convert analogue signals into digital spike trains mimicking the neural spiking features of human cutaneous receptors[2,24,25,27–29]. The discrimination accuracy resulting from these spike-trained signals has been quantified through the artificial neural network. However, whether these neuromorphic afferent tactile signals and machine learning-based decoding algorithms could represent the biological sensing mechanism is unclear. The spike-trained tactile sensors were also mounted onto the prosthetic or robotic hand to provide haptic feedback for restoring human hand performance[5,7,30]. Pioneering research has been conducted to control the moving speed of the robotic arm[31,32] through the brain-computer interface based on the efferent motor signals of the human subjects. Simple pain-reflex or slipping reactive controls alongside object recognition were also implemented[6,8,31,33–35]. Specific work on tactile feedback system has also been developed, including practical sensory feedback applications in bidirectional hand prostheses by Raspopovic et al.[36], exploration of long-term bionic hand adaptation by Ortiz-Catalan et al.[37], and investigation of multisensory perception by Preatoni et al.[38]. Unlike these studies, the research focuses on model-based sensorimotor control strategies and neural coding mechanisms. These strategies are specifically applied to develop closed-loop systems for enhancing neuroprosthetic functionality.

Toward these two main goals in this study, a method of using a finite element (FE) hand-based multi-level numerical model to effectively compute the afferent tactile signals under active touch was applied. Human afferent tactile signals were captured through microneurography to validate and optimize the numerical model. The dynamic relationship between afferent tactile signals and motor neuron signals was encapsulated in 'transduction functions', that offer the potential to restore sensorimotor performance in robotics or prosthetics. These 'concise transduction functions' represent a significant methodological advancement in this research, distilling complex sensorimotor interactions into simplified mathematical models. They capture the essential dynamics between afferent tactile signals and efferent motor responses, facilitating a deeper understanding of the underlying mechanisms crucial for neuroprosthetic development. The accuracy and applicability of the decoded sensorimotor transduction functions were demonstrated through a self-developed artificial tactile sensory system (ATSS) featuring neuromorphic tactile feedback, as shown in Fig. 1. The ATSS developed in this study exhibited sensing performance comparable to human subjects at the 2nd order cuneate neural signals stage, with differences in discrimination accuracy between the human subjects and the ATSS below 15%.

## Results

### Development of the neural dynamic model

A two-layered neural dynamic model (see Fig. 2) was adopted in this study to compute the biological and neuromorphic afferent tactile signals: the 1st order tactile neuron model predicting the cutaneous spike trains and the 2nd order cuneate neuron model for postprocessing the cutaneous neural dynamics[39].

The Izhikevich neural dynamic model was used as the core component to model the 1st order tactile neuron owing to its computing efficiency and ability to reproduce the spiking, bursting response, and adaption properties of mechanoreceptors[40,41]. A general porting function was applied to tune the current signals of the tactile sensor and fed into the Izhikevich neural dynamic model to compute neuromorphic afferent tactile signals[42]. The porting function and Izhikevich neural dynamic model were integrated to imitate the spiking features of the slowly adapting type I (SAI) and fast adapting type I (FAI) mechanoreceptor as presented in Eq. (1) to (4).

$$SAI: \frac{dv(t)}{dt} = 0.04v(t)^2 + 5v(t) + 140 - u(t) + \frac{K1}{C_m}I(t) \qquad (1)$$

$$FAI: \frac{dv(t)}{dt} = 0.04v(t)^2 + 5v(t) + 140 - u(t) + \frac{K2}{C_m}\frac{dI(t)}{dt} \qquad (2)$$

$$\frac{du(t)}{dt} = a(bv(t) - u(t)) \qquad (3)$$

$$If \ v \geq 30mv \begin{cases} v \leftarrow c \\ u \leftarrow u + d \end{cases} and \ I(t) = K \times I_{Sensor}(t) \qquad (4)$$

Where a, b, c, d are neuron parameters, u is the membrane recovery variable. K is the gain factor modulating the current signal of the sensor. The five parameters including the gain factor K and the neuron parameters a, b, c, d of the Izhikevich model were optimized using the response surface method (RSM) against the human afferent tactile signal captured through microneurography. The experimental process of microneurography and the raw afferent tactile signals captured are presented in Fig. S1 in the supplementary material, the measurement was performed by inserting a tungsten electrode into the median nerve at the wrist. The RSM method aims to find the specific magnitudes of parameters that produce the best goodness of fit (See Eq. 5) between the microneurography results and the neuromorphic signals. The initial values are a = 0.02 Ohm, b = 0.2, c = −65 mv, d = 6 mv obtained from the literature[24], the initial value of gain K was set as 25.

$$FSS = 1 - \frac{\sum_{i=1}^{n}[(exp-f)_i - (pre-f)_i]^2}{\sum_{i=1}^{n}(exp-f)_i^2} \qquad (5)$$

Where the exp stands for the measured spiking rate, pre is the predicted spiking rate computed through the optimised 1st order neuron model, n is the number of the data points. The optimized results were a = 0.02 Ohm, b = 0.205, c = −65 mv, d = 6.20 mv, K = 55 for SAI tactile unit and a = 0.02 Ohm, b = 0.210, c = −65.5 mv, d = 6.15 mv, K = 56 for FAI tactile unit.

The prediction process of the biological population-level 1st order afferent tactile signals is similar to that of computing the neuromorphic ones. The only difference is that the mechanoelectrical transduction model was used for deriving the membrane current rather than directly modulating the current output from the sensor through the gain function. A subject-specific FE human hand model was applied to simulate active touch. The strain energy density (SED) was extracted at the site of the mechanoreceptors and fed into the mechanoelectrical transduction model to derive the membrane current. The tactile signals elicited from the first-order neurons were then computed through the Izhikevich neural dynamic model based on the membrane current flow over the cutaneous receptor. The detailed process of predicting the human afferent tactile signals was illustrated in our previous research[43].

The 2nd order tactile neuron model was developed based on 1st order neuron model (see Fig. 2). The afferent tactile signals were

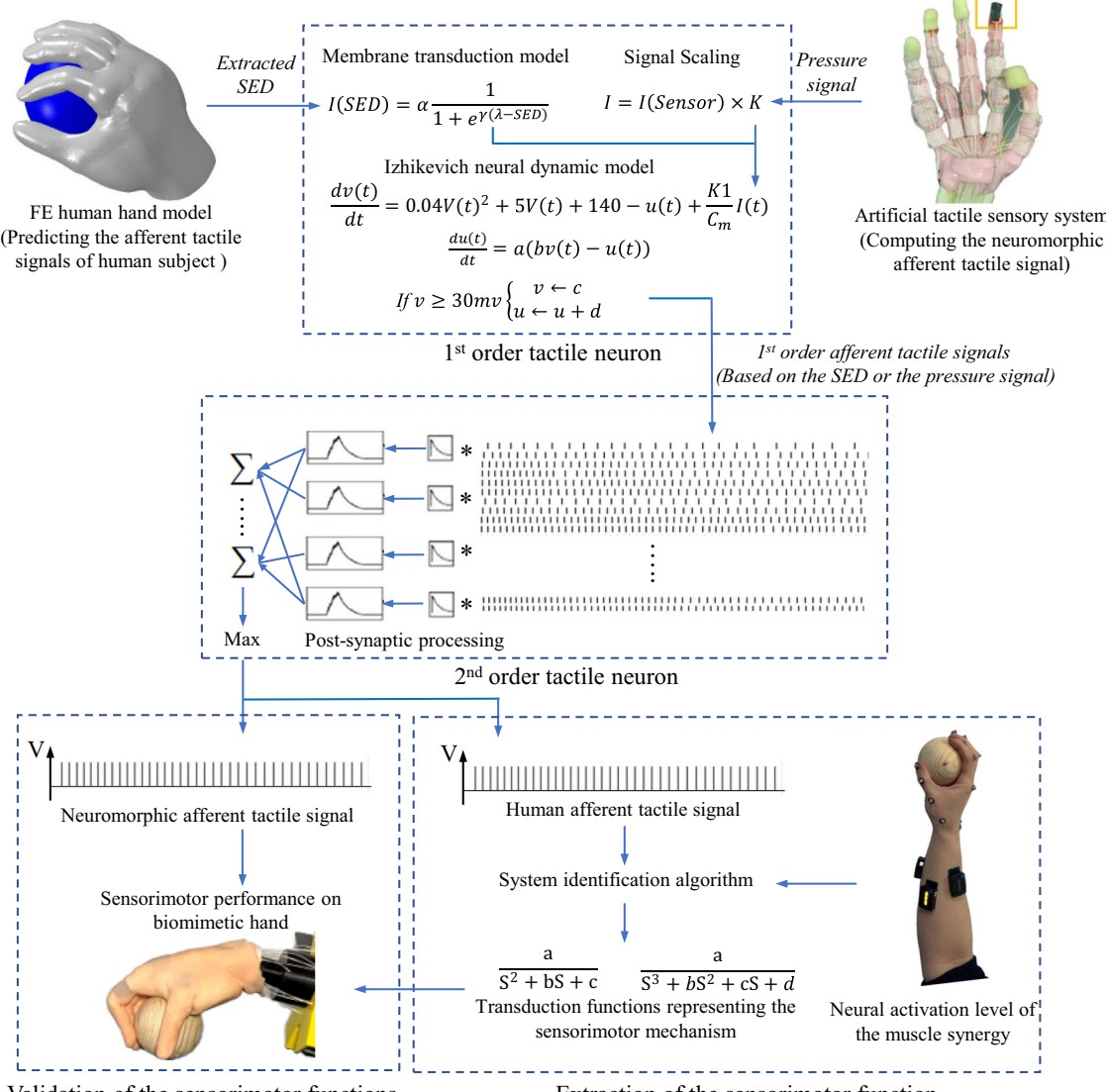

**Fig. 1 | The two-layered neural dynamic model for computing the biological and neuromorphic afferent tactile signals.** A validated FE human hand model was employed to simulate the active touch, the strain energy density (SED) was extracted at the site of mechanoreceptors and fed into the mechanoelectrical transduction model for deriving the membrane current. The biological 1st order afferent tactile signal was then computed through the Izhikevich neural dynamic model. The post-synaptic neural action potentials were integrated based on the cutaneous neural dynamics. Detailed process for predicting biological afferent tactile signals were presented in our previous work. The neuromorphic afferent signals were calculated based on a similar procedure. The gain function was adopted to modulate the current output of the tactile sensor array rather than the mechanoelectrical transduction model for predicting the biological membrane current.

convolved with the postsynaptic potential (PSP) waveform before being integrated. Equation 5, which describes the PSP waveform, is relevant to the neural model and can be explained in the context of the study. It represents the dynamics of the PSP that is crucial in the information processing of afferent tactile signals in the neural model, where the mathematical models and algorithms used in the research are described. The PSP waveform described by Eq. 5 plays a role in the neural processing of afferent tactile signals and contributes to the generation of neural dynamics in the cuneate neurons.

$$PSP_i = \exp\left(\frac{t}{\tau_{decay}}\right) - \exp\left(\frac{t}{\tau_{rise}}\right) \quad (6)$$

$$PSP_{total} = \sum_{i \in 36} PSP_i \quad (7)$$

The time decay $\tau_{decay} = 4$ ms and rise time $\tau_{rise} = 12.5$ ms[7] determine the shape of the $PSP_i$ kernel[44]. The mapping from the afferent fibers to the cuneate neurons was defined based on neuroanatomical data.

The average divergence/convergence ratio of 1700/300 corresponding to the fast feed-forward encoding/decoding process of the 2nd neuron level[45] was adopted. Therefore, if 100 SAI units are activated, there would be at least 567 cuneate neurons recruited to post-process these afferent tactile signals. The convolved 1st order afferent tactile signals were selected randomly and integrated as post-synaptic potentials according to the divergence/convergence ratio[39]. The maximum post-synaptic potential $PSP_{total}$ was picked and fed into the Izhikevich model to compute the neural dynamics of the cuneate neurons according to the winner-take-all algorithm[44].

This cuneate model incorporates two primary types of neurons to emulate the neural dynamics within the cuneate nucleus. These

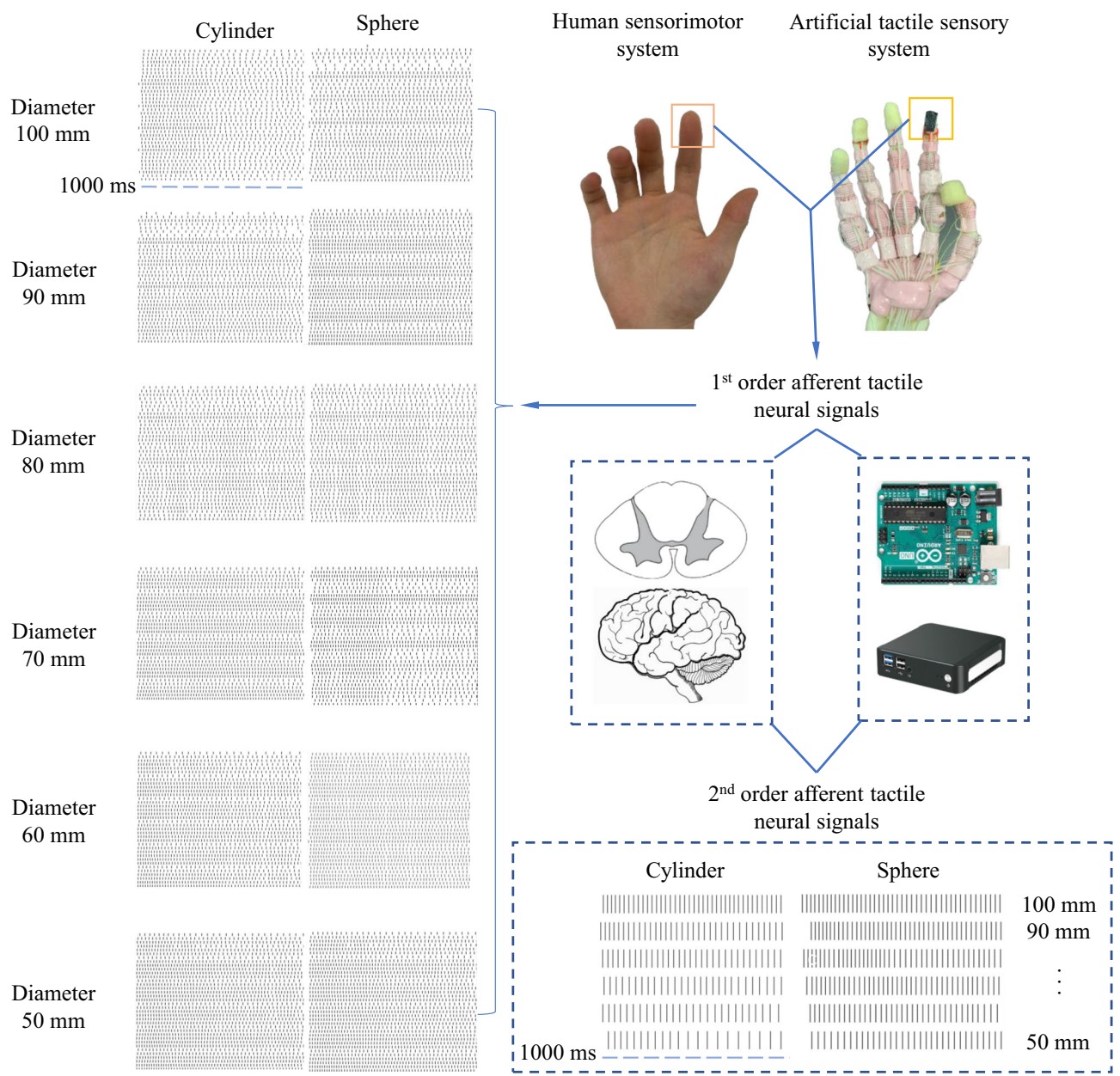

**Fig. 2 | The neuromorphic tactile signals elicited based on the 6 by 6 tactile sensor array.** The neuromorphic cutaneous (1st order tactile signals) and cuneate (2nd order tactile signals) signals (SAI units) elicited during active touch with cylinders and spheres ranging in diameter from 50 to 100 mm are presented. These tactile signals from two distinct levels are displayed alongside their corresponding positions within the human sensorimotor system and the artificial tactile sensory system.

neuron types include excitatory neurons and inhibitory interneurons. Excitatory neurons serve as the principal conveyors of sensory information, transmitting tactile signals from the peripheral mechanoreceptors to subsequent neural stages. In contrast, inhibitory interneurons play a pivotal role in modulating and fine-tuning neural activity by exerting inhibitory control over excitatory neurons. This dual-neuron representation aligns with the known neurobiology of the cuneate nucleus, contributing to a more biologically faithful simulation. The architecture of our cuneate model closely mirrors the anatomical organization of the cuneate nucleus observed in the human brainstem. The cuneate nucleus exhibits somatotopic organization, wherein different regions within the nucleus correspond to specific anatomical regions of the upper limb, particularly the hand and fingers. In our model, we faithfully replicate this somatotopic arrangement, allowing for the spatial mapping of tactile information. This organization ensures that tactile signals from distinct regions of the upper limb are processed separately within the cuneate nucleus before onward transmission.

## Extract the sensorimotor control strategy

After acquiring the population-level afferent tactile signals as the input for the sensorimotor control algorithm, the output neural activation level of the muscle synergy was extracted based on in-vivo grasping experiments. The detailed experimental setup and process are explained in the 'Methods' section. The neural activation level of the muscle synergy during active touch was extracted from the EMG signals using the non-negative matrix factorization algorithm (NMF)[46,47]. The optimal number of muscle synergy was regarded as its minimum value that achieved a mean variance account for (VAF) value above 85% with less than a 6% increment after adding another synergy[48]. Only a

single muscle synergy dominating the motor control of the forearm muscles was recognized during active and reactive grasping in this study. The muscle synergy neural activation level was regarded as the output of the sensorimotor controlling strategy, whereas the input was the 2nd order afferent tactile signals[49,50]. The use of synergy in our study is based on the idea that the human motor system often operates by combining the motions of individual fingers into coordinated patterns or synergies when performing various grasping tasks. These synergies represent coordinated patterns of muscle activations that simplify the control of multi-fingered hands, allowing for efficient and robust manipulation of objects. Our goal was to capture the essence of this human motor control strategy. The use of synergy in our modeling approach allows us to create a simplified yet effective representation of the sensorimotor control strategy employed by humans during grasping.

In this study, five additional human subjects also participated in the same in-vivo grasping experiments, and the results are presented in Fig. S2. The consistent patterns observed across all six subjects ensure the broad applicability and generalizability of our conclusions. However, our study was designed around subject-specific modeling, tailoring various components, including the multi-level neural model for calculating human afferent tactile signals, biomechanical model, and soft robotic hand, to the characteristics of the initial participant. This approach ensured a high level of consistency across all study aspects. Furthermore, subject-specific data and models allowed us to maintain a uniform experimental setup, facilitating meaningful comparisons and insights into sensorimotor control strategy. It mitigated potential confounding variables associated with inter-individual variability. The dynamic transduction mechanism between the input tactile signals and output neural activation level was determined using the system identification algorithm[51]. Researchers have deduced that the selective responses of the SAI and FAI unit are critical for sensorimotor control[52,53]. To achieve a more accurate representation of the sensorimotor mechanism, the neuromorphic tactile signals of both the SAI and FAI tactile units were computed and integrated as the afferent tactile input. In terms of all the three gasping postures, the transduction function in terms of $\frac{a}{S^2+bS+c}$ achieved the largest goodness of fit between the predicted and biological neural activation levels among the functions adopting the different numbers of pole/zeros below the 10th order. Therefore, the transduction function of $\frac{a}{S^2+bS+c}$ was applied to represent the dynamic relationship between the afferent tactile input and the output activation level of muscle synergy during active grasping. A reactive grasping experiment was also conducted; a cylinder or sphere was grasped firmly by the human subject and a 20 gm weight was lifted to a specified altitude and dropped onto the grasped object (see Fig. S3). The subject was blindfolded and could only maintain stable grasping based on tactile feedback. The transduction mechanism between the input afferent neural dynamics evoked by the slippage and the reactive neural activation level muscle synergy were extracted through the system identification algorithm. The transduction function of $\frac{a}{S^3+bS^2+cS+d}$ was summarized to represent the sensorimotor strategy of reactive grasping. The values of the poles of these transduction functions are presented in Table. S1–10, and the comparison between the biological and computed neural activation level based on these transduction functions are shown in Fig. S4.

The population-level afferent tactile signals under active touch were computed through the validated multi-level numerical model and studied under the resulting perception in this study. The sensorimotor control mechanism between the input afferent post-synaptic cuneate neuron dynamics and the output activation neural activation level of muscle synergy was summarized and implemented on the ATSS to demonstrate its applicability and accuracy. Similar grasping and sensorimotor control performance with the human subject were achieved by the ATSS, based on the neuromorphic afferent tactile signals,

providing the possibility and a reliable process for developing the next generation neuroprosthetic restoration of human sensing and grasping capability.

## The sensing performance of the artificial tactile sensing system with the optimised tactile neural model

The signal of the 6 × 6 tactile sensor array was collected and processed by the optimized neural dynamic model to produce the neuromorphic tactile signals during active grasping. The discrimination accuracy of the human subject and ATSS were evaluated. The cylinders and spheres were required to be differentiated from a baseline cylinder or sphere with a diameter of 100 mm. it has been found by researchers that the response from SAI tactile units are critical to recognizing edges, corners, and curvature[54]. Therefore, the discrimination accuracy was estimated based on the neuromorphic tactile signals of SAI units. Passive stimuli (the same group of objects were used to compress the tactile sensor on the index fingertip with the same pressure experienced by the fingertip during active touch) were also placed on the tactile sensor to examine the effect of active touch on the sensing performance of the ATSS. Figure 2 shows the neuromorphic cutaneous afferent tactile signals of the 36 sensing units and the postsynaptic neural firing dynamics within the time duration of 1000 ms under active touch. The firing rates of these neuromorphic tactile signals for both SAI and FAI tactile units are shown in Fig. 3. The tactile signals during contact with cylinders and spheres ranging in diameter from 80 to 100 mm are depicted here. Tactile signals during contact with all objects are presented in Figures S5 to S8 in the supplementary material. The spiking rate of the artificial tactile units ranged from 29 to 52 spike/s and 32 to 57 spike/s in terms of cylindrical and spherical grasping, respectively. The largest firing rates of 52 and 57 spike/s were presented on the central sensing elements contacting cylinder and sphere with a diameter of 50 mm. Although firing rates of fewer than 45 spikes/s were obtained when in contact with objects with a diameter of 100 mm, more intense neural action potentials were observed when touching objects with a larger curvature, this relationship between firing rate and object curvature was observed in the post-synaptic cuneate neuron signals. A similar trend can also be observed in terms of the first spike latency, with smaller diameters resulting in shorter first spike latencies.

The signal detection theory was applied to quantify the discrimination accuracy of the ATSS according to the neural signal features of the rate coding and Victor−Purpura distance extracted from the 2nd order neuromorphic tactile signals. As is shown in Fig. 4, The hit rates based on neuromorphic tactile signals under active grasping agreed well with that of the human participant and the relative differences were below 10%. The discrimination accuracies increased with the curvature of the objects. The ATSS could differentiate the cylinder/sphere with a diameter of 90 mm from those of 100 mm with a hit rate above 68%, which was higher than that of the human subject. The disclination accuracy based on the neuromorphic signal tended to be lower than that of the human subject when discriminating objects with diameters smaller than 80 mm. A discrimination accuracy of above 94% was achieved by the ATSS when differentiating objects with diameters of 50 and 60 mm; conversely, the human subject could reach an accuracy of 100%. Most of the hit rates based on passive stimuli were lower than those under active touch. The MATLAB code for calculating the Victor−Purpura distance is provided in Data S1 of the supplementary material.

To evaluate the effectiveness of our approach compared to human tactile sensing, we conducted a comprehensive statistical analysis on neurodynamic features, focusing on firing rates and Victor−Purpura distances. Using the Mann-Whitney U test, suitable for non-normally distributed data, we tested the null hypothesis that there is no significant difference between these features and human

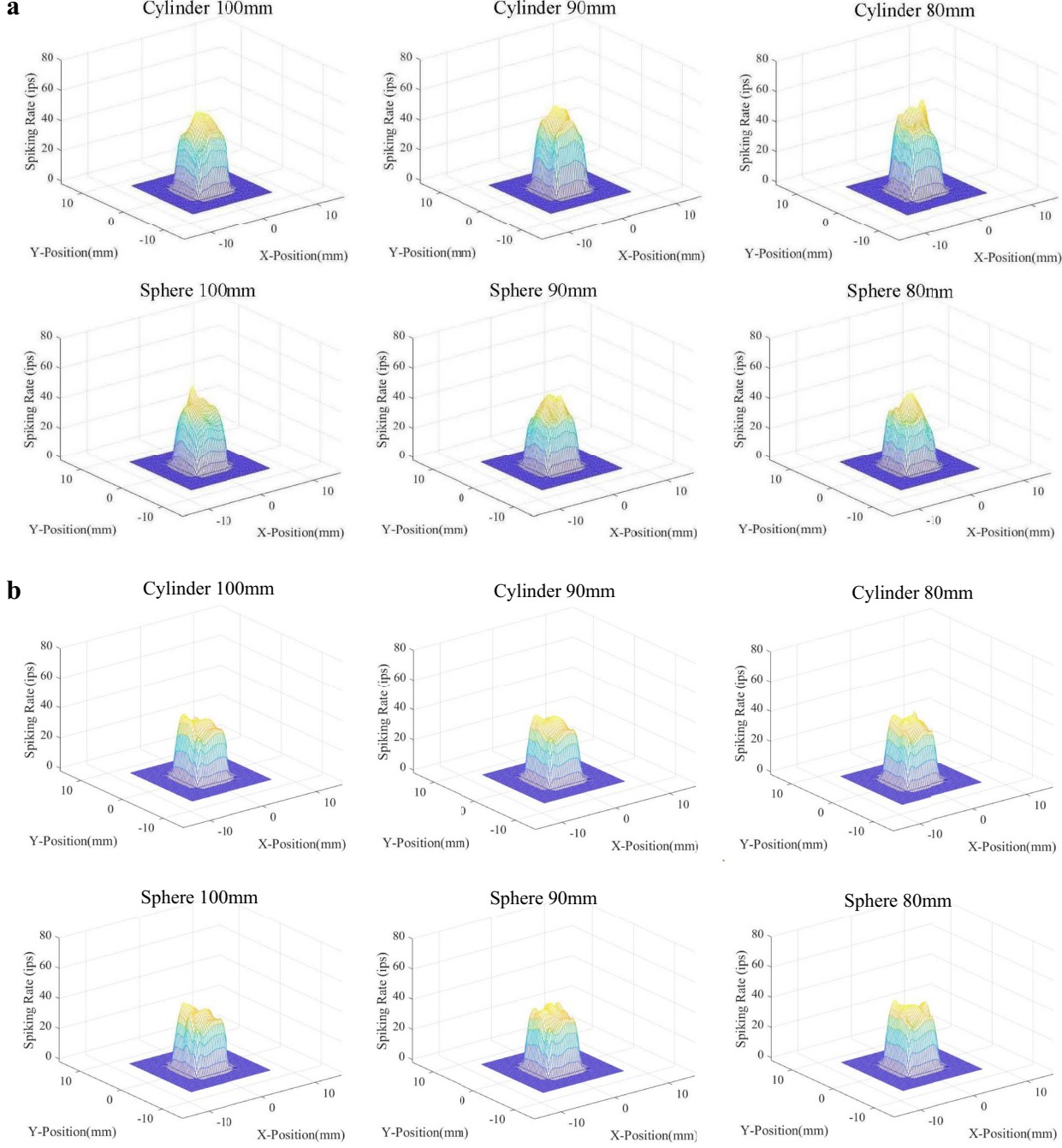

**Fig. 3 | The firing rates of the neuromorphic tactile signals.** The distribution of the spiking rate of the neuromorphic tactile signals computed over the 6 by 6 tactile sensing elements. The horizontal axis stands for the locations of tactile sensing elements within the contact area, the vertical axis is the spiking rate. **a** The spiking rate of the neuromorphic tactile signal elicited under cylindrical and spherical grasping for SAI. **b** The spiking rate of the neuromorphic tactile signal elicited under spherical grasping for FAI.

performance. The results showed significant differences, with p-values of <0.03 for firing rates and <0.01 for Victor–Purpura distances, indicating that Victor–Purpura distances outperform firing rates in distinguishing objects of different sizes. This superior discriminatory power of Victor–Purpura distances, highlighted by their lower p-value, demonstrates their effectiveness in mimicking human tactile sensing and suggests their potential to enhance artificial tactile systems. These findings validate our methods and underscore the potential of Victor–Purpura distances as a critical neurodynamic feature for advancing neuromorphic tactile feedback technologies in prosthetics, robotics, and neuroprosthetic development.

## Decoded sensorimotor transduction functions and their impact on artificial tactile sensing performance

The ATSS performed active and reactive grasping under the control of the summarized transduction functions. Transduction functions are mathematical models that convert complex sensory inputs into motor responses, encapsulating the dynamic relationship between afferent tactile signals and motor neuron signals. These models enable accurate representation and prediction of sensorimotor system behavior, as presented in Fig. 5. More transduction functions extracted based on more grasping trials are presented in the supplementary material. The biological and predicted neural activation level together with the

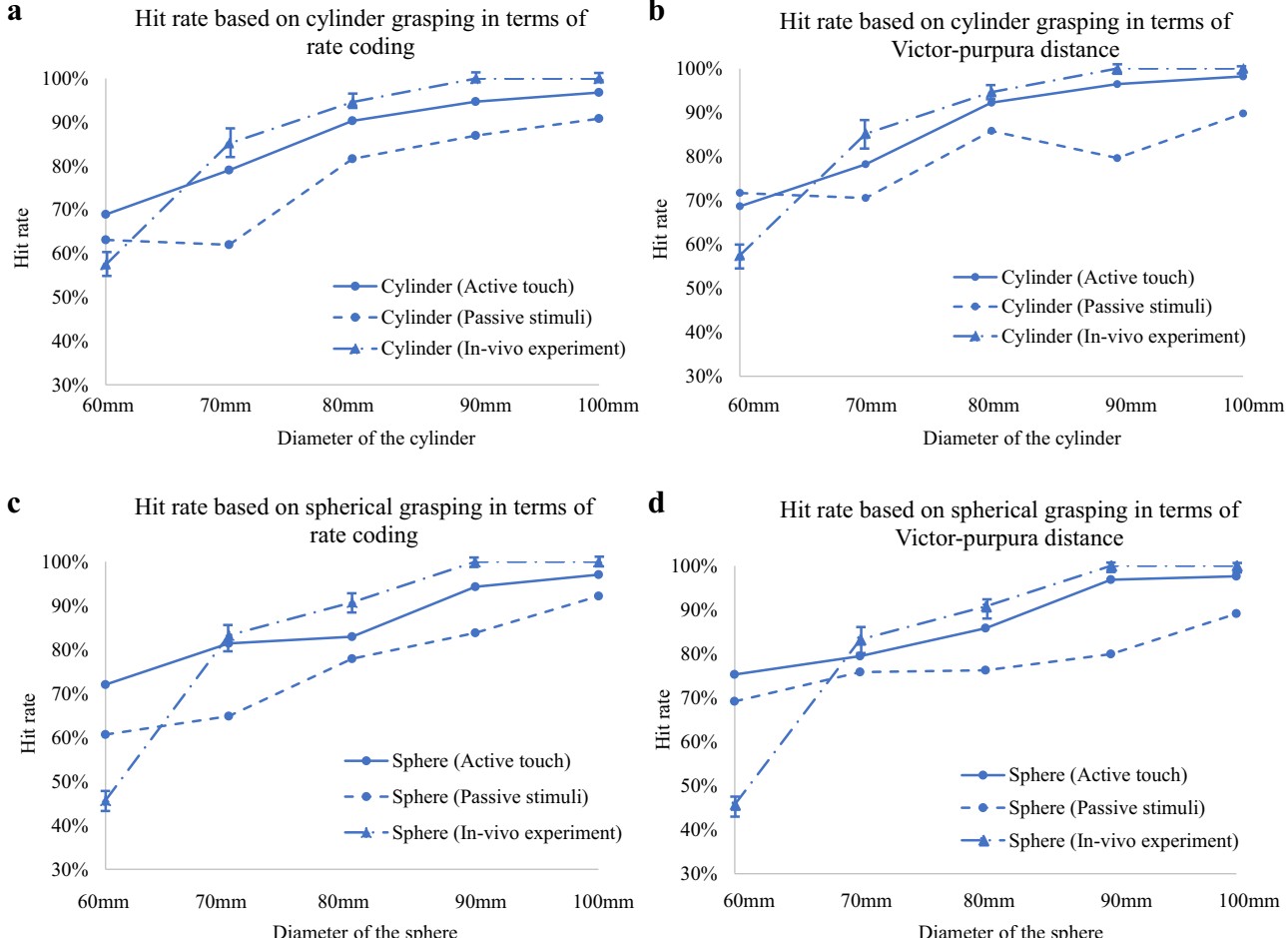

**Fig. 4 | Comparative discrimination accuracy (hit rate) of the artificial tactile sensory system (ATSS) against multi-subject human performance benchmarks.** **a** Hit rate based on cylinder grasping via rate coding displays the ATSS's capabilities for diameter identification through active touch, passive stimuli, and direct in-vivo experiments, incorporating the standard deviation across six subjects to emphasize consistency of performance. **b** Hit rate based on cylinder grasping using the Victor-Purpura distance demonstrates the ATSS's precision in timing-based discrimination tasks, compared to human subjects over a variety of cylinder diameters. **c** Hit rate for spherical grasping by rate coding highlights the ATSS's proficiency in distinguishing spherical diameters, detailing active and passive interaction modes, supplemented by in-vivo test data. **d** Hit rate for spherical grasping as determined by Victor-Purpura distance underscores the system's temporal resolution in identifying spheres, showcasing the system's effectiveness against the backdrop of human sensory and motor responses. The enhanced standard deviation representation for all panels underscores the robustness of the ATSS across diverse human experiences, illustrating a comprehensive view of the system's performance in alignment or deviation from the human subjects' tactile discrimination.

contact pressure under spherical grasping as shown in Fig. 6. Stable grasping with appropriate contact pressure was achieved by ATSS based on the sensorimotor control algorithm. Active and reactive grasping under 100% and 10% of the maximum voluntary contraction (MVC) forces were performed by ATSS to demonstrate the superiority of the sensorimotor function. Stable grasping was achieved under the MVC force, whereas a high contact pressure with intense neural spikes (see Fig. S9a) was initiated, which is unsuitable for the robust grasping of delicate objects. An unstable grasp was observed under 10% of the MVC force (see Fig. S9b), the biomimetic hand lost contact with the object after external impact. The contact pressure dropped to zero and no neuromorphic signal was observed. Therefore, the summarized sensorimotor control algorithm is critical for the restoration of human-like grasping performance on the biomimetic hand. The measured neural activation levels and those predicted based on the summarized transduction function for the other five subjects are presented in Fig. S10. Additionally, Fig. S11 displays the contact pressures exerted on the index fingers of both human and biomimetic hands during active and reactive grasping, as observed in the other five subjects. The active and reactive grasping performed on the ATSS is presented in

Movie S1. A dynamic force applied to the grasped ball demonstrates stable grasping modulated by the sensorimotor control strategy. An unstable grasping without sensorimotor control is also shown for comparison.

The pressure of the tactile sensor and spiking rate of the ATSS during active and reactive grasping were recorded and compared with those of the human subject to determine whether human-like contact mechanism could be achieved (see Fig. 7). The spiking rate of the 2nd order biological and neuromorphic afferent tactile signals are presented in Fig. 7a, and the relative difference between the spiking rates of the biological and neuromorphic tactile signals were below 10%. The firing rates of both the neuromorphic and human afferent signals were increased with the increased contact pressure, whereas the neuromorphic signals were more intense than the biological afferent neural dynamics. The response time between the unset of the slipping and intense neural signal feedback of the ATSS was higher than that of the human subject. The biological neural signal responded approximately 80 ms earlier than the ATSS after the impact of external stimuli. Similar magnitudes of pressure were observed on the biomimetic and human hand with a relative difference below 25% (see Fig. 7b). The ATSS

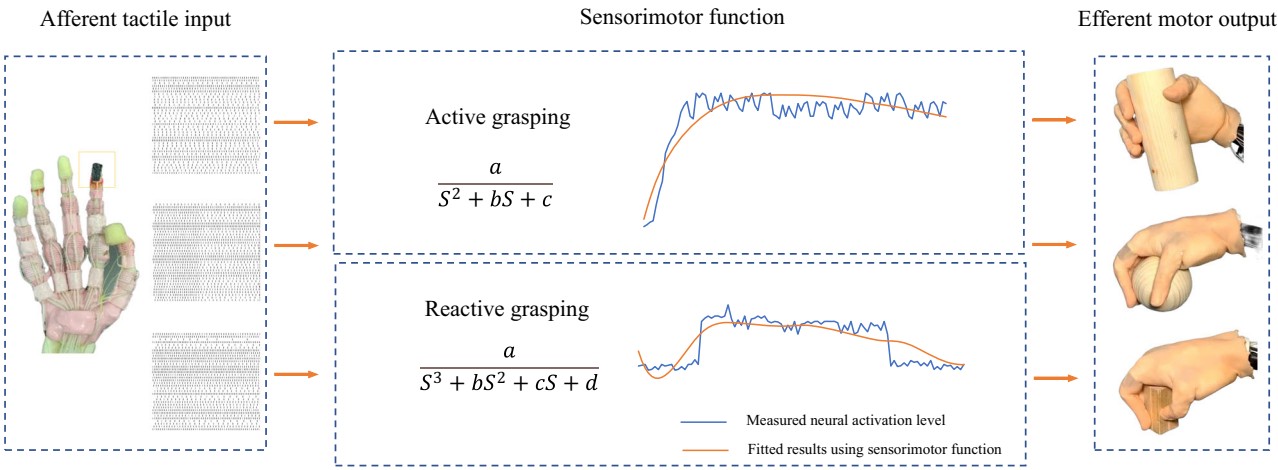

**Fig. 5 | Dynamic modelling of sensory input and motor output relationships.** This figure displays the mathematical functions derived from system identification techniques, quantifying the dynamic relationship between sensory inputs and motor outputs. Each curve represents a model fit, demonstrating how the system adapts to varying inputs and predicts motor responses, closely mimicking the adaptive responses observed in human neurophysiological processes. The transduction functions in this figure represent the dynamic relationship between afferent tactile signals and efferent motor responses, quantifying how sensory input (S) influences motor output over time and modelling the adaptive responses observed in human neurophysiological processes. The functions were derived using the 'System Identification' toolbox in MATLAB®, based on input-output data from our experiments. For active grasping, the function is $\frac{a}{S^2+bS+c}$, and for reactive grasping, the function is $\frac{a}{bS^3+cS^2+dS+e}$, where a, b, c, d, and e are parameters optimized to fit the observed data. These functions capture the nonlinear and complex dynamics between sensory inputs and motor outputs. The detailed development process is presented in the 'Methods' section, and the MATLAB code used to derive these functions is provided as Data S2 in the supplementary material.

displayed similar sensorimotor performance, neural dynamics, and contact mechanics with the human subject.

## Discussion

In-vivo afferent tactile and efferent motor signals have been studied over the past half-century[10–14]. Owing to the difficulties of measuring a large amount of afferent neural signals under the active touch condition, the dynamic relationship between the population-level afferent tactile signals and the motor neuron output under the sensorimotor control strategy remains unexplored. An effective method is required to overcome these obstacles. Recent research has focused on developing neuroprosthetic and robotics with simple sensing and reactive functions[5,7,22–25,30]. However, the hand dexterity and sensorimotor performance of human subjects are yet to be summarized and restored properly in robotics. The closed-loop control from cutaneous signals to efferent motor neurons should be explicitly studied to make a step further toward the application of next-generation prosthetics[2].

In this work, a multi-level numerical model developed in previous research was used to calculate the afferent tactile signals. This model offers comprehensive integration of active touch mechanics and detailed 3D geometry of the human hand, providing a more accurate and realistic simulation of tactile interactions compared to other numerical models in the literature[55]. Its multi-level simulation capabilities, from skin mechanics to neural firing and response dynamics, allow for in-depth analysis of tactile sensing mechanisms. Additionally, the enhanced predictive accuracy, especially in active touch scenarios, makes it a better choice for advanced applications in neuro-engineering and prosthetic development. The population-level human afferent cutaneous and post-synaptic cuneate neuron signals during active touch were computed and related under the resulting perception and motor neural signals in this study. The summarized sensorimotor strategy was then implemented and validated on a tendon-driven biomimetic hand. It was found that the transduction functions of $\frac{a}{S^2+bS+c}$ and $\frac{a}{bS^3+cS^2+dS+e}$ could fairly represent the dynamic relationship between the afferent post-synaptic signals and the efferent neural activation level of muscle synergy under active and reactive grasping, respectively. The transduction functions extracted from the neural activation levels of the other five human subjects are presented in Tables S11–20, the gender and age of all the subjects are presented in Table S21. Analysis revealed that the sensorimotor transduction functions are similar among the five subjects studied, with variations in the magnitudes of the poles in control strategy magnitudes below 15%. This similarity underscores the robustness and applicability of our sensorimotor control strategies across diverse individuals, enhancing the model's relevance for real-world prosthetic applications.

The tendon-driven biomimetic hand and tactile sensory array mounted on the distal index finger were implemented using the optimized neural dynamic model as the ATSS. This study presented a reliable method to restore human-like sensing and sensorimotor performance on robotics/prosthetics. The neuromorphic cutaneous signals under active touch were obtained and applied to differentiate cylinders and spheres with different sizes based on the rate coding and Victor–Purpura distance. The hit rates based on the neuromorphic tactile signals varied between 69% and 98% which were comparable with those of the human subject. The participant could differentiate cylinders or spheres with diameters of 60 mm and 50 mm from the baseline object with an accuracy of 100%. However, slightly lower discriminating accuracy was observed in the ATSS. This may be due to the adoption of the signal detection theory[56], which assumes that the 'signal' and 'noise' are normally distributed. The hit rate of the ATSS increased with the curvature of the object because the firing rate and Victor–Purpura distance are larger than those initiated by being in contact with the baseline object. Our previous research demonstrated that a more intense variation of hand contact strain/stress could be imitated when touching the object with a larger surface curvature[43]. This leads to a larger membrane current through cutaneous receptors and more intense neural spiking. The hit rates of the ATSS under passive stimuli were less than those under active touch based on rate coding and Victor–Purpura distance. Other researchers found that the hand contact biomechanics under active muscle modulation is different from that under passive external stimuli[57]. Different contact

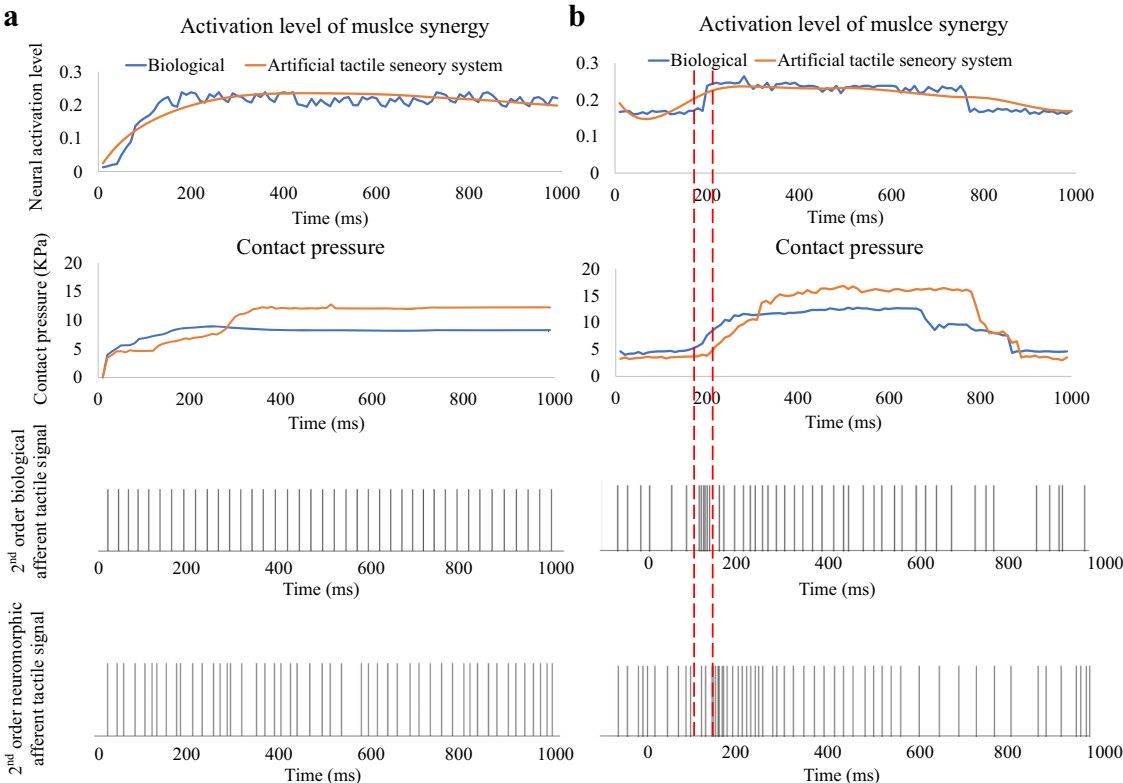

**Fig. 6 | Validation and performance comparison of the ATSS sensorimotor algorithm. a** Active Grasping Phase: The left column illustrates the synchronous evolution of neural activation level of muscle synergy and contact pressure captured by the ATSS during an active grasping task. Computed biological and ATSS-generated 2nd order tactile afferent signals are juxtaposed, revealing the system's ability to emulate human tactile feedback patterns. Here the 2nd order biological afferent tactile signals were computed through our numerical 1st and 2nd neuron model presented before. **b** Reactive Grasping Phase: The right column presents the reactive phase where the ATSS adjusts to sudden contact, indicated by the dashed lines, with delayed neural activation and pressure adaptation. This response is

compared with the biological benchmarks, with the lower charts displaying the corresponding afferent tactile signals. These graphs collectively demonstrate the temporal accuracy and the neuromorphic efficacy of the ATSS, simulating human like sensorimotor functions as per the summarized transduction function. Statistical evaluations using Mann-Whitney U tests reveal significant differences in firing rates and Victor-Purpura distances ($p < 0.03$ and $p < 0.01$, respectively), demonstrating the enhanced capability of Victor-Purpura distances in discriminating tactile stimuli. This supports their potential utility in developing advanced tactile feedback mechanisms for prosthetic and robotic applications.

biomechanics could affect the strain/stress distribution and the neural electro transduction mechanism of the mechanoreceptors[41,58], resulting in different neural dynamic features.

The summarized sensorimotor algorithm was implemented on the ATSS and the performance was compared with that of the human subject. The dynamic relationship between the input afferent postsynaptic tactile signals and output activation level of muscle synergy was applied to control the biomimetic hand. Similar contact mechanics and afferent tactile signals with the participant were obtained. The spiking rates of the neuromorphic tactile signals were more intense than those of the biological ones under most circumstances owing to the larger contact pressure experienced by the biomimetic fingertip during firm grasping compared with the human hand. The longer response time of the ATSS between the onset of external stimuli and feedback of the intense neuromorphic neural signals could be attributed to the time delay of the electric motors and tactile sensor array. Therefore, similar sensing and sensorimotor performance with the human subject were achieved by the ATSS based on the neuromorphic afferent tactile signals. The accuracy of these transduction functions representing the dynamic relationship between the afferent tactile input and efferent motor output was then validated. In this study, the multi-level numerical model predicting the population-level afferent tactile signals combined with the microneurography and validating experiment on robotics provided a reliable research method to study human tactile sensing and sensorimotor functions. The implication of

the validated neuromorphic tactile signals and sensorimotor algorithm on neuroprosthetic/robotics were also presented. Population-level afferent tactile signals were computed and studied under the resulting perception in this study. The sensorimotor control mechanism was explicitly summarized and implemented on the ATSS to validate its applicability and accuracy. Similar sensing capability and contact biomechanics to those of human subjects were achieved based on the efferent and afferent signals measured from all six human subjects. The grasping performance of the biomimetic hand was improved by integrating the sensorimotor algorithm. The response time of this control loop was affected by the size and weight of the objects. The dynamic relationship between the afferent tactile signals and the neural activation level of the forearm muscles can be effectively simplified as transduction functions and applied to the control of robotics in this research. The comparable sensing and sensorimotor performance based on the neuromorphic afferent tactile signals of this ATSS ensured the accuracy of the summarized biological sensorimotor function. Therefore, using a multi-level numerical model to compute afferent tactile signals for studying human tactile sensing and motor feedback was demonstrated to be effective and with the potential to provide a step towards the application of next generation neuroprosthetic.

This research primarily focuses on the intricate neural firing and response dynamics of tactile sensing and the integration of these sensory inputs into effective motor control, which is vital for

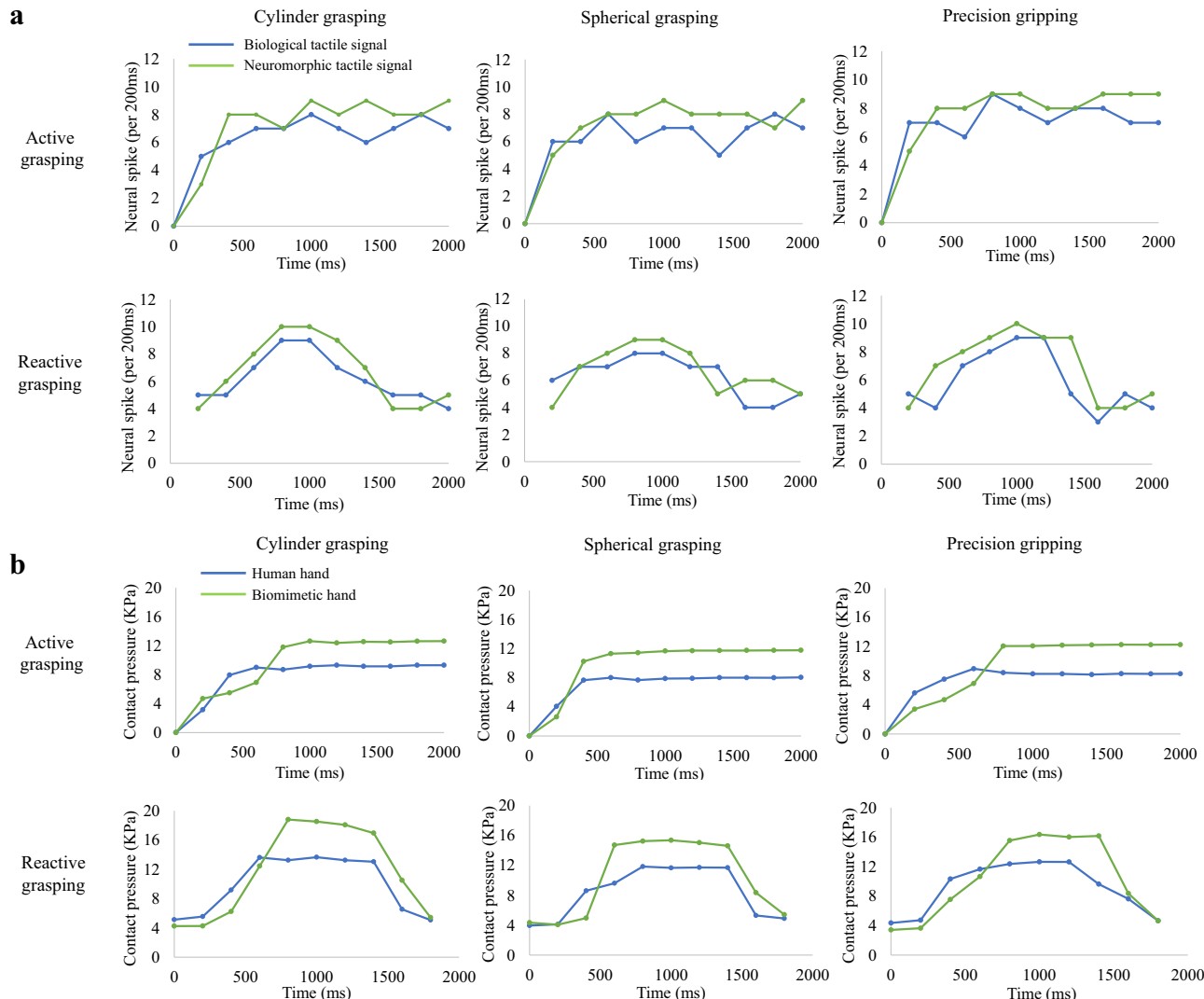

**Fig. 7 | The comparisons between the human subject and the ATSS in terms of the contact pressure and the neuromorphic/biological spiking rate. a** The spiking rates of the neuromorphic and biological afferent tactile signals under active and reactive grasping, the spiking times was counted every 200 ms. **b** The contact pressure on human fingertip was compared with that measured by the tactile sensor on biomimetic hand under the active and reactive grasping.

understanding how tactile sensations are processed and utilized in the complex domain of sensorimotor control—an essential aspect in the development of advanced prosthetics and rehabilitation techniques. Compared to other similar published works on closed-loop biomimetic hands or prosthetics, our study complements the ground-breaking research on real-time bidirectional hand prostheses by providing a deeper understanding of the neural mechanisms underlying sensory feedback loops. Compared to other published works on closed-loop biomimetic hands or prosthetics, this study enhances our previously developed multi-level neural dynamic model by integrating it with new experimental data on neural activation signals and applying updated sensorimotor controlling algorithms in a bio-robotic system, restoring the human-like hand performance. This approach improves the understanding of neural processing in prosthetic functionalities. Additionally, unlike other research developing tactile feedback systems, such as Raspopovic's practical sensory feedback applications in bidirectional hand prostheses[36], Ortiz-Catalan's exploration of long-term bionic hand adaptation[37], and Preatoni's investigation into multisensory perception[38], this research focuses specifically on tactile sensation and its direct translation into motor actions. This emphasis provides a detailed understanding of tactile-based sensorimotor integration, instrumental for tactile-based rehabilitation and therapeutic

interventions. While each of these prior works has made significant contributions[36,38,59], this study advances the field by providing comprehensive insights into the neural processing of tactile information and sensorimotor control mechanisms. It contributes to more effective tactile-based interventions, essential for enhancing prosthetic devices and rehabilitation techniques. Our work complements and extends other researchers' previous work, significantly contributing to this evolving field and paving the way for advanced prosthetic and rehabilitative solutions.

In summary, this study advances understanding of human tactile sensing and sensorimotor control, crucial for neuroprosthetic and robotics. A simplified transduction function has been developed that effectively translates afferent tactile signals into forearm muscle activation, enabling human-like sensorimotor performance on a biomimetic hand. This work complements and extends the prior research. The subject-specific sensorimotor system demonstrates significant adaptability for amputees by supporting the development of customized prosthetic limbs. By adjusting the geometry based on individual anatomical data and employing advanced imaging and 3D printing techniques, each prosthetic is tailored to meet specific user needs. This enhances prosthetic functionality and contributes to broader applications in robotic and biomedical engineering, paving a

way for more natural and effective prosthetic and rehabilitative solutions.

## Methods

### Development of tactile sensory system

To validate the summarized sensorimotor control strategy and make a step further towards the applications of neuroprosthetic, ATSS was developed for implementing the sensorimotor control and object recognition based on neuromorphic tactile signal feedback. The core component was a tactile sensor array fully 3D printed onto the distal index phalange of a tendon-driven biomimetics hand. The $6 \times 6$ sensing elements were connected with the customized electric circuit consisting of an Arduino board, shift registers, and multiplexers to collect the pressure signal for further processing (see Fig. S12 and Movie S1). A tendon-driven biomimetic hand containing the intact hand bone skeleton, interphalangeal ligaments, tendon and skin was employed as the main component of the ATSS. The skeleton of the hand was 3D printed using polylactic and the soft tissues were modelled using silicone rubber. Five electric motors (Dynamixel MX-12W, Robotics Inc.) were used to drive the biomimetic hand. The anthropomorphic size of the biomimetic hand was reconstructed based on the same subject recruited for the in-vivo grasping and microneurography experiment. This anatomical accuracy was crucial for our study. For the object recognition and actively grasping experiments, we employed cylindrical and spherical objects with varying diameters (ranging from 50 mm to 100 mm) as stimuli/grasping objects. The comprehensive details regarding the development of the tactile sensor, the design of the biomimetic hand, and the intricate multi-level numerical model employed for the computation of afferent tactile neural signals are provided below.

### Tactile sensor array

A tactile sensor with enhanced sensing capabilities, fabricated through a custom 3D printing process was integrated with the soft robotic hand as our artificial tactile sensory system. This tactile sensor was developed in our previous work[60], and more detailed information regarding its fabrication process and sensor performance is presented in. The sensor's fabrication commenced with the preparation of a graphene/carbon nanotube (CNT)/silicone rubber composite, selected for its piezoresistive and thermosensitive properties. To create the electrode material, we mixed silver-coated copper powder with silicone rubber, achieving optimal conductivity and printability. Employing a customized 3D printing platform, the tactile sensor was printed directly onto various surfaces, including an anthropomorphic robotic hand and human bone models. This direct printing process utilized the optimized graphene/CNT/silicone composite, ensuring a conformal fit and efficient fabrication. The sensor design features a dual-layered structure with upper and lower papilla-auxetic sensing layers, sandwiched between flexible electrode layers. When external pressure is applied, the change in the sensor's electrical conductivity is detected by these electrodes, allowing for precise pressure signaling. The integrated biomimetic interlock structure enhances the sensor's ability to discriminate between different directions of external stimuli, making it highly effective in applications requiring nuanced tactile feedback.

### Biomimetic Robotic hand

A bioinspired soft robotic hand, integrating several human-hand-like features to replicate the biomechanical advantages found in human fingers, served as the platform for the artificial tactile sensory system. This biomimetic robotic hand was developed in our previous work, and more detailed information about its development is provided in ref. 61. The robotic hand was constructed using a multilayer approach, emulating the structural components of a human finger. The base layer consists of 3D-printed phalanges and metacarpal bones, using UV white photopolymer resin, based on CT scanning data of a human

hand. This provides the necessary rigidity and serves as the foundation for further layers. The second layer comprises the capsuloligamentous structure, including artificial joint ligaments and capsules. Polyethylene terephthalate (PET) fiber ribbons, mimicking the crimp pattern of human ligaments, were sintered onto the bones, ensuring anatomic joint position and stiffness. Silicone rubber capsules, with triangular-shaped folds, were used to mimic human joint capsules, contributing to joint stability. The third layer involves the tendons and tendon sheaths. Tendon networks were fabricated using polyester Dacron fibers and fishing lines, while silicone rubber membranes represented the tendon sheaths. This layer replicates the complex tendon routing and functionality found in human hands. The design and materials employed in each layer contribute to the overall dexterity and adaptability of the robotic hand, making it suitable for the artificial tactile sensory system in this research that require human-like manipulation capabilities.

### Multilevel numerical model for computing the population-level afferent tactile signals

A comprehensive multi-level numerical model to simulate the tactile sensing mechanisms of the human hand during active touch was employed in this study to calculate the afferent tactile signals[43]. This model uniquely integrates finite element (FE) hand modeling with Izhikevich neural dynamic modeling to predict the behavior of first-order cutaneous neurons and their role in tactile perception. The Izhikevich model was selected due to its ability to capture the rich dynamics of neuronal behavior using a minimalistic set of equations and parameters. This model effectively balances biological plausibility and computational simplicity, making it an ideal choice for simulating complex neural dynamics without the computational burden associated with more detailed models. This multi-level numerical model was developed and validated in our previous research. Its performance was compared with other published models, such as 'TouchSim'. Detailed information on the development of the model and the advantages of predicting afferent tactile signals under active touch are explained in ref. 43.

The foundation of our model is the subject-specific FE human hand model[62]. This model accurately replicates the geometric and material properties of the human hand. The FE model incorporates the intricate mechanics of epidermis, dermis, subcutaneous tissue, and bones, facilitating a detailed analysis of stress and strain distribution during tactile interactions. On top of the mechanical model, the Izhikevich neural dynamic model was applied to simulate the neural response of the hand's cutaneous receptors. By integrating data from in-vivo microneurography, the model effectively predicts neural dynamics, including the action potentials of slowly adapting type I (SAI) and fast adapting type I (FAI) mechanoreceptors. These predictions are validated against experimental microneurography data, ensuring the model's accuracy and relevance. The model's capacity to predict tactile sensation is further enhanced by incorporating active touch scenarios. It accounts for the complex interaction of mechanical stimuli, skin mechanics, and neural response, providing insights into the tactile perception process under various conditions. This approach enables a better understanding on how the human hand perceives and interprets tactile information during active manipulation, making it a valuable tool for studying sensorimotor control and tactile perception in this study.

### Derivation of the sensorimotor transduction function

In the data collection phase of experiments, extensive datasets of tactile sensory input (S) and corresponding neural activation levels (output) were recorded using a tendon-driven biomimetic hand equipped with neuromorphic tactile sensors during reactive grasping tasks. Subsequently, employing system identification techniques, the intricate relationship between tactile sensory input (S) and neural activation

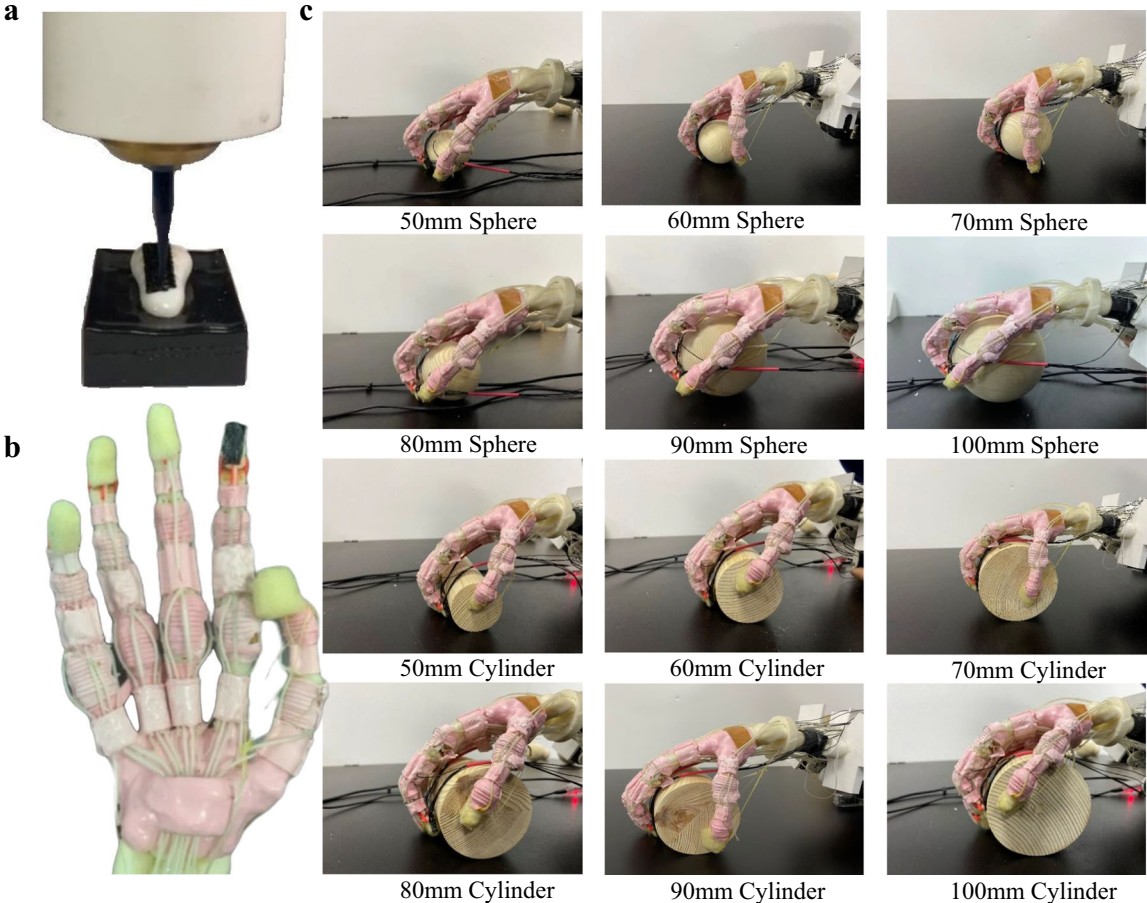

**Fig. 8 | The artificial tactile system with tactile sensor printed onto the biomimetic hand. a** The tactile sensor was 3D printed directly onto the index fingertip of the biomimetic hand. **b** The flexible tactile sensor (highlighted in circle) on the index finger of the biomimetic hand. **c** The active touch through the artificial tactile system for recognizing the sphere and cylinder with different diameters.

levels (output) was uncovered through meticulous analysis of the collected data and mathematical modeling. The mathematical model, taking the forms of $\frac{a}{S^2+bS+c}$ (active grasping) and $\frac{a}{bS^3+cS^2+dS+e}$ (reactive grasping) to represent the dynamics of sensorimotor strategy. The Laplace coefficients in these transduction functions were primarily determined through a data-driven approach. The 2nd order neuromorphic tactile signals were computed as the inputs of our transduction functions while the neural activation level extracted based on those EMG signals collected during in-vivo experiment from the human subject were employed as the output. These experiments provided us with a rich dataset of neural responses, tactile signals, and motor outputs during active and reactive grasping tasks. The system identification technique was then employed to extract the Laplace coefficients that best represented the relationship between afferent tactile signals and efferent motor neuron activations. The parameters related to active and reactive grasping were optimized against the neural activation level extracted based on the EMG signals collected during the in-vivo experimental results. Tuning these parameters involved an iterative optimization process to ensure that the model accurately mimicked the sensorimotor control strategy observed in human subjects during grasping tasks. The model parameters are unique to each subject but are derived using a consistent methodology. The parameters for the transduction functions related to active and reactive grasping for all six human subjects are specified in Table S1–20 of supplementary material. The MATLAB code for generating and optimizing those transduction functions for both active and reactive grasping were contained in Supplementary information, Data S2.

## Object reorganization and sensorimotor performance of ATSS

Object recognition was performed by the ATSS to demonstrate the accuracy of the neuromorphic tactile signals and relate it with the resulting perception. (see Movie S2). The capability of object recognition of the participant was also quantified and compared with that of the ATSS by calculating the discrimination accuracy of both sides to show its implications on the neuroprosthetic. Spheres and cylinders with diameters of 50, 60, 70, 80, 90, and 100 mm were used for object recognition. (see Fig. 8). The tactile sensing capacity of the human subject and ATSS were evaluated. The cylinders and spheres were required to be differentiated from a baseline cylinder or sphere with a diameter of 100 mm. The SAI tactile unit densely innervates the skin (~100 unit/cm²) which is sensitive to edges, corners, and curvature; their responses are critical for feature recognition. Therefore, the discrimination accuracy was estimated based on the neuromorphic tactile signals of SAI units. The neural dynamic features including the spiking rate and Victor–Purpura distance[63,64] were extracted as the input of the signal detection theory to quantify success recognition. Therefore, the neuromorphic signals of the SAI tactile unit elicited when touching the cylinders and spheres were collected and used for discriminating these objects. For instance, the spiking rate of the neuromorphic tactile signals evoked when touching the baseline object was regarded as 'noise', whereas the neural spikes elicited when touching the target object for discrimination was regarded as the 'signal'. To quantify the discrimination accuracy based on the Victor–Purpura distance, the 'noise' of the signal detection theory was defined as the Victor–Purpura distance among the tactile signals elicited when touching the baseline objects (cylinder or sphere with a

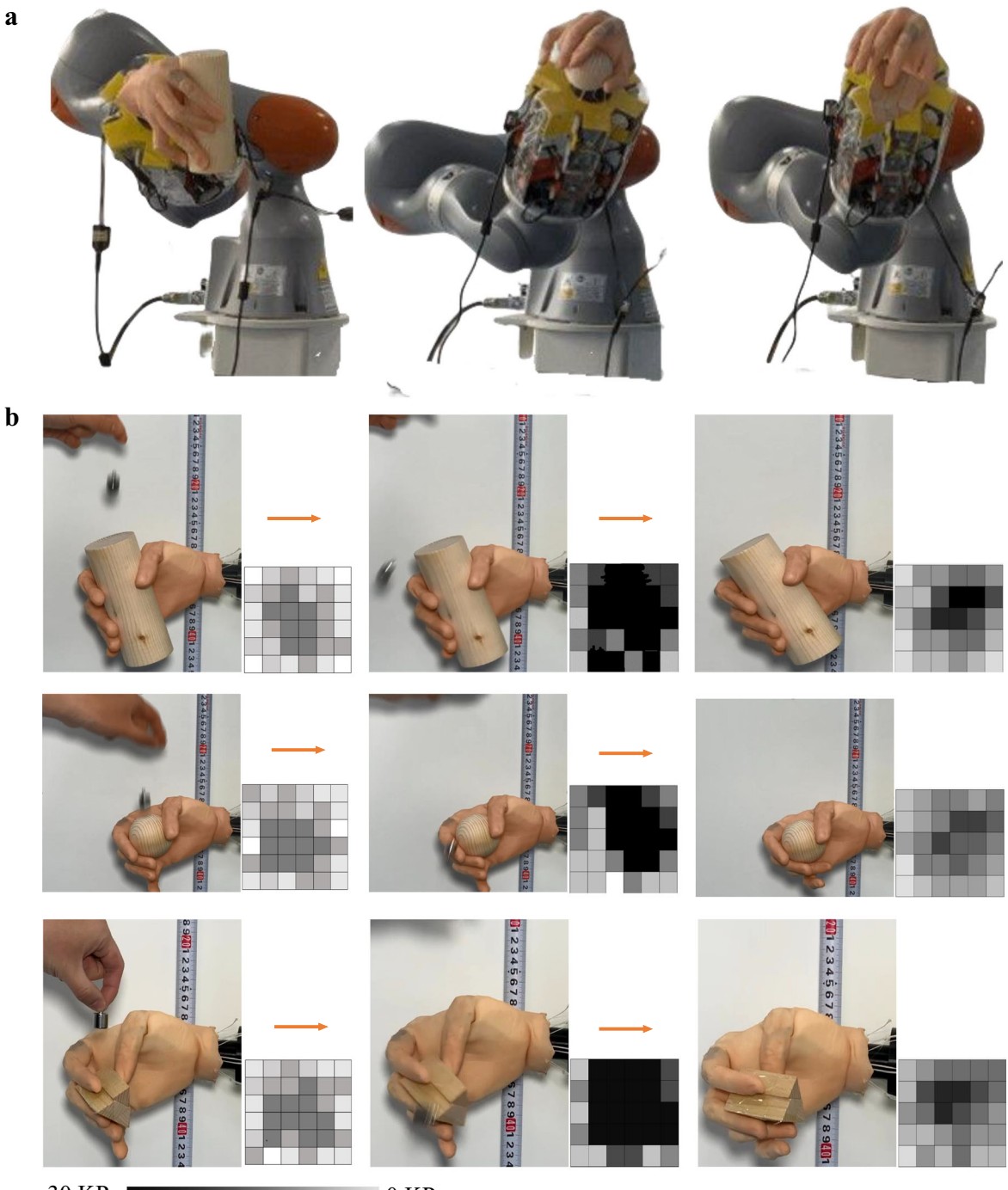

**Fig. 9 | The active and reactive grasping performed by this ATSS under sensorimotor control. a** Active grasping performed by this ATSS. Cylindrical, spherical grasping and precision gripping are presented. **b** Reactive grasping performed by this ATSS. A customized graphical user interface (GUI) was developed to visualize the pressure distribution across the 36 tactile sensing elements mounted on the index of the biomimetic hand. the magnitude of the contact pressure was visualized using the greyscale. The reactive grasping was presented in three stages: before impact, firm grasping after impact and normal stable grasping. A 20 g weight was lifted to a specified altitude and dropped onto the object, the shear force initiated by the slippage was detected. The grasping force of the biomimetic hand was increased under the modulation of sensorimotor control strategy.

diameter of 100 mm). The Victor–Purpura distances between the neural dynamics evoked by touching the target and baseline objects were regarded as the 'signal'. Each cylinder/sphere was touched 10 times. A total of 50 Victor–Purpura distances were computed to quantify the discriminating accuracy. The same method was adopted to compute the discrimination accuracy of recognizing the spheres with different diameters.

As is shown in Fig. S13, the neural dynamics evoked during contact with the baseline object were recorded. The mean and standard deviation of the spiking rates during 10 touches of the baseline object were summarized as the 'noise' signal in signal detection theory (SDT). Meanwhile, the mean and standard deviation of the spiking rates during contact with other objects, used to differentiate them from the baseline object, were regarded as the 'signal' in SDT. Discrimination

accuracy computed based on Victor-Purpura distance. The 'noise' signal in signal detection theory was defined as the Victor-Purpura distance among tactile signals during 10 touches of the baseline object (diameter 100 mm). Victor-Purpura distances between neural dynamics evoked when touching the baseline and other objects with diameters less than 100 mm were regarded as the 'signal' in SDT to evaluate discrimination accuracy. A total of 100 Victor-Purpura distances were computed. All cylinders were perceptually tested 10 times. The Victor-Purpura distances were calculated for differentiating the neuromorphic tactile signals evoked by contacting cylinders with diameters of 100 mm and 90 mm, 100 mm and 80 mm, 100 mm and 70 mm, 100 mm and 60 mm, 100 mm and 50 mm. The same method was used to compute discrimination accuracy for recognizing spheres with different diameters. Therefore, there were a total of 50 Victor-Purpura distances for differentiating cylinders and another 50 for spheres from the baseline objects. The detailed evaluation process of the discrimination accuracy is presented in Fig. S13 and these Victor-Purpura distances are given in Table S22 of the supplementary material. An in-vivo discrimination test was also conducted to quantify the tactile sensing capability of the human subject for comparison with ATSS.

Active and reactive grasping were performed by the ATSS to validate the accuracy of the sensorimotor algorithm and present its necessity on robotics/prosthetics control (see Fig. 9). The active grasping performed by the ATSS is shown in Fig. S14. In this study, active grasping is characterized by intentional object manipulation using precise, sensory-guided motor commands, showcasing the system's dexterity. Reactive grasping occurs in response to unexpected disturbances, requiring reflexive adjustments based on immediate feedback, which assesses the system's adaptability. Three grasping postures were selected and performed on the ATSS, handling round/cylindrical objects and pickup tasks, based on their prevalence in robotics and prosthetics research[65]. These tasks are crucial for evaluating prosthetic hand functionality, as they reflect common daily activities. They offer a measurable means to assess the sensory input and feedback mechanisms essential for advanced prosthetic design. This alignment with established benchmarks ensures that our study facilitates meaningful comparisons and advancements in the field. A python program (Data. S3) was developed for processing the pressure signal and controlling the motors to produce similar contraction forces with the human forearm muscles. The transduction functions of $\frac{a}{S^2 + bS + c}$ (active grasping) and $\frac{a}{bS^3 + cS^2 + dS + e}$ (reactive grasping) were applied to modulate the torque of the motors, imitating human hand sensorimotor performance based on the neuromorphic tactile feedback.

## In-vivo object reorganization and sensorimotor experiments

Three postures, including cylindrical and spherical grasping, as well as precision gripping, were involved. Objects used for grasping included spheres and cylinders with diameters of 50, 80, and 100 mm, and a triangular prism with uniform triangle base lengths of 10, 20, and 30 mm for precision gripping. Six human subjects were recruited to actively grasp the objects and hold them for 5 s. Among the subjects was a 24-year-old male, who was employed to provide DICOM images, microneurography data, and develop the numerical model for calculating afferent neural signals for the biomimetic robotic hand. Electromyography (EMG) signals of three muscles including the flexor digitorum profundus (FDP), flexor digitorum superficialis (FDS), and flexor pollicis longus (FPL) were recorded (see Fig. S3) using the Delsys Trigno (Delsys Inc., Boston, US). The kinematics of the hand were captured through Vicon Systems (Vicon Motion Systems Ltd, Oxford, UK). During the in-vivo object recognition and grasping experiments, the participant executed each grasping task (active and reactive) on

the objects a total of 10 times, consistent with the procedures performed by ATSS.

Regarding the general applicability of our simulation and experimental data, as well as the sensorimotor strategies and the entire robotic system, our subject-specific FE human hand model—initially developed from detailed CT scans and microneurography of a specific subject—is designed for adaptability across different individuals, including amputees. The model's geometric and material properties can be adjusted to match the unique anatomical data of new users, which is particularly crucial for amputees. For bilateral amputees, where no direct comparative model exists, linear scaling of existing 3D models can be employed to fabricate custom prosthetic components using techniques such as 3D printing. Additionally, for unilateral amputees, CT or MRI scans of the remaining limb provide a template for creating a mirror-image prosthetic. This standardized yet customizable approach allows our model to efficiently accommodate diverse anatomical variations, supporting its application in both clinical settings and biomedical research.

## Reporting summary

Further information on research design is available in the Nature Portfolio Reporting Summary linked to this article.

## Data availability

The Supplementary Figs. S1-S12, Tables S1-S22, and Supplementary Movies S1 and S2 are provided in the supplementary information. The sensorimotor performances of the biomimetic hand, modulated by the sensorimotor functions, are presented in Movie S1, while the active grasping performances are shown in Movie S2.

## Code availability

Data S1 and S2, which are the MATLAB codes for computing the Victor-Purpura distance and transduction functions, respectively, along with Data S3, the Python code for implementing sensorimotor control on the artificial tactile sensory system, are also included as supplementary information.

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

## Acknowledgements
We would like to thank our research group members in University of Manchester, University of Oxford, Aalto University, University of Liverpool, Liverpool John Moores University and Jilin University for their great support and assistance to this study. This work was supported in part by the National Natural Science Foundation of China (NSFC) under Grant 91948302 and Grant 52021003.

## Author contributions
Y.W., L.R. and G.W. conceived this study. Y.W. designed the overall research, performed the experiments, analysed the data, and wrote the manuscript. A.G.M. and F.M. provided scientific guidance, revised the manuscript, and helped perform the microneurography measurements. A.M., Y.Z. and L.Y. performed some experiments. LR and G.W. supervised the research work, provided guidance, analysed the data, and revised the manuscript.

## Competing interests
The authors declare no competing interests.
