## [Peer Review File · Nature Communications]

REVIEWER COMMENTS

Reviewer #2 (Remarks to the Author):

Even after thoroughly reading the article, many key aspects remain unclear. I cannot honestly say I fully grasp what the authors did and what they were trying to achieve.

Some notes that may be useful to the authors:

- While it is widely acknowledged that recruiting participants for prosthetic studies is difficult (and therefore low numbers of participants are often acceptable), this study uses a single participant, who grasps each cylinder only 10 times. This is not enough data to draw any meaningful conclusions. It should be fairly easy to include more able-bodied subjects in this study. It seems like the data presented here could be collected in a matter of minutes?

- The methods are significantly lacking. They do not contain nearly enough information to understand what the authors did, let alone properly evaluate whether the approach was scientifically sound.

- The authors fail to mention, let alone consider and incorporate, entire swaths of relevant literature (for instance, Sliman Bensmaia's work on simulating tactile afferent activity cannot be ignored here). These significant omissions prevent the authors from appropriately articulating the novelty of their work and how it relates to research by other groups.

My suggestion to the authors would be to rework this article, with a focus on answering a few simple questions:

- What experiments did you conduct? How were they set up? What were the participants doing? How many times?

- What analyses did you perform? How did you perform them (i.e. technical details)?

- What is the relevance of your work with respect to the literature on this topic. Some suggestions of relevant authors to read include: Sliman Bensmaia, Silvestro Micera, Stanisa Raspopovic, Max Ortiz-Catalan, etc.

Reviewer #3 (Remarks to the Author):

Human tactile sensing and sensorimotor mechanism: from afferent 1 tactile signals to efferent motor control

This is a well written, technically high quality work to model the sensory receptors, neural spiking activity and integrate these in a motor control system. This work is practical (design of a biomimetic sensor and motorized hand), computational (SNN model, as well as the systems level modeling and control).

There are some very interesting comprehensive aspects, e.g.

- Combining microneurography data
- Active touch
- Dynamical mechanical model (this is the more unique aspect, to close the loop from sensing to motor actions)

There are some add-ons, s

- They have developed tactile sensor array (not so unique)
- Hand/finger mechanism (not so unique)
- Interesting add on of reflex motion.

Much of the work builds on prior or well-known work on different fast, slow receptors, touch and even active touch (which is, nevertheless a good add-on). But the dynamical model gives makes this paper more unique or cutting edge for our consideration.

The paper has potential but overall it lacks any exciting result or conclusion. That is, the model for tactile is conventional neuromorphic/neurally inspired and acceptable but not significantly novel. What makes this work novel and useful is the closing the loop with the biomechanical model.

- The positive is that the model works, produces quality results mimicking biology, e.g. Figs. 9 and 10

- The most intriguing aspect to this reviewer is “The dynamic transduction mechanism between the input tactile signals and output neural activation level was determined using the system identification algorithm⁴⁴. But this is already published?

- How was the following derived? “The transduction function of aS^3+bS^2+cS+d was summarized to represent the sensorimotor strategy of reactive grasping. The values of the poles of these transduction functions are presented in Table.S1-10,...

-

But there are several weaknesses

- The biomechanical model, transfer function, is also quite empirical and not very informative.

- The performance result on grasping different objects is not very compelling

- The entire system is tested on only one subject. It is not applied to amputees.

There are several omissions of importance to understanding or replicating the work.

- The authors should explain how Izekevich model parameters were selected.

- They repeatedly mention “function. A subject-specific FE human hand model was applied to 133 simulate active touch. The strain energy density (SED)...”. But no details of the FEM model are given; indeed, this should be a very important supplementary material.

- What was the anatomical structure and detail?

- Was the model deformable?

- How are the tissue property (skin, muscle/fat, bone...even finer details) modeled and what data were used?

- What was the meshing details and algorithm?

Where to Eq. 5 and fit in Fig. 1? Is this the Layer 2 (EPSP)?

- Where is your cuneate model?

Extraction of motor output is important; the author employ synergy. This is a well established concept, but it is not justified. i.e. what was the rationale of using synergy (for different grasps) vs individual fingers? Mainly you need to describe synergy in terms of grasps – or even avoid, since you are really not breaking up tracked finger/hand motions into synergies.

On p. 8 you have the cuneate model, but its specifics, e.g. types of neurons, architecture (organization) etc is not explained. This is rushed, not clearly explained or justified.

I found Fig. 5 rather unhelpful...too much spike data, not resolvable, and not very meaningful. E.g. difference between objects is not differentiable and not interesting.

- You can hardly tell the difference with the active touch (what/when and how is different other than some vague idea of different spike patterns).

☐ Instead, it would have been useful to model the skin layer and the cuneate layer and show these data, i.e. how the cuneate layer 'helps'.

Fig. 6 is also hardly helpful. The hat-top images conform to what you'd expect. Nothing new or interesting. In addition,

- why are not all receptors (SA/FA) included?

- what would really help would be give temporal profile of the receptors near/around the object? (the shape aspect is not very helpful)

Fig. 7 is confusing. You mention Victo-Pappura distance but don't give any data. Then, you show hit-rate.

- There is no statistical test on the differences among the 3 methods.

It is not clear why the result, "The ATSS could differentiate the cylinder/sphere with a diameter of 90 mm from those of 100 mm with a hit rate above 68%," is so exciting or compelling.

Fig. 8 is interesting and novel for this reviewer; how were the model parameters (Laplace coefficients) obtained? How were active/reactive parameters tuned? This is mysterious.

- Also, It was found that the transduction functions of aS^2+bS+c and abS^3+cS^2+dS+e

fairly represents.... How was this model determined? There is a high empiricism (or mystery) to this.

Figs. 9 and 10 results are very good, and supportive.

Reviewer #4 (Remarks to the Author):

The paper proposes a closed loop anthropomorphic upper limb system encompassing spiking tactile sensors, artificial cuneate-like neurons and motor control strategies. As a whole, the attempt is interesting and has innovative potential. However, several details should be revised and added for the sake of providing scientific support to the technical claims and to enhance reproducibility, in my opinion.

The introductory sentences provide great emphasis on Cuneate nucleus analogue, however the present manuscript does not provide enough methodological details on the biological Cuneate data/model used to support the contribution of the paper. I suggest integrating these aspects. As an example, the following sentence would require more supporting methods: "The ATSS developed in this study exhibited similar sensing performance with the human subjects at the 2nd order cuneate neural signals stage".

On the other side, in some cases the results section contains too much recall of the methods, that instead I would leave in the methods only. As an example, in the section on "Sensorimotor control strategy" (results) the following paragraph is too much methodological and may be simplified, leaving the details in the actual methods section:

"Electromyography (EMG) signals of three muscles including the flexor digitorum profundus (FDP), flexor digitorum superficialis (FDS), and flexor pollicis longus (FPL) were recorded (see Fig. S1) using the Delsys Trigno (Delsys Inc., Boston, US). The kinematics of the hand were captured through Vicon Systems (Vicon Motion Systems Ltd., Oxford, UK). The neural activation level of the muscle synergy during active touch was extracted from the EMG signals using the non-negative matrix factorization algorithm (NMF) 43,44."

Figure 5 should better correlate the experimental conditions (manipulated object and motor control strategy). Moreover, the authors should better explain whether the raster plots depict experimental repetitions over a single channel, or a single experiment with multi-channel data. A more structured figure with multiple panels would help better understanding the claimed results.

Figure 7 is about an interesting behavioural comparison between machine performance and human performance. Instead, the previous figures focus on the signal outputs from the spiking artifact. I see a missing argument in the proposed machine-human similarity: human electrophysiological recordings at peripheral nervous system level could be an added value in the main figures of the paper.

Figure 9 shows an interesting comparison between 2nd order neuron firing in the developed artifact and biological recordings. However, I would suggest clarifying the meaning of "biological" (in terms of

species, recording techniques, ...) in the figure and related caption. The same comment applies to Figure 10. Such comment are in line with the general comment on Cuneate data, reported above.

Minor edits:

- In Fig. 10, please correct KPa to kPa, as the capital "K" stands for Kelvin, not for kilo
- In Fig. S3, please correct "seneory" in "sensory"

Response to Reviewers' Comments

Paper title: Human tactile sensing and sensorimotor mechanism: from afferent tactile signals to efferent motor control

Submitted to: Nature Communications

Manuscript number: NCOMMS-23-34899

General Response

We thank the reviewers for their constructive suggestions which have greatly helped us to improve the manuscript. We respond to the individual points in detail below, indicating the quality of the changes that we have made to the manuscript.

Specific Responses to Reviewer #2

1. *While it is widely acknowledged that recruiting participants for prosthetic studies is difficult (and therefore low numbers of participants are often acceptable), this study uses a single participant, who grasps each cylinder only 10 times. This is not enough data to draw any meaningful conclusions. It should be fairly easy to include more able-bodied subjects in this study. It seems like the data presented here could be collected in a matter of minutes.*

Response:

We appreciate the reviewer's concern regarding the use of a single participant in our study. While it is true that our primary experimental data presented in the manuscript were obtained from one participant, it's important to note that we have conducted similar *in-vivo* grasping experiments with an additional five human subjects. The results from these supplementary experiments consistently demonstrated very similar patterns in terms of EMG signal responses and sensorimotor control strategies.

All the six subjects were recruited (including the 24-year-old male mentioned in manuscript) to actively grasp the objects and hold them for 5s. Electromyography (EMG) signals of three muscles including the flexor digitorum profundus (FDP), flexor digitorum superficialis (FDS), and flexor pollicis longus (FPL) were recorded using the Delsys Trigno (Delsys Inc., Boston, US). The kinematics of the hand were captured through Vicon Systems (Vicon Motion Systems Ltd., Oxford, UK). The neural activation level of the muscle synergy during active touch was extracted from the EMG signals using the non-negative matrix factorization algorithm (NMF). The optimal

number of muscle synergy was regarded as its minimum value that achieved a mean variance account for (VAF) value above 85% with less than a 6% increment after adding another synergy. The muscle synergy dominating the motor control of the forearm muscles was recognized during active and reactive grasping in this study. The muscle synergy neural activation level was regarded as the output of the sensorimotor controlling strategy, whereas the input was the 2nd order afferent tactile signals.

The primary reason for focusing on the data from a single participant in the main manuscript is twofold:

Consistency and Subject-Specific Modelling: Our research approach emphasized subject-specific modelling. This means that various components of our study, including the multi-level neural model for calculating human afferent tactile signals, the biomechanical model, and the soft robotic hand, were all tailored to the specific characteristics of the initial participant. This subject-specific approach was a deliberate choice to ensure the highest level of consistency across all aspects of the study.

Advantage of Consistency: By using subject-specific data and models, we were able to maintain a high level of consistency in the experimental setup. This consistency allowed us to draw meaningful comparisons and insights into the sensorimotor control strategy without introducing potential confounding variables that might arise when comparing results across different individuals.

However, in response to the reviewer's valuable feedback and to further strengthen the comprehensiveness of our study, **we have now included the data and experimental results from all six subjects in the supplementary material** (Figure R1 below or Figure S4). This additional information not only supports the main findings but also provides a broader perspective on the consistency of our results across multiple participants.

Furthermore, in the updated manuscript, we have included a detailed discussion and clarification regarding the rationale behind our subject-specific approach and the significance of maintaining consistency throughout our study. We believe that these additions will address the concerns raised by the reviewer and enhance the overall quality and transparency of our research.

Fig. R1. Experimental results based on all six human subjects. Here, subject 1 is the 24-year-old male subject employed for developing the multi-level numerical model for computing afferent neural signals and developing the biomimetic hand in this study. The neural activation levels extracted from the EMG signals captured from all six subjects during active and reactive grasping are shown.

2. The methods are significantly lacking. They do not contain nearly enough information to understand what the authors did, let alone properly evaluate whether the approach was scientifically sound.

Response:

We appreciate the reviewer's feedback and emphasizing the importance of providing comprehensive methodological details to ensure transparency and scientific rigor.

The related research on the multi-level numerical model used for calculating human afferent tactile neural signals, the 3D-printed tactile sensor, and the biomimetic soft robotic hand employed in this artificial tactile sensory system was not published when we initially prepared this manuscript. However, so far, all the related work has been published. Hence, to better interpretate that methods used in this paper, we have now

included a brief explanation of the tools and methods used, and cited the relative work in the revised manuscript.

To address this concern, in detailed, we have made the following improvements:

1. **Tactile Sensor Array (Published on Communications Engineering, DOI: <https://doi.org/10.1038/s44172-023-00131-x>):** We expanded upon the methods used for the development of the tactile sensor array on the distal index phalange of the biomimetic hand. This included more detailed information on the materials, fabrication techniques, the customized electric circuit, including the specific components used, their connections, and their roles in collecting pressure signals. The detailed explanation below has been added to the updated manuscript:

A novel tactile sensor with enhanced sensing capabilities, fabricated through a custom 3D printing process was integrated with the soft robotic hand as our artificial tactile sensory system.

The sensor's fabrication commenced with the preparation of a graphene/carbon nanotube (CNT)/silicone rubber composite, selected for its piezoresistive and thermosensitive properties. To create the electrode material, we mixed silver-coated copper powder with silicone rubber, achieving optimal conductivity and printability.

Employing a customized 3D printing platform, the tactile sensor was printed directly onto various surfaces, including an anthropomorphic robotic hand and human bone models. This direct printing process utilized the optimized graphene/CNT/silicone composite, ensuring a conformal fit and efficient fabrication. The sensor design features a dual-layered structure with upper and lower papilla-auxetic sensing layers, sandwiched between flexible electrode layers. When external pressure is applied, the change in the sensor's electrical conductivity is detected by these electrodes, allowing for precise pressure signalling. The integrated biomimetic interlock structure enhances the sensor's ability to discriminate between different directions of external stimuli, making it highly effective in applications requiring nuanced tactile feedback.

2. **Biomimetic Robotic Hand (Published on IEEE Transaction on Robotics, DOI: 10.1109/TRO.2022.3200006):** We offered a more detailed explanation of the construction of the biomimetic hand, including the 3D printing materials, the modeling of soft tissues using silicone rubber, and the mechanics of the hand. The detailed explanation below has been added to the 'Method' section of our modified manuscript:

A bioinspired soft robotic hand, integrating several human-hand-like features to replicate the biomechanical advantages found in human fingers, served as the platform for the study of artificial tactile sensory system. The robotic hand was constructed using a multilayer approach, emulating the structural components of a human finger.

The base layer consists of 3D-printed phalanges and metacarpal bones, using UV white

photopolymer resin, based on CT scanning data of a human hand. This provides the necessary rigidity and serves as the foundation for further layers. The second layer comprises the capsuloligamentous structure, including artificial joint ligaments and capsules. Polyethylene terephthalate (PET) fiber ribbons, mimicking the crimp pattern of human ligaments, were sintered onto the bones, ensuring anatomic joint position and stiffness. Silicone rubber capsules, with triangular-shaped folds, were used to mimic human joint capsules, contributing to joint stability. The third layer involves the tendons and tendon sheaths. Tendon networks were fabricated using polyester Dacron fibers and fishing lines, while silicone rubber membranes represented the tendon sheaths. This layer replicates the complex tendon routing and functionality found in human hands. The design and materials employed in each layer contribute to the overall dexterity and adaptability of the robotic hand, making it suitable for the artificial tactile sensory system in this research that require human-like manipulation capabilities.

- 3. Multilevel numerical model (Published on IEEE Transaction on Biomedical Engineering, DOI: 10.1109/TBME.2022.3177006):** We also added explanations how the multi-level numerical model was used to calculate the human afferent tactile signals, the detailed explanation below has been added into the ‘Method’ section of the revised manuscript :

A comprehensive multi-level numerical model to simulate the tactile sensing mechanisms of the human hand during active touch was employed in this study to calculate the afferent tactile signals. This model uniquely integrates finite element (FE) hand modelling with Izhikevich neural dynamic modelling to predict the behaviour of first-order cutaneous neurons and their role in tactile perception.

The foundation of our model is the subject-specific FE human hand model. This model accurately replicates the geometric and material properties of the human hand. The FE model incorporates the intricate mechanics of epidermis, dermis, subcutaneous tissue, and bones, facilitating a detailed analysis of stress and strain distribution during tactile interactions. On top of the mechanical model, the Izhikevich neural dynamic model was applied to simulate the neural response of the hand's cutaneous receptors. By integrating data from in-vivo microneurography, the model effectively predicts neural dynamics, including the action potentials of slowly adapting type I (SAI) and fast adapting type I (FAI) mechanoreceptors. These predictions are validated against experimental microneurography data, ensuring the model's accuracy and relevance. The model's capacity to predict tactile sensation is further enhanced by incorporating active touch scenarios. It accounts for the complex interaction of mechanical stimuli, skin mechanics, and neural response, providing insights into the tactile perception process under various conditions. This approach enables us to understand better how the human hand perceives and interprets tactile information during active manipulation, making it a valuable tool for studying sensorimotor control and tactile perception in this study.

- 4. Sensorimotor Control Algorithm:** We elaborated on the sensorimotor control

algorithm used for active and reactive grasping. This involved a more in-depth explanation of the Python program developed for processing pressure signals and controlling the motors. And also, the different transduction functions representing sensorimotor control strategy extracted based on all the six subjects were added into the supplemental material.

5. **Data Availability:** We emphasized the availability of source data, supplementary figures, tables, and movies to facilitate a better understanding of the methods and results in the updated manuscript. The MATLAB codes for extracting the sensorimotor transduction function and the corresponding sample afferent/efferent signals were also included in the updated supplementary material to make sure that other researchers can also replicate our work.

By addressing these specific areas and providing more comprehensive information in the methods section, we aim to enhance the clarity and scientific rigor of our study. And now other researchers can completely replicate our work based on the detailed method and codes that were provided. We thank the reviewers for their valuable input, and these revisions will ensure that our approach is properly documented and scientifically sound.

3. The authors fail to mention, let alone consider and incorporate, entire swaths of relevant literature (for instance, Sliman Bensmaia's work on simulating tactile afferent activity cannot be ignored here). These significant omissions prevent the authors from appropriately articulating the novelty of their work and how it relates to research by other groups.

Response:

We appreciate the reviewer's feedback regarding the need to incorporate relevant literature to better contextualize our work and highlight its novelty in relation to research by other groups. We acknowledge the importance of recognizing and properly citing previous research, including the work from Sliman Bensmaia and others in the field of simulating tactile afferent activity. Actually we compared Sliman Bensmaia's computational model with this multi-level in our previous publication on IEEE Transaction on Biomedical Engineering (*DOI: 10.1109/TBME.2022.3177006*), and in this published manuscript we mentioned that the advantages and new achievements compare with the 'TouchSim' model developed by Sliman et al. :

Integration of Active Touch Mechanics: The new model incorporates the mechanics of active touch, including muscle-driven hand movements. This feature is pivotal for a more realistic simulation of tactile interactions, which is not addressed in the 'TouchSim' model, as 'TouchSim' primarily simulates passive stimuli.

3D Geometry and Hand Kinematics: The model includes a detailed 3D geometry of the human hand and finger pad kinematics. This level of detail enables a more accurate prediction of neural responses under various touch scenarios, surpassing the capabilities of 'TouchSim', which relies on simpler geometrical representations.

Multi-Level Simulation Capabilities: The numerical model operates on multiple levels – from skin mechanics to neural dynamics. This multi-level approach allows for a comprehensive analysis of tactile sensing mechanisms, a feature that 'TouchSim' lacks.

Enhanced Predictive Accuracy: The model shows improved predictive accuracy in simulating afferent tactile signals under passive stimuli when compared with 'TouchSim'. This is particularly notable in scenarios involving active touch, where the new model's advanced mechanics provide a more nuanced understanding of tactile interactions.

Potential for Advanced Applications: The inclusion of detailed hand mechanics and active touch simulation opens up possibilities for more advanced applications, such as the development of sophisticated prosthetic hands and haptic systems, which could benefit from the model's detailed representation of tactile interactions.

For this study, a validated numerical model was employed to calculate the biological and neuromorphic afferent tactile signals for the artificial tactile sensor system. The focus was on extracting a human-like sensorimotor controlling strategy and applying it to robotics and neuroprosthetic. To provide a clear and comprehensive presentation of the work, the revised manuscript highlights the advantages and improved performance of the multi-level numerical model for predicting afferent tactile signals during active touch. The detailed explanations below have been added to the updated manuscript:

In this work, a multi-level numerical model developed in previous research was used to calculate the afferent tactile signals. This model offers comprehensive integration of active touch mechanics and detailed 3D geometry of the human hand, providing a more accurate and realistic simulation of tactile interactions compared to other numerical models in the literature [34]. Its multi-level simulation capabilities, from skin mechanics to neural dynamics, allow for in-depth analysis of tactile sensing mechanisms. Additionally, the enhanced predictive accuracy, especially in active touch scenarios, makes it a better choice for advanced applications in neuro-engineering and prosthetic development.

By addressing these points, we aim to enhance the manuscript's clarity and demonstrate a better understanding of the broader research landscape. We thank the reviewer for pointing out this important aspect, and these revisions will ensure that our work is appropriately contextualized and acknowledges the relevant prior research.

4. My suggestion to the authors would be to rework this article, with a focus on answering a few simple questions: What experiments did you conduct? How were they set up? What were the participants doing? How many times? What analyses did you perform? How did you perform them (i.e. technical details)?

Response:

We appreciate the reviewer's suggestion to rework the article with a focus on providing a more detailed description of our experiments, including their setup, participant activities, and the frequency of repetitions. We agree that this will significantly enhance the clarity and transparency of our work. In the revised manuscript, we have addressed these as follows:

Experiments Conducted:

In the revised manuscript, we provided a more comprehensive and step-by-step description of the experiments conducted in our study. This included details on the setup, equipment used, and the specific tasks assigned to the participants. Specifically, we conducted experiments involving active and reactive grasping using the ATSS to investigate the closed-loop control of tactile signals to motor neuron outputs. Additionally, we conducted object recognition experiments in which the ATSS was tasked with differentiating between cylinders and spheres of various diameters. This part has been added to Line 576-581, Page 29 of the revised manuscript.

Experimental Setup:

The ATSS was designed as the central component of our experiments and was fully integrated into a tendon-driven biomimetic hand. The tactile sensor array, comprising 6x6 sensing elements, was meticulously 3D printed onto the distal index phalange of the biomimetic hand. The biomimetic hand itself was constructed to faithfully replicate the anatomical structure of the human hand, including the skeletal framework, interphalangeal ligaments, tendons, and soft tissues. This anatomical accuracy was crucial for our study. For the object recognition experiments, we employed cylindrical and spherical objects with varying diameters (ranging from 50 mm to 100 mm) as stimuli. This part has been added to Line 475-524, Page 25-27 of the revised manuscript.

Participant Tasks:

While our primary focus was on the development and validation of the ATSS for potential robotic and neuroprosthetic applications, we conducted *in-vivo* grasping experiments with one human participant.

The participant was tasked with performing both active and reactive grasping, allowing us to study the sensorimotor control strategies and the response time in these scenarios. This part has been added to Line 623-635, Page 31 of the revised manuscript.

Number of Repetitions or Trials:

During the *in-vivo* grasping experiments, the participant executed each grasping task

(active and reactive) on the objects a total of 10 times.

For the object recognition experiments, multiple trials were conducted for each object size (cylinders and spheres of different diameters. This part has been added to Line 600-608, Page 30 of the revised manuscript.

Technical Details of Analyses:

Computation of neural signals: We employed multi-level numerical models to calculate population-level afferent tactile signals.

Signal transduction functions: We determined the dynamic relationship between afferent tactile signals and neural activation levels using a system identification algorithm.

Muscle synergy extraction: The neural activation level of muscle synergy during active touch was extracted from electromyography (EMG) signals using the non-negative matrix factorization algorithm.

Active and Reactive Grasping Analysis: We implemented active and reactive grasping experiments using the ATSS, simulating sensorimotor control based on neuromorphic tactile feedback. The analysis involved torque modulation using transduction functions. In the revised manuscript, we will provide a step-by-step explanation of how the torque modulation was achieved, including the mathematical formulations and control algorithms employed.

Object Recognition Analysis: We conducted an object recognition analysis using the ATSS and the participant. This analysis involved collecting neuromorphic tactile signals, extracting features such as spiking rates and Victor–Purpura distance, and using signal detection theory for object discrimination. In the revised manuscript, we will describe the feature extraction process in detail, including the specific algorithms used and any relevant parameters.

Comparison with Human Performance: We compared the performance of the ATSS with that of the human participant. To do this, we calculated discrimination accuracy and grasping force-related metrics. In the revised manuscript, we will detail the specific metrics used, the calculations performed, and the statistical methods employed for comparisons.

By including these technical details in the revised manuscript, we aim to provide readers with a comprehensive understanding of the analytical procedures used in our study, thereby facilitating reproducibility and evaluation of the scientific rigor of our work.

5. *What is the relevance of your work with respect to the literature on this topic. Some suggestions of relevant authors to read include: Sliman Bensmaia, Silvestro Micera, Stanisa Raspopovic, Max Ortiz-Catalan, etc.*

Response:

We appreciate the reviewer's suggestion to consider the relevance of our work in the context of existing literature in the field of neuroprosthetics and tactile sensing. Indeed, our research builds upon and contributes to the ongoing advancements in this area. These three papers were all cited and discussed in our updated manuscript, here is how our work is relevant to the literature:

Our work primarily focuses on the intricate neural dynamics of tactile sensing and the integration of these sensory inputs into effective motor control. This research is vital for understanding how tactile sensations are processed and utilized in the complex domain of sensorimotor control, a crucial aspect in the development of advanced prosthetics and rehabilitation techniques.

Comparison with Raspopovic's Work: Raspopovic's pioneering research on real-time bidirectional hand prostheses, employing transversal multichannel intrafascicular electrodes, is a groundbreaking advancement in providing natural sensory feedback in prosthetic devices. Our study complements this by delving deeper into the neural mechanisms that underlie these sensory feedback loops. While Raspopovic's work demonstrates the practical application of sensory feedback in prosthetic control, our research provides the foundational understanding of the neural processing that enables such advanced functionalities.

Comparison with Max Ortiz-Catalan's Work: Ortiz-Catalan's research, particularly on the chronic use of sensitized bionic hands and their impact on the sense of touch, offers valuable insights into the long-term implications and adaptation processes associated with prosthetic usage. Our study extends this understanding by exploring how the brain processes afferent tactile signals and integrates them into efferent motor controls, which is critical for the continuous improvement and acceptance of such bionic devices.

Comparison with Greta Preatoni's Work: Preatoni's exploration of Full Body Illusion using multisensory platforms presents a novel perspective on sensory perception. In contrast, our research focuses on the specifics of tactile sensation and its direct translation into motor action, offering a more detailed view of the tactile aspects of sensorimotor integration, crucial for the development of tactile-based rehabilitation and therapeutic interventions.

In conclusion, our work addresses a critical aspect of sensorimotor integration, offering a comprehensive understanding of the neural dynamics involved in tactile sensing and motor control. This understanding is essential for enhancing the functionality of prosthetic devices and rehabilitation techniques, complementing and extending the

remarkable contributions of researchers like Raspopovic, Ortiz-Catalan, and Preatoni. We believe that our study significantly contributes to this evolving field, paving the way for more advanced and natural prosthetic and rehabilitative solutions.

The comprehensive discussion explicitly addressing the relevance of the work to the existing literature, including references to the authors mentioned above and their contributions. This discussion has been added to both introduction and discussion session (Line 77-83, Page 4 and Line 455-459, Page 24 of the revised manuscript). This will assist readers in gaining a clearer understanding of the novelty and contributions of the research within the broader context of neuroprosthetic and tactile sensing.

Specific Responses to Reviewer #3

1. The paper has potential but overall, it lacks any exciting result or conclusion. That is, the model for tactile is conventional neuromorphic/neutrally inspired and acceptable but not significantly novel. What makes this work novel and useful is the closing the loop with the biomechanical model.

Response:

Many thanks for the comments. We agree that the core neuromorphic model for tactile sensing may not be significantly novel on its own. We have incorporated additional crucial experimental results, which encompass the temporal cues of afferent tactile signals and the neural dynamics of both SAI and FAI mechanoreceptors (refer to Fig. R2, 3, or Fig. 6). Additionally, we have included cuneate neuron signals (2nd order afferent tactile signals), as illustrated in Figure 5, to demonstrate the functioning of our multi-level numerical model in calculating neural signals during active touch.

In addition, we also would like to highlight the specific aspects that make our study novel and valuable, particularly in closing the loop with the biomechanical model:

Integration of Neuromorphic Tactile Sensing: While the concept of neuromorphic tactile sensing has been explored in the literature, our study takes a step further by implementing this concept in a practical and applied manner. We 3D printed a tactile sensor array onto a tendon-driven biomimetic hand, allowing us to capture tactile data directly from the hand's interactions with objects. This integration of neuromorphic tactile sensors onto a physical hand model is a unique aspect of our research.

Sensorimotor Control Strategy: Our study goes beyond the mere simulation of tactile data and incorporates a sensorimotor control strategy. We use the neuromorphic tactile feedback to control the biomimetic hand in both active and reactive grasping tasks. This sensorimotor control strategy, based on the dynamic relationship between tactile input and motor output, demonstrates the feasibility of applying neuromorphic sensing in real-world tasks.

Biomechanical Model Integration: The key novelty of our work lies in closing the loop between the neuromorphic tactile sensing and a biomechanical model of the hand. By doing so, we create a closed-loop system that mimics the interaction between sensory information and motor control in a human hand. This integration allows us to study and validate sensorimotor strategies and grasp performance, offering insights into how humans use tactile information for object recognition and manipulation.

Applications in Neuroprosthetic and Robotics: Our research has direct implications for the fields of neuroprosthetic and robotics. By demonstrating the accuracy of our sensorimotor control strategy and object recognition based on neuromorphic tactile

signals, we provide a valuable step towards the development of next-generation prosthetics and robotic systems with human-like tactile sensing and control capabilities.

Further literature review on the work related to neuroprosthetic to offer a more comprehensive presentation of the exciting highlights of our work: This understanding is essential for enhancing the functionality of prosthetic devices and rehabilitation techniques, complementing and extending the remarkable contributions of researchers like Raspopovic, Ortiz-Catalan, and Preatoni. We believe that our study significantly contributes to this evolving field, paving the way for more advanced and natural prosthetic and rehabilitative solutions.

In conclusion, the novel contributions on the pioneering approach integrating a finite element hand model with a neural dynamic model, finely tuned using microneurography data. This unique fusion goes beyond boundaries, enabling the prediction of collective responses of cutaneous neurons during active touch. In a transformative revelation, the dynamic interplay between afferent tactile signals and neural activation levels of forearm muscles converges into concise transduction functions. These functions empower the replication of human-like sensorimotor performance on a biomimetic hand, propelling humanity closer to the next era of prosthetics endowed with neuromorphic tactile feedback. This work transcends convention, amalgamating human touch with cutting-edge biomimetic control strategies. It reveals unprecedented prospects in prosthetic design, robotics, and the understanding of the mind-body connection. Join this captivating journey, where humanity and technology unite through the marvel of tactile sensation, shaping the future of human-technology synergy.

These descriptions and summary above of our significantly novel contributions of our work has been added to Line 445-473, Page 24-25 of the revised manuscript. In summary, while the neuromorphic model for tactile sensing may be conventional, our work's novelty and utility stem from the integration of this model with a biomechanical hand model and the practical application of sensorimotor control. This approach allows us to bridge the gap between sensory input and motor output, offering insights and potential applications in neuroprosthetic and robotics. The additional description above can assist readers in gaining a straightforward and direct understanding of the novel highlights of this work. Furthermore, the abstract has been rewritten to provide a more exciting statement regarding our innovative achievements as is shown below:

Abstract

In the realm of tactile sensing, a critical frontier emerges: the intricate journey from afferent tactile signals to efferent motor commands. This pioneering approach integrates a finite element hand model with a neural dynamic model, finely tuned using microneurography data. This unique fusion goes beyond boundaries, enabling the prediction of collective responses of cutaneous neurons during active touch. In a transformative revelation, the dynamic interplay between afferent tactile signals and

neural activation levels of forearm muscles converges into concise transduction functions. These functions empower the replication of human-like sensorimotor performance on a biomimetic hand, propelling humanity closer to the next era of prosthetics endowed with neuromorphic tactile feedback. Discoveries extend beyond prosthetics, unveiling shared sensorimotor strategies among human subjects. In gripping experiments, remarkably similar approaches emerge as individuals manipulate objects. The tempo of this control varies with object size and weight, enhancing comprehension of sensorimotor dynamics. This work transcends convention, amalgamating human touch with cutting-edge biomimetic control strategies. It reveals unprecedented prospects in prosthetic design, robotics, and the understanding of the mind-body connection. Join this captivating journey, where humanity and technology unite through the marvel of tactile sensation, shaping the future of human-technology synergy. The successful development of the artificial tactile sensory system with human-like sensing and grasping performance in this research represents a significant milestone. It underscores a profound realization: the intricate connection between sensory perception and motor control, once considered insurmountable, is not an impenetrable enigma. Instead, it is a dynamic synergy that can be navigated with precision and grace, poised to redefine prosthetics and robotics through the essence of human touch.

2. The positive is that the model works, produces quality results mimicking biology, e.g. Figs. 9 and 10. The most intriguing aspect to this reviewer is "The dynamic transduction mechanism between the input tactile signals and output neural activation level was determined using the system identification algorithm⁴⁴. But this is already published?"

Response:

Thank you for your thoughtful inquiry regarding the application of the system identification method in our work, particularly in reference to the dynamic transduction mechanism between input tactile signals and output neural activation levels.

Indeed, the system identification technique we employed is based on the methodology presented in the referenced paper (Fagergren et al., "Precision Grip Force Dynamics: A System Identification Approach"). This approach is not novel in itself, as it has been previously published and applied in various contexts. However, the novelty and significance of our work lie not in the use of the method but in its unique application to the specific domain of tactile sensing and neural activation.

In our study, we adapted this technique to a different and complex setting: the intricate dynamics of tactile sensing in the human sensory system and its translation into neural activation patterns. The application of system identification in this context is innovative and reveals new insights that are distinct from the original context of precision grip force dynamics.

Our work extends the application of this method to a new domain, offering valuable contributions to our understanding of tactile signal processing and its impact on neural activation. This approach has allowed us to unveil specific characteristics and dynamics unique to tactile sensing and sensorimotor control, providing a fresh perspective and advancing the field in a novel direction. The novelty of our work lies in the integration of neuromorphic tactile sensors with a biomechanical hand model. The system identification algorithm is employed to characterize the relationship between sensory input (tactile signals) and motor output (neural activation level) within this closed-loop system. This integration and its application to the sensorimotor control of a physical hand model make our work unique. Our study not only applies the system identification algorithm but also validates its effectiveness in a real-world scenario. We use the algorithm to develop a sensorimotor control strategy for active and reactive grasping tasks. The quality results presented in Figs. 9 and 10 demonstrate the algorithm's utility in achieving human-like grasp performance based on tactile feedback. We believe that the application of established methods to new, unexplored areas is a vital aspect of scientific progress, and our work exemplifies this principle by bridging the gap between tactile signal processing and neural response analysis.

We appreciate the reviewer's interest in our study and their observation regarding the system identification algorithm. While the system identification algorithm itself may have been previously published in various contexts, it's important to emphasize the unique application and contribution of this algorithm within the context of our research. In summary, while the system identification algorithm itself may have been previously published, its application within the specific context of our research, including the integration with neuromorphic tactile sensors and the validation in a practical sensorimotor control scenario, represents a novel contribution. We believe this application of the algorithm adds value to the field of neuromorphic sensing and prosthetics.

3. How was the following derived? “The transduction function of aS^3+bS^2+cS+d was summarized to represent the sensorimotor strategy of reactive grasping. The values of the poles of these transduction functions are presented in Table.S1-10?”

Response:

The transduction function aS^3+bS^2+cS+d , which represents the sensorimotor strategy for reactive grasping, was derived through a systematic process based on the dynamics of our neuromorphic tactile feedback system. Below is a brief explanation of how this transduction function and its pole values in Table S1-10 were determined:

Data Collection: We collected extensive data from our experiments, which involved the use of a tendon-driven biomimetic hand equipped with neuromorphic tactile sensors.

During these experiments, the hand performed reactive grasping tasks, and we recorded both tactile sensory input and corresponding neural activation levels.

System Identification: The first step was to apply system identification techniques to understand the relationship between the tactile sensory input (S) and the neural activation levels (output). This involved analyzing the recorded data and using mathematical modeling techniques.

Model Fitting: We fitted a mathematical model to the data, and in this case, the model took the form of aS^3+bS^2+cS+d to represent the dynamics of the sensorimotor strategy for reactive grasping. The parameters a , b , c , and d were determined through a fitting process that minimized the error between the model predictions and the actual data.

Pole Extraction: Once we had the model aS^3+bS^2+cS+d , we could extract its poles. In control theory, poles are essential because they describe the system's dynamic behavior, including stability and response time. The values of these poles were obtained through mathematical analysis of the model.

The detailed explanation of how we extracted the transduction function has been added to 'Method' section on Line 552-574, Page 28-29 of the revised manuscript. And to provide transparency and allow for replication of our work, we presented the values of the poles in Table S1-10 in our manuscript. This table serves as a reference for readers to understand the characteristics of the derived transduction functions. In summary, the process involved data collection, system identification, model fitting, and subsequent extraction of pole values from the mathematical model aS^3+bS^2+cS+d . These poles are crucial for understanding the dynamic behavior of the sensorimotor strategy for reactive grasping in our neuromorphic tactile feedback system.

4. The biomechanical model, transfer function, is also quite empirical and not very informative.

Response:

We appreciate the reviewer's feedback and would like to address the concerns regarding the empirical nature of our biomechanical model and its informativeness. Our biomechanical model serves as an essential component in our study, and we developed it based on a combination of experimental data and our previously established principles of biomechanics. Your comment highlights an important aspect of our research, and I appreciate the opportunity to provide further clarity.

In our research, we developed a comprehensive multi-level numerical model that integrates a biomechanically detailed Finite Element (FE) hand model with a neural dynamic model. This integration is detailed in our two publications: "Subject-Specific

Finite Element Modelling of the Human Hand Complex: Muscle-Driven Simulations and Experimental Validation" and "Predicting Afferent Neural Dynamics During Active Touch and Perception: A Multi-level Numerical Model".

The FE hand model, as elaborated in our first publication, forms the foundation of our biomechanical analysis. It incorporates the detailed anatomy of the hand, including bones, ligaments, tendons, and other soft tissues, providing a highly accurate representation of the hand's biomechanics. This model is instrumental in simulating the mechanical responses of the hand to various stimuli, capturing the complexities of hand movements and interactions with objects. Our second publication focuses on the integration of this FE hand model with a neural dynamic model. The transfer function in this integrated model plays a critical role in translating biomechanical stimuli into neural signals. This function is derived from a combination of empirical data and theoretical understanding of tactile transduction mechanisms.

To calculate the transfer function, we first utilize the FE hand model to simulate the mechanical responses of the hand under various conditions. These responses include parameters like strain, stress, and strain energy density. The neural dynamic model then processes these biomechanical outputs, especially the strain energy density, and translates them into predicted afferent tactile neural activities using our optimized Izhikevich neural model. These calculated afferent tactile signals are subsequently employed as inputs for the transduction functions. The output efferent neural activation levels, which control the muscles for manipulation and reactive control, were measured during our in-vivo grasping experiment and serve as the outputs of the transduction function. The transfer function encapsulates the complex relationship between the mechanical stimuli or the afferent neural signals (input) and the efferent motor signals (output), relying on empirical data derived from experiments and our validated/published multi-level numerical model.

While the transfer function is empirical, it is grounded in extensive experimental validation, ensuring that its predictions are consistent with observed neural responses to tactile stimuli. This empirical approach allows us to accommodate the nonlinear and dynamic nature of tactile transduction, which is often challenging to capture through purely theoretical models. In summary, our multi-level numerical model, developed by combining the FE hand model with the neural dynamic model, provides a comprehensive tool for understanding and predicting the interaction between biomechanical stimuli and afferent neural signals. The empirical nature of the transfer function, while a point of complexity, is a necessary aspect of modeling the intricate and dynamic processes involved in tactile sensation and perception.

We have included the detailed explanations above in the 'Methods' section of the revised manuscript to further enhance our paper, providing more comprehensive information on how we utilized a previously published numerical model to restore human-like sensorimotor performance in the bionic soft robotic system.

5. *The performance result on grasping different objects is not very compelling, the entire system is tested on only one subject. It is not applied to amputees.*

Response:

We appreciate the reviewer's feedback and comment regarding the perceived lack of compelling performance results in our study. We would like to provide additional context to clarify the significance of our findings:

Limited Participant Data: We appreciate the reviewer's concern about using only one participant in our study. While our primary data were obtained from one participant, we conducted similar experiments with five additional subjects, consistently observing similar patterns in EMG signals and sensorimotor control. All six subjects actively grasped objects for 5 seconds, with EMG signals from three muscles (FDP, FDS, FPL) recorded. Hand kinematics were captured through Vicon Systems. Muscle synergy for forearm muscles during grasping was extracted via NMF. It was the output of sensorimotor control, with 2nd order afferent tactile signals as input.

Focusing on one participant in the main manuscript served two key purposes: Consistency and Subject-Specific Modeling: Our approach emphasized subject-specific modeling across all study components, ensuring consistency. Advantage of Consistency: Subject-specific data and models maintained experimental consistency, facilitating meaningful comparisons.

To address the concern, we've included data from all six subjects in the supplementary material. Additionally, we've clarified the rationale for our subject-specific approach and its significance in the updated manuscript. This single-subject concern was also addressed in response to question 1 from reviewer 2 above. A more detailed revised information can be found there.

Comparison with Human Performance: In our study, we compared the grasping performance of the ATSS with that of the human participant. The ATSS successfully imitated human-like tactile sensing and sensorimotor performance, achieving accuracy levels comparable to those of the human subject. Specifically, the ATSS was able to differentiate objects with different sizes and shapes with hit rates ranging from 69% to 98%, and it achieved 100% accuracy in distinguishing certain objects. This demonstrates the potential of our approach to replicate human sensorimotor capabilities.

Relevance to Neuroprosthetics and Robotics: Our study's primary aim was to develop and validate a sensorimotor control strategy for neuroprosthetics and robotics, with an emphasis on closing the loop between tactile sensing and motor control. While the current study focused on a limited set of grasping tasks, the significance lies in the foundation we have established for future work. The developed ATSS and sensorimotor algorithm can serve as a platform for more complex tasks and applications, such as

object manipulation and dexterous hand control.

This detailed information has been added into Line 575-635, Page 29-31 of 'Method' section in the revised manuscript to make a more detailed interpretation of this work. Our research represents a crucial step in the development of neuroprosthetics and robotics designed to enhance the lives of individuals with limb loss or limb impairment. While we have not yet conducted experiments with amputees, our study serves as a foundational platform upon which further research can be built. We fully acknowledge the importance of conducting experiments with amputees to assess the clinical applicability and real-world impact of our system. Moving forward, we plan to collaborate with clinical experts and institutions to conduct studies involving amputees. studies will focus on evaluating the potential benefits of our sensorimotor control strategy in enhancing the functionality and quality of life for individuals with limb loss.

In conclusion, we recognize the limitation of our study's participant pool and the absence of amputee testing. However, our research represents a critical first step in establishing the feasibility of our innovative sensorimotor control strategy. We are committed to expanding our research to include amputees, with the ultimate goal of improving the lives of those who can benefit from advanced neuroprosthetic technologies. Our focus is on building a foundation for future research and applications in this exciting and evolving field.

6. *The authors should explain how Izhikevich model parameters were selected.*

Response:

We appreciate the reviewer's interest in understanding the selection process for the Izhikevich model parameters in our study.

In our manuscript, we have chosen the Izhikevich model as the foundation for our neural dynamic simulations, a decision that is elaborated upon in our previous publication, "Predicting Afferent Neural Dynamics During Active Touch and Perception: A Multi-level Numerical Model". The choice of the Izhikevich model was made after careful consideration of its suitability for our specific research objectives. The Izhikevich model was selected due to its ability to capture the rich dynamics of neuronal behavior using a minimalistic set of equations and parameters. This model effectively balances biological plausibility and computational simplicity, making it an ideal choice for simulating complex neural dynamics without the computational burden associated with more detailed models.

When it comes to parameter selection for the Izhikevich model, we grounded our approach in both theoretical understanding and empirical data. The parameters were chosen to reflect the specific types of neurons we were modeling, particularly those involved in tactile sensation. These choices were based on a combination of the

following aspects:

1. Literature Review: Extensive review of existing neurophysiological studies provided insights into typical parameter values for neurons involved in tactile processing.
2. Empirical Data: Where available, empirical data from experiments studying tactile neural responses were used to inform our parameter choices. This ensured that our model was closely aligned with observed neural behaviors in tactile perception.
3. Simulation and Validation: The selected parameters were further refined through iterative simulation and validation processes. By comparing the model's output with known neural responses to tactile stimuli, we fine-tuned the parameters to achieve a high degree of accuracy in our predictions.

While the rationale for selecting the Izhikevich model has been extensively elucidated in our previously published work, our updated manuscript also underscores the specific motivations behind this choice (Line 527-537, Page 27). Including the Izhikevich model's ability to replicate various types of spiking and bursting patterns observed in biological neurons, with a relatively lower computational cost, made it particularly suitable for our multi-level numerical model that aims to predict afferent neural dynamics during active touch.

We believe that our methodological approach in selecting and tuning the Izhikevich model parameters ensures that our model provides a realistic and computationally efficient representation of neural dynamics in the context of tactile sensing. Thank you for allowing us to clarify this crucial aspect of our research. We are committed to providing transparent and detailed explanations of our methodologies to contribute effectively to the field of tactile sensing and neuro-engineering.

7. *They repeatedly mention “function. A subject-specific FE human hand model was applied to 133 simulate active touch. The strain energy density (SED)...”. But no details of the FEM model are given; indeed, this should be a very important supplementary material. What was the anatomical structure and detail? Was the model deformable? How are the tissue property (skin, muscle/fat, bone...even finer details) modeled and what data were used? What was the meshing details and algorithm?*

Response:

We appreciate the reviewer's interest in the finite element model (FEM) of the human hand used in our study. The FE human hand model we employed is extensively detailed in our previously published paper titled "Subject-Specific Finite Element Modelling of the Human Hand Complex: Muscle-Driven Simulations and Experimental Validation." This publication provides an in-depth description of the anatomical structure, material

properties, and simulation techniques used in the development of the model.

To address your question, we briefly summarized our previous work on the FE model as follows:

Anatomical Structure and Detail: The FE human hand model was meticulously developed to accurately represent the intricate anatomy of the human hand. It includes detailed geometries of phalanges, carpal bones, wrist bones, ligaments, tendons, subcutaneous tissue, and skin. These components were reconstructed based on CT and MRI scans, ensuring a high level of anatomical fidelity.

Deformability of the Model: Yes, the model is deformable. It was designed to simulate the complex biomechanical behavior of the hand during active touch. The model accounts for the heterogeneous, anisotropic, and viscoelastic properties of various tissues, particularly the skin and subcutaneous tissues. This allows for realistic simulation of the mechanical responses of the hand to external stimuli, including deformation under various loading conditions.

Recognizing the significance of these details for a comprehensive understanding of our study, despite their inclusion in our previous publication, we have provided a concise introduction and summary of the FE hand model in the supplementary material. This addition aims to offer a clearer presentation of how this previously developed model was utilized and its relevance to the current work:

Supplementary Material: Summary of the FE Human Hand Model

In our research, we have utilized a sophisticated and subject-specific Finite Element (FE) model of the human hand. This model is central to our study, enabling us to simulate the biomechanical aspects of active touch with high accuracy. Below is a summary of the model's structure, material properties, and the validation process:

Anatomical Structure:

The FE model incorporates a detailed representation of the human hand anatomy, including bones (phalanges, carpal, and wrist bones), ligaments, tendons, subcutaneous tissue, and skin. These components were reconstructed from CT and MRI scans of a specific subject, providing a highly realistic and anatomically accurate model.

Material Properties:

Skin: Modeled as a heterogeneous, anisotropic, and viscoelastic material, capturing the complex behavior of skin under mechanical stress.

Bones: Treated as isotropic linear elastic materials with specific Young's modulus and Poisson's ratio.

Ligaments and Tendons: Simulated using spring elements to represent their supportive role in joint movement.

Subcutaneous Tissues: Characterized with properties that allow for realistic simulation of soft tissue deformation.

Deformability and Biomechanics:

The model is designed to be deformable, simulating the biomechanical behavior of the hand during interactions, such as grasping or touching. This includes the accurate representation of joint movements, tissue deformation, and response to external forces.

Validation Process:

The model underwent rigorous validation against in-vivo experimental data. This included comparing predicted mechanical responses, such as contact pressure and area, with actual measurements from hand interactions.

Sensitivity Analysis: We conducted sensitivity analyses to understand the influence of various material properties and loading conditions on the model's predictions.

Implementation in Current Research: In our study, the FE hand model is used to simulate the mechanical responses of the hand during active touch scenarios. The model's outputs, such as strain and stress distributions, are crucial inputs for our multi-level numerical model that predicts afferent neural signals.

This FE model of the human hand, with its detailed anatomical structure and validated biomechanical properties, plays a pivotal role in our research. It allows us to bridge the gap between biomechanical interactions and neural processing, enhancing our understanding of tactile perception.

We thank the reviewer for highlighting the importance of these details, and we are committed to ensuring that they are made available as supplementary material to enhance the transparency and reproducibility of our research.

8. *Where to Eq. 5 and fit in Fig. 1? Is this the Layer 2 (PSP)? Where is your cuneate model?*

Response:

Thanks for the comments. Eq. 5 and its fit in Fig. 1 are indeed crucial components of our study, and we appreciate the opportunity to clarify their placement and relevance.

Equation 5 (PSP Computation): Equation 5 represents the computation of the Postsynaptic Potential (PSP) waveform for individual afferent tactile units (indexed as "i"). The equation describes the temporal dynamics of PSP, including decay and rise times. In figure 1, the visual representation of the PSP (Equation 5) was shown as the 2nd order tactile neuron.

Location in Figure 1: Equation 5 is applied at the level of the "Post-Synaptic Neural

Action Potentials" of the **2nd order tactile neuron shown in** Figure 1. Specifically, it is part of the process where biological 1st order afferent tactile signals are computed. These signals represent the neural responses at the first layer of neural processing in the model.

Cuneate Model (2nd order tactile neuron): The cuneate model, also referred to as the 2nd order tactile neuron as shown in Figure 1, is noted to clarify that the biological term 'Cuneate Model' corresponds to the '2nd order tactile neuron' of our multi-level numerical model in our revised manuscript.

To give a clear presentation of how our cuneate model (2nd order tactile neuron) works in the whole research process shown in Figure 1, the detailed explanation below has been added to Line 163-189, Page 8-9 of the revised manuscript:

Equation 5, which describes the postsynaptic potential (PSP) waveform, is relevant to the neural model and can be explained in the context of the study. It represents the dynamics of the PSP that is crucial in the information processing of afferent tactile signals in the neural model, where the mathematical models and algorithms used in the research are described. The PSP waveform described by Equation 5 plays a role in the neural processing of afferent tactile signals and contributes to the generation of neural dynamics in the cuneate neurons. Therefore, it is part of the neural model and its role can be emphasized when discussing the details of the model's architecture and functioning.

9. *Extraction of motor output is important; the author employ synergy. This is a well established concept, but it is not justified. i.e. what was the rationale of using synergy (for different grasps) vs individual fingers? Mainly you need to describe synergy in terms of grasps – or even avoid, since you are really not breaking up tracked finger/hand motions into synergies.*

Response:

We appreciate the reviewer's interest in the extraction of motor output and their query regarding the rationale behind employing synergy in our study. We have updated the manuscript to clarify why we employed synergy as the foundation for our sensorimotor control and the rationale behind using synergy as a modeling concept.

Rationale for Synergy:

The use of synergy in our study is based on the idea that the human motor system often operates by combining the motions of individual fingers into coordinated patterns or synergies when performing various grasping tasks. These synergies represent coordinated patterns of muscle activations that simplify the control of multi-fingered hands, allowing for efficient and robust manipulation of objects. Our goal was to capture the essence of this human motor control strategy.

Modeling Synergy:

While we acknowledge that the concept of synergy is well-established, it's important to clarify that in our study, we are not attempting to break up tracked finger or hand motions into specific pre-defined synergies. Instead, we use mathematical transduction functions to model the dynamic relationship between afferent tactile signals and motor neuron activation levels, which can be influenced by the concept of synergy. This allows us to simulate and control grasping tasks effectively.

Practical Application:

The use of synergy in our modeling approach allows us to create a simplified yet effective representation of the sensorimotor control strategy employed by humans during grasping. This simplification aids in the control of complex multi-fingered robotic hands and prosthetics. While individual finger control is certainly valuable and may be applicable in certain contexts, synergy-based control has demonstrated its practicality and efficiency in a wide range of real-world applications. Also, based on our experimental results and the existing literature, it's extremely challenging to replicate human-like dexterity through individual finger control strategies.

In the revised paper, we further elaborated on the rationale behind employing synergy as a modeling concept and provide a clearer description of how it relates to specific grasping tasks as is shown above (Line 198-215, Page 9-10). We hope this clarification helps address the reviewer's concerns.

10. On p. 8 you have the cuneate model, but its specifics, e.g. types of neurons, architecture (organization) etc is not explained. This is rushed, not clearly explained or justified.

Response:

We appreciate the reviewer's feedback regarding the cuneate model, and we apologize for any lack of clarity in our explanation. In response to the reviewer's valuable feedback, we would like to provide a more comprehensive explanation of the cuneate model utilized in our study. The cuneate model is a critical component of our neurocomputational framework, responsible for simulating the neural processing of tactile information at a subcortical level.

Types of Neurons: Our cuneate model incorporates two primary types of neurons to emulate the neural dynamics within the cuneate nucleus. These neuron types include excitatory neurons and inhibitory interneurons. Excitatory neurons serve as the principal conveyors of sensory information, transmitting tactile signals from the peripheral mechanoreceptors to subsequent neural stages. In contrast, inhibitory interneurons play a pivotal role in modulating and fine-tuning neural activity by

exerting inhibitory control over excitatory neurons. This dual-neuron representation aligns with the known neurobiology of the cuneate nucleus, contributing to a more biologically faithful simulation.

Architecture and Organization: The architecture of our cuneate model closely mirrors the anatomical organization of the cuneate nucleus observed in the human brainstem. The cuneate nucleus exhibits somatotopic organization, wherein different regions within the nucleus correspond to specific anatomical regions of the upper limb, particularly the hand and fingers. In our model, we faithfully replicate this somatotopic arrangement, allowing for the spatial mapping of tactile information. This organization ensures that tactile signals from distinct regions of the upper limb are processed separately within the cuneate nucleus before onward transmission.

Functional Role: The cuneate nucleus serves a pivotal role in the somatosensory pathway by receiving, processing, and encoding tactile information originating from peripheral mechanoreceptors, including those present in the hand. Its primary function is to relay this tactile information to higher-order brain centers, notably the somatosensory cortex. Within the cuneate nucleus, tactile signals undergo crucial processing and encoding, shaping the neural representation of sensory stimuli. These encoded signals are subsequently transmitted to the cortex, where they contribute to the perception and interpretation of tactile sensations.

We have provided these additional details above (added to Line 177-189, Page 8-9 in the revised manuscript) to offer a more thorough understanding of our cuneate model's characteristics, encompassing neuron types, architectural fidelity, and functional relevance. These enhancements serve to underscore the alignment of our computational framework with the known neurobiology of the cuneate nucleus and contribute to the overall biological fidelity of our study.

11. I found Fig. 5 rather unhelpful...too much spike data, not resolvable, and not very meaningful. E.g. difference between objects is not differentiable and not interesting. You can hardly tell the difference with the active touch (what/when and how is different other than some vague idea of different spike patterns). Instead, it would have been useful to model the skin layer and the cuneate layer and show these data, i.e. how the cuneate layer 'helps'.

Response:

Thanks for the comments and suggestion.

We appreciate the reviewer's feedback on the presentation of Figure 5 and acknowledge the concerns raised regarding the clarity and informativeness of the figure. We apologize for any confusion caused by the initial presentation of the raw data. In

response to the reviewer's suggestions, we made the following improvements in our revised manuscript:

Incorporating a Model: As suggested by the reviewer, we have included a diagram that illustrates the postsynaptic afferent (2nd order tactile neuron), providing clarity on how the cuneate layer contributes to the processing of 1st order cutaneous tactile information. Both 1st and 2nd order afferent neural signals have been presented, accompanied by anatomical images indicating where these signals are processed in the human subject (see Fig. R2 below). This comprehensive representation enhances the comprehensibility of the entire neuromorphic neuron model, making it more accessible and understandable within the context of this study.

Highlighting Key Findings: To make the figure more meaningful and relevant, we also emphasized the key findings related to how the cuneate layer contributes to tactile signal processing. Include highlighting specific data points or patterns that demonstrate the difference in neural responses between different objects and the role of the cuneate layer in shaping these responses.

Fig. R2. The neuromorphic tactile signals elicited based on the 6 by 6 tactile sensor array. The neuromorphic cutaneous (1st order tactile signals) and cuneate (2nd order tactile signals) signals (SAI units) elicited during active touch with cylinders and spheres ranging in diameter from 50 to 100 mm are presented. These tactile signals from two distinct levels are displayed alongside their corresponding positions within the human sensorimotor system and the artificial tactile sensory system.

12. Fig. 6 is also hardly helpful. The hat-top images conform to what you'd expect. Nothing new or interesting. In addition, why are not all receptors (SA/FA) included? what would really help would be give temporal profile of the receptors near/around the object? (the shape aspect is not very helpful)

Response:

We appreciate the reviewer's feedback on Figure 6 and understand the desire for more detailed and informative representations of tactile receptor responses. In response to the reviewer's suggestions, we made the following improvements to our revised manuscript.

Temporal diagrams were added:

We acknowledge that the current representation of hat-top images in Figure 6 may not provide significant new insights. In the revised manuscript, we added the temporal information which is the first spike latency of the artificial mechanoreceptors, these temporal clues will give a more comprehensive. This can provide a clearer understanding of the dynamics of tactile information processing and help convey more meaningful insights.

Inclusion of all receptors:

We understand the importance of including data from both SA (slowly adapting) and FA (fast adapting) receptors. In our revised manuscript, we included all the data from both receptor types, offering a more comprehensive view of tactile receptor responses.

By adding more information on these tactile signals shown in Figure 6 and 7 in the updated manuscript and Fig. S5 to 9 in the supplementary material, we aim to enhance the value and clarity, providing a more informative representation of tactile receptor responses and their temporal profiles.

13. Fig. 7 is confusing. You mention Victor-Pappura distance but don't give any data. Then, you show hit-rate. There is no statistical test on the differences among the 3 methods.

Response:

We appreciate the reviewer's feedback regarding Figure 7, and we acknowledge the need for clarity and additional information. In our revised manuscript, we have made the following improvements:

Data Presentation:

We included more detailed data to provide a clear comparison among the three methods (human subject, ATSS with rate coding, and ATSS with Victor–Purpura distance). The calculated and Victor–Purpura distances for the computation of the hit rates were added to Table S4 and 5 respectively. Furthermore, we have provided a more detailed explanation and analysis of the results presented in Figure 7, along with an explanation of how the Victor–Purpura distance was calculated (Line 299-320, Page 17 of the revised manuscript). In addition, we have included the MATLAB code for calculating the Victor–Purpura distance in the supplementary material to facilitate replication by other researchers.

Statistical Tests:

As suggested by the reviewer, we performed statistical tests to evaluate the differences in performances of two neurodynamic features. The difference between the hit rates

achieved by firing rates/Victor–Purpura distances and the human subject were quantified and analyzed (Line 301-313, Page 17), more discussions on these results and how these differences will affect the performance of our ATSS were added to the ‘Results’ section, Line 314-320, Page 17 of the revised manuscript. This will help provide a rigorous assessment of the effectiveness of our approach compared to human tactile sensing capabilities.

By implementing these improvements, we aimed to make Figure 7 more informative and transparent, allowing for a better understanding of the comparative performance of the different methods and the statistical significance of these differences.

14. It is not clear why the result, “The ATSS could differentiate the cylinder/sphere with a diameter of 90 mm from those of 100 mm with a hit rate above 68%,” is so exciting or compelling.

Response:

Thanks for the reviewer's feedback. In the revised manuscript, we have now provided further context regarding the significance of the result indicating that "The ATSS could differentiate the cylinder/sphere with a diameter of 90 mm from those of 100 mm with a hit rate above 68%."

The hit rate of above 68% was calculated based on the neural dynamic features of the neuromorphic signals including the firing rates and Victor–Purpura distances. The following more detailed description on how to calculate the hit rates or discrimination accuracy were added to the updated manuscript (Line 585-645, Page 29-30 of the revised manuscript) and the caption of Fig. S7 of the supplementary material:

As is shown in Fig. S7, the neural dynamics evoked during contact with the baseline object were recorded. The mean and standard deviation of the spiking rates during 10 touches of the baseline object were summarized as the 'noise' signal in signal detection theory (SDT). Meanwhile, the mean and standard deviation of the spiking rates during contact with other objects, used to differentiate them from the baseline object, were regarded as the 'signal' in SDT. Discrimination accuracy computed based on Victor-Purpura distance. The 'noise' signal in signal detection theory was defined as the Victor-Purpura distance among tactile signals during 10 touches of the baseline object (diameter 100mm). Victor-Purpura distances between neural dynamics evoked when touching the baseline and other objects with diameters less than 100mm were regarded as the 'signal' in SDT to evaluate discrimination accuracy. A total of 120 Victor-Purpura distances were computed. All cylinders were perceptually tested 10 times. The Victor-Purpura distances were calculated for differentiating the neuromorphic tactile signals evoked by contacting cylinders with diameters of 100mm and 90mm, 100mm and 80mm, 100mm and 70mm, 100mm and 60mm, 100mm and 50mm. The same method was used to compute discrimination accuracy for recognizing spheres with different

diameters. Therefore, there were a total of 50 Victor-Purpura distances for differentiating cylinders and another 50 for spheres from the baseline objects. These Victor-Purpura distances are given in Table S4 of the supplementary material.

While the result itself may not seem exceptionally compelling on its own, it is noteworthy in the context of neuroprosthetic and sensorimotor control. Here's why this result is significant:

Demonstrating Discrimination Capability:

The ability of the ATSS to differentiate between objects of similar sizes (e.g., 90 mm vs. 100 mm) with a reasonable hit rate demonstrates the system's capacity to discriminate fine tactile details. In the realm of neuroprosthetic, this capability is crucial for enabling users to interact with their environment effectively.

Realistic Tactile Discrimination:

Achieving a hit rate above 68% in distinguishing objects of these sizes aligns with human tactile discrimination abilities. This suggests that the ATSS is approaching the tactile discrimination performance of a human subject. This is significant because it brings neuroprosthetic and robotics closer to replicating human-like tactile sensing, which can greatly enhance the quality of life for individuals with limb loss.

Foundation for Further Development:

While this result may not be the ultimate endpoint, it serves as a foundational step in the development of tactile sensory systems for neuroprosthetic. Further refinement and optimization of the system, as well as potential integration with other sensory modalities, can lead to even more impressive results in the future.

In summary, while the specific result may not appear groundbreaking in isolation, it represents a significant step forward in the field of neuroprosthetic and sensorimotor control, particularly in terms of replicating human-like tactile discrimination abilities in artificial systems.

15. Fig. 8 is interesting and novel for this reviewer; how were the model parameters (Laplace coefficients) obtained? How were active/reactive parameters tuned? This is mysterious. Also, it was found that the transduction functions of $aS^2 + bS + c$ and $abS^3 + cS^2 + dS + e$ fairly represents.... How was this model determined? There is a high empiricism (or mystery) to this.

Response:

We appreciate the reviewer's interest in Fig. 8 and the questions regarding parameter estimation and tuning. The process of establishing these functions involved a

combination of empirical data collection and systematic modeling. The Laplace coefficients and active/reactive parameters are essential components of our model, and their determination and tuning involve several steps:

1. Laplace Coefficients (a, b, c, d, e):

Data-Driven Approach: The Laplace coefficients in our model were primarily determined through a data-driven approach. This involved collecting empirical data from experiments conducted on human subjects. The 2nd order neuromorphic tactile signals were computed and used as the inputs of our transduction functions while the neural activation level computed based on those EMG signals collected during *in-vivo* experiment from the human subject. These experiments provided us with a rich dataset of neural responses, tactile signals, and motor outputs during active and reactive grasping tasks.

System Identification: We employed system identification techniques to extract the Laplace coefficients that best represented the relationship between afferent tactile signals and efferent motor neuron activations. This process involved fitting the model to the collected data to optimize the coefficients for accurate representation of the system dynamics. (The MATLAB code for generating and optimizing those transduction functions for both active and reactive grasping were added to Supplementary information as Data. S2 and S3)

2. Active/Reactive Parameters Tuning:

Experimental Data: The parameters related to active and reactive grasping were optimized against the neural activation level extracted based on the EMG signals collected during the in-vivo experimental results. Tuning these parameters involved an iterative optimization process to ensure that the model accurately mimicked the sensorimotor control strategy observed in human subjects during grasping tasks. The model's performance was continually assessed by minimizing the Victor-Purpura distances between the neuromorphic and biological neural dynamics. This iterative feedback loop allowed us to fine-tune the parameters until the model closely replicated the observed behaviors.

The MATLAB code for generating and optimizing those transduction functions for both active and reactive grasping were added to Supplementary information as Data. S2 and S3. In summary, the determination of Laplace coefficients and the tuning of active/reactive parameters were data-driven processes that relied on experimental data collected from both the ATSS and human subjects. The goal was to ensure that our model accurately represented the sensorimotor control strategies observed in biological systems. The added MATLAB codes for generating and optimizing the transduction functions with detailed explanation in 'Method' ensured that the other researchers can replicate our work on restoring human-like performance on those robotic/soft hand similar to our ATSS in this study.

Specific Responses to Reviewer #4

1. *The introductory sentences provide great emphasis on Cuneate nucleus analogue, however the present manuscript does not provide enough methodological details on the biological Cuneate data/model used to support the contribution of the paper. I suggest integrating these aspects. As an example, the following sentence would require more supporting methods: "The ATSS developed in this study exhibited similar sensing performance with the human subjects at the 2nd order cuneate neural signals stage".*

Response:

We appreciate the reviewer's feedback and agree that providing additional methodological details regarding the biological cuneate data/model is essential to support the contributions of our study. To address this concern, we have made the following improvements to the manuscript:

Methodological Details on Cuneate Data/Model: We incorporated a dedicated section in the methods that provides a comprehensive description of the biological cuneate data and the modelling techniques used to integrate it into our research (Line 525-551, Page 27-28 of the revised manuscript).

Integration of Cuneate Model: We expanded upon the integration of the cuneate model into our artificial tactile sensory system (ATSS) and clarify how it relates to the neuromorphic tactile signals and sensorimotor control strategy presented in the study (Line 198-215, Page 9-10 of the revised manuscript).

Supporting Evidence: In the revised methods section, more supporting evidence for the statement, "The ATSS developed in this study exhibited similar sensing performance with the human subjects at the 2nd order cuneate neural signals stage" were added (Line 272-283, Page 15 of the revised manuscript).

These detailed information below regarding to methodological details on the biological Cuneate data/model were added to the 'Method' section as follows:

Types of Neurons: Our cuneate model incorporates two primary types of neurons to emulate the neural dynamics within the cuneate nucleus. These neuron types include excitatory neurons and inhibitory interneurons. Excitatory neurons serve as the principal conveyors of sensory information, transmitting tactile signals from the peripheral mechanoreceptors to subsequent neural stages. In contrast, inhibitory interneurons play a pivotal role in modulating and fine-tuning neural activity by exerting inhibitory control over excitatory neurons. This dual-neuron representation aligns with the known neurobiology of the cuneate nucleus, contributing to a more

biologically faithful simulation.

Architecture and Organization: The architecture of our cuneate model closely mirrors the anatomical organization of the cuneate nucleus observed in the human brainstem. The cuneate nucleus exhibits somatotopic organization, wherein different regions within the nucleus correspond to specific anatomical regions of the upper limb, particularly the hand and fingers. In our model, we faithfully replicate this somatotopic arrangement, allowing for the spatial mapping of tactile information. This organization ensures that tactile signals from distinct regions of the upper limb are processed separately within the cuneate nucleus before onward transmission.

Functional Role: The cuneate nucleus serves a pivotal role in the somatosensory pathway by receiving, processing, and encoding tactile information originating from peripheral mechanoreceptors, including those present in the hand. Its primary function is to relay this tactile information to higher-order brain centers, notably the somatosensory cortex. Within the cuneate nucleus, tactile signals undergo crucial processing and encoding, shaping the neural representation of sensory stimuli. These encoded signals are subsequently transmitted to the cortex, where they contribute to the perception and interpretation of tactile sensations.

Our goal is to enhance the clarity and transparency of the methods section, allowing readers to understand how the cuneate data and model contribute to the overall research framework. And make sure the other researchers can also replicate our work. By implementing these changes, we aim to provide a more comprehensive and well-supported account of the biological cuneate data/model and its relevance to our study. This will strengthen the scientific foundation of our work and address the reviewer's valuable feedback.

- 2. On the other side, in some cases the results section contains too much recall of the methods, that instead I would leave in the methods only. As an example, in the section on “Sensorimotor control strategy” (results) the following paragraph is too much methodological and may be simplified, leaving the details in the actual methods section: “Electromyography (EMG) signals of three muscles including the flexor digitorum profundus (FDP), flexor digitorum superficialis (FDS), and flexor pollicis longus (FPL) were recorded (see Fig. S1) using the Delsys Trigno (Delsys Inc., Boston, US). The kinematics of the hand were captured through Vicon Systems (Vicon Motion Systems Ltd., Oxford, UK). The neural activation level of the muscle synergy during active touch was extracted from the EMG signals using the non-negative matrix factorization algorithm (NMF) 43,44.”*

Response:

We appreciate the reviewer's constructive feedback regarding the level of detail in our

results section. We have taken the reviewer's suggestion to heart and have revised the text in the 'Sensorimotor control strategy' section to provide a more streamlined presentation of our methods. The details of data acquisition and processing, including the use of EMG signals, kinematic tracking, and the application of the non-negative matrix factorization algorithm (NMF), have been appropriately moved to the methods section for clarity. The revised text in results has been moved and simplified to 'Method' now:

After acquiring the population-level afferent tactile signals as the input for the sensorimotor control algorithm, we conducted in-vivo grasping experiments with a 24-year-old male subject. The experiments involved three hand postures: cylindrical and spherical grasping, as well as precision gripping, using objects with various diameters and shapes. During these experiments, we recorded electromyography (EMG) signals from three key muscles: the flexor digitorum profundus (FDP), flexor digitorum superficialis (FDS), and flexor pollicis longus (FPL). EMG signals were captured using the Delsys Trigno system (Delsys Inc., Boston, US), and hand kinematics were simultaneously tracked with the Vicon motion capture system (Vicon Motion Systems Ltd, Oxford, UK). To extract the neural activation level of the muscle synergy during active touch, we employed the non-negative matrix factorization algorithm (NMF). The determination of the optimal number of muscle synergies was based on achieving a mean variance accounted for (VAF) value exceeding 85%, with less than a 6% increment when adding additional synergies. Our analysis revealed that a single muscle synergy predominantly governed forearm muscle control during both active and reactive grasping in this study.

The muscle synergy neural activation level served as the output of our sensorimotor control strategy, while the input consisted of the 2nd order afferent tactile signals. To characterize the dynamic transduction mechanism connecting the input tactile signals and the output neural activation level, we employed a system identification algorithm. Notably, we considered the selective responses of the SAI and FAI tactile units crucial for sensorimotor control. To enhance the accuracy of our sensorimotor model, we computed and integrated neuromorphic tactile signals from both the SAI and FAI tactile units as the afferent tactile input. Across all grasping postures, the transduction function

$\frac{a}{s^2+bs+c}$ exhibited the best fit between the predicted and biological neural activation

levels. Therefore, we applied this transduction function to represent the dynamic relationship between afferent tactile input and the output activation level of muscle synergy during active grasping. Additionally, a reactive grasping experiment was conducted, where the subject grasped a cylinder or sphere and lifted a 20-gram weight to a specified altitude before dropping it onto the grasped object. In this case, the transduction mechanism between the input afferent neural dynamics, evoked by slippage, and the reactive neural activation level of muscle synergy, was extracted

through the system identification algorithm. The transduction function of $\frac{a}{s^3+bs^2+cs+d}$

was summarized to represent the sensorimotor strategy of reactive grasping. Further

details, including the values of the poles of these transduction functions, can be found in Table.S1-10, and a comparison between the biological and computed neural activation levels based on these transduction functions is presented in Fig. S2.

The revised section shown above now offers a concise overview of our experimental setup, emphasizing the critical aspects of our data acquisition and analysis without delving into methodological intricacies. This adjustment aims to enhance the readability of the results section, making it more focused on the outcomes and implications of our study. We believe that this revision successfully addresses the reviewer's concern and improves the overall flow of our manuscript. We thank the reviewer for their valuable input, which has contributed to the refinement of our work."

3. *Figure 5 should better correlate the experimental conditions (manipulated object and motor control strategy). Moreover, the authors should better explain whether the raster plots depict experimental repetitions over a single channel, or a single experiment with multi-channel data. A more structured figure with multiple panels would help better understanding the claimed results.*

Response:

We appreciate the reviewer's feedback regarding Figure 5 and the need for improved correlation between experimental conditions and clarification of data presentation. We agree that a more structured figure with multiple panels would enhance the understanding of the claimed results.

To address this concern, we have made the following improvements to Figure 5:

Correlation Between Experimental Conditions: We revised Figure 5 to clearly correlate the experimental conditions, including the corresponding manipulated objects and motor control strategies. This involved labelling and grouping the data points to provide a more intuitive visual representation of the experimental design (See Fig. R2 above).

Structured Figure with Multiple Panels: We have created a figure with multiple panels to present the data in a structured manner. A diagram that illustrates the postsynaptic afferent (2nd order tactile neuron) after the 1st afferent neural signals were added, providing clarity on how the cuneate layer contributes to the processing of 1st order cutaneous tactile information. Both 1st and 2nd order afferent neural signals have been presented, accompanied by anatomical images indicating where these signals are processed in the human subject (see Fig. R2 above). This comprehensive representation enhances the comprehensibility of the entire neuromorphic neuron model, making it more accessible and understandable within the context of this study.

Highlighting Key Findings: To make the figure more meaningful and relevant, we also

emphasized the key findings related to how the cuneate layer contributes to tactile signal processing. Include highlighting specific data points or patterns that demonstrate the difference in neural responses between different objects and the role of the cuneate layer in shaping these responses.

By implementing these changes, we aim to improve the clarity and comprehensibility of Figure 5, ensuring that readers can readily understand the relationships between experimental variables and the corresponding neural data. We thank the reviewer for this valuable suggestion, and we are committed to enhancing the quality of our manuscript.

4. *Figure 7 is about an interesting behavioral comparison between machine performance and human performance. Instead, the previous figures focus on the signal outputs from the spiking artifact. I see a missing argument in the proposed machine-human similarity: human electrophysiological recordings at peripheral nervous system level could be an added value in the main figures of the paper.*

Response:

We appreciate the reviewer's insightful suggestion regarding Figure 7 and the inclusion of human electrophysiological recordings at the peripheral nervous system level. We agree that incorporating human electrophysiological data, such as microneurography results, could provide valuable context and strengthen the comparison between machine and human performance in our study.

The human electrophysiological recordings of the peripheral nervous system, including the microneurography test, were extensively described and presented in our previously published work (IEEE Transaction on Biomedical Engineering, DOI: 10.1109/TBME.2022.3177006), with the citation provided in this manuscript. However, in response to the valuable feedback, we have now integrated human microneurography results and the experimental diagrams into the supplementary material (Fig. S5 or Fig. R3 below). These results offer a direct insight into human peripheral nervous system activity and the microneurography test for collecting tactile neural signals.

We believe that incorporating these results into the current manuscript will provide a more comprehensive and cohesive narrative. This addition will enable readers to better understand the relationship between our neuromorphic model and actual human peripheral nervous system activity, emphasizing the relevance and validity of our approach.

5. *Figure 9 shows an interesting comparison between 2nd order neuron firing in the developed artifact and biological recordings. However, I would suggest clarifying the meaning of “biological” (in terms of species, recording techniques, ...) in the figure and related caption. The same comment applies to Figure 10. Such comments are in line with the general comment on Cuneate data, reported above.*

Response:

Thank you for your valuable feedback regarding Figures 9 and 10. We acknowledge the importance of providing clear and specific information about the biological data used in these figures. To address this, we have provided detailed clarification regarding the term "biological" and specify the relevant species and recording techniques.

In Figures 9 and 10, when we refer to "biological" data, we are specifically referring to data obtained from *in-vivo* recordings in human subjects. The neural activation level of the human subject was derived based on the EMG signals recorded by the

To ensure clarity, we have revised the figure captions and added the note to explicitly state that the "biological" data pertains to recordings from human subjects using microneurography. This clarification will help readers better understand the source and nature of the biological data presented in the figures.

We appreciate your attention to detail, and we will make the necessary revisions to enhance the clarity and specificity of our manuscript.

6. *Minor edits:*

In Fig. 10, please correct KPa to kPa, as the capital “K” stands for Kelvin, not for kilo. In Fig. S3, please correct “seneory” in “sensory”

Response:

Thanks for the comments.

The KPa has been corrected to kPa. Also, the ‘seneory’ has been corrected. We are sorry for those typos.

REVIEWER COMMENTS

Reviewer #3 (Remarks to the Author):

Manuscript # NCOMMS-23-34899A

Title Human tactile sensing and sensorimotor mechanism: from afferent tactile signals to efferent motor control

Overall:

Strengths - This work address a need to translate neuroscience models for prosthesis. The key issues that warrant publications are the biomechanical transfer function model, synergy-based motor function, and the multilayer neuromorphic sensing model. As the previous review stated, closing the loop is important aspect of the paper. The authors have now presented 1+5 subject data comparing the human vs model (although they do not completely show the capabilities on these subjects).

Weaknesses or issues deserving further clarification.

- The functions of the hand i.e. the tasks presented grasp of round/cylindrical objects and pickup are underwhelming...slow and not really challenging to the sensory input and feedback loop of the biomechanical system.
- As state by the authors, "The foundation of our model is the subject-specific FE human hand model." How applicable is this specific model created from one individual to other subjects, especially amputees?
 - o While this reduces the confounding variables in this study, the authors should discuss how much the model would need to be changed for a new subject. Especially if the model needs to be changed from the ground-up.
 - o How would the subject specific model be created for amputee subjects, especially if they are bilateral amputees?
- In the discussion line 448, the authors mention that this work establishes foundational knowledge of neural processing. However, it seems more like this work is an application to a previous model that has been adapted. Indeed, this model has already been published!
- The preliminary data from the other 5 subjects don't provide enough insight into the performance of the study as a whole. The authors mention that individualization is necessary, but then each individual should participate in the closed loop or human vs algorithm.

Specific Questions tied to the Figures.

- What are the takeaways from figures 3 and 4? Is there any insight that can be gained by looking at the firing rates and spike latency from the images for the different objects? While these distribution results can be helpful, a subset of these would likely be enough in the main text, with the rest in the supplementary.
- Caption for Fig. 5 is quite inadequate. (Previous critique also pointed out Fig 5 limitation). The discrimination accuracy (hit rate) of the artificial tactile system compared with human subject. Label each a, b, c, d and provide their key message. Is the Figure 5 results only for the one subject? This is confusing, since you mention additional 5 subjects.
- Fig. 6 - Unclear how the functions represent the dynamic relationship. Need more details.
- Fig. 7 - Need more details in the caption interpreting the results. The captioning could be improved. How do you explain the human/biological vs ATSS differences? You have done the statistics. Statements on the differences would be helpful.

Items needing clarification in your writing and presentation

Clarify what you mean by “converges into concise transduction functions.”

Neural dynamics term should be clearly defined and consistently used (it can mean different things in different contexts).

A minor issue but you may cite the first author et al Raspopovic's work

Alternatively, Raspopovic and team...

The same with Ortiz-Catalan and team

The same for lines 453-455

Please explain “Active and reactive grasping” (differentiate these earlier on, defining clearly).

Additional (supplementary) information

Fig. 8 is interesting and novel for this reviewer; how were the model parameters (Laplace coefficients) obtained? How were active/reactive parameters tuned? This is mysterious. Also, it was found that the

transduction functions of aS^2+bS+c and abS^3+cS^2+dS+e _fairly represents.... How was this model determined? There is a high empiricism (or mystery) to this.

☒ A very detailed response is given in the rebuttal. But it is important that this modeling information is available in the supplementary information.

The reviewer (and readers will too) appreciates including the Cuneate model information.

Reviewer #4 (Remarks to the Author):

The authors applied several edits to the manuscript, enriching the methodological details provided, for the sake of reproducibility of the study.

Nevertheless, I still have some questions that I kindly ask to address:

- Please give information about the recording site of the reported microneurographic data of 1st order cutaneous receptors. I guess they are acquired from the median nerve, but please better explain the methods, as an example whether the recording site is at the level of the elbow or of the wrist (as both sites are reported in microneurographic literature).

- I still do not understand how 2nd order recordings at Cuneate level have been performed: the authors now claim that the plots shown in Figure 7 (plot in the 3rd row from top) are human recordings, nevertheless to the best of my knowledge this would be the first time in the electrophysiological literature that Cuneate data are shown on human subjects, while published data so far are limited to animal subjects. Please clarify, as this would be a major breakthrough that should be properly reported and detailed in the manuscript.

Response to Reviewers' Comments

Paper title: Human tactile sensing and sensorimotor mechanism: from afferent tactile signals to efferent motor control

Submitted to: Nature Communications

Manuscript number: NCOMMS-23-34899A

General Response

We thank the reviewers for their constructive suggestions which have greatly helped us to improve the manuscript. We respond to the individual points in detail below, indicating the corresponding changes that we have made to the manuscript.

Responses to Reviewer #3

1. The functions of the hand i.e. the tasks presented grasp of round/cylindrical objects and pickup are underwhelming...slow and not really challenging to the sensory input and feedback loop of the biomechanical system.

Response:

Thank you for your feedback regarding the tasks chosen for assessing the functions of the hand in our study. We appreciate the opportunity to clarify the rationale behind our choice of grasping tasks.

Justification of Task Selection: The three grasping postures performed in our study—grasping round/cylindrical objects and pickup tasks—are indeed widely used within the fields of robotics and prosthetics for assessing hand functions. These tasks were selected based on their established relevance and common application in both academic research and practical assessments of prosthetic performance.

Literature Basis: Our selection of tasks is grounded in established protocols widely recognized in the fields of robotics and prosthetics. For instance, tasks such as grasping cylindrical objects simulate common daily activities and provide a measurable way to evaluate the functional capabilities of prosthetic hands. The relevance of these tasks is supported by literature, including comprehensive studies like the one detailed in "The GRASP Taxonomy (added as reference [65])", which classifies and describes various grasping types pertinent to both human and robotic hand studies. In our revised manuscript, we have provided a more detailed explanation and appropriate citations to

justify the use of these specific grasping patterns (Line 630-638, Page 29). This will help in demonstrating their relevance and widespread acceptance in both academic research and clinical applications.

Adaption for Testing Sensory and Feedback Systems: While the tasks may appear straightforward, they are particularly effective for testing the sensory input and feedback loops of biomechanical systems. These tasks require precise control and feedback to successfully manipulate objects without dropping or crushing them, thus providing a robust platform for evaluating the effectiveness of sensory feedback mechanisms in prosthetic designs.

Achievements in Biomimetic Performance:

Similar Afferent and Efferent Neural Activation: By implementing these tasks, we successfully demonstrated that our artificial system could achieve neural activation patterns similar to those of a human, both in terms of afferent tactile signals and efferent motor responses. This similarity is crucial for restoring human-like tactile sensation and motor control functions in neuroprosthetic hands.

Contact Pressure: Additionally, the contact pressures observed on our artificial tactile sensory system closely matched those experienced by human subjects, further validating the effectiveness of our biomimetic approach. These findings are detailed in Figures 7 and 8, showcasing the robustness of our sensorimotor performance.

The highlights mentioned above have been added to the updated manuscript (Line 329-378, Page 16-19) to provide readers with a clearer explanation of our contributions to restoring human-like hand performance in biomimetic robot.

Importances of the sensorimotor controlling strategy: To highlight the importance of our chosen sensory input and feedback loop, we have included the results from additional experiments where our artificial tactile sensory system operates without the summarized sensorimotor controlling strategy and the neural model (Figure S11-21). This comparison underlines the critical role of our sophisticated control mechanisms in achieving life-like function. Detailed descriptions were added to the revised manuscript (Line 401-406, Page 20).

We acknowledge the reviewer's point on the potential for incorporating more complex tasks and will consider this for future studies to further challenge the system's capabilities.

2. (1)As state by the authors, "The foundation of our model is the subject-specific FE human hand model." How applicable is this specific model created from one individual to other subjects, especially amputees? (2)While this reduces the confounding variables in this study, the authors should discuss how much the model would need to be changed for a new subject. (3)Especially if the model needs to be changed from the ground-up. How would the subject specific model be created for amputee subjects, especially if they are bilateral amputees?

Response:

Thank you for the insightful questions regarding the adaptability of our subject-specific finite element (FE) human hand model to other subjects, particularly amputees. Your queries touch on critical aspects of our research aimed at enhancing the applicability of prosthetic technology. The follows are detailed explanations addressing each part of the question raised by the reviewer:

1. Applicability to Other Subjects, Including Amputees:

Biomechanical Model Adaptability: The subject-specific finite element (FE) human hand model forms the foundation of our study and is crucial for the precise replication of tactile and mechanical responses observed in human hands. This model's applicability to other subjects, particularly amputees, is an important aspect of our research. The model is based on detailed CT scanning data and microneurography, allowing for accurate replication of individual anatomical and neural characteristics. While the model was initially validated using data from a specific subject, the underlying methodology is designed to be adaptable to other subjects by adjusting the geometric and material properties to fit their unique anatomical data, which can be obtained in similar and simple ways.

Neural and Sensorimotor Strategy Generalization: Our findings indicate that neural dynamics models and sensorimotor controlling strategies exhibit significant similarities across different human subjects. This similarity extends to the microneurography signals measured from various subjects, which consistently show comparable patterns as supported by both our experimental data and existing literature.

2. Modifications for New Subjects:

Minimal Changes for Neural and Sensorimotor Components: Based on our experiments involving multiple subjects (detailed in Figure S1 of the supplementary material), the neural activation level extraction and the sensorimotor controlling strategy require minimal adjustments between individuals. This standardization is possible due to the fundamental similarities in neuromuscular structures and functions across humans.

Customization for Amputees: For amputees, particularly those who are bilateral, changes are primarily required in the anthropomorphic aspects of the model. This involves scaling the existing 3D models of the skeletal and subcutaneous structures to fit the individual's specific residual limb anatomy.

3. Customization for Bilateral Amputees:

Anthropomorphic Adjustments: For bilateral amputees, where no intact limb provides a direct comparative model, the customization involves linearly scaling the existing 3D models. These scaled models can then be used to fabricate biomechanical components like bone, tendons, and joint capsules through techniques such as 3D printing.

Use of Medical Imaging: In cases where an amputee has lost only one limb, CT or MRI scans of the remaining limb can provide a mirror-image model that can be adapted for creating a prosthetic counterpart. This approach ensures that the prosthetic is tailored to the individual's unique physiological and anatomical requirements.

The detailed descriptions provided above, concerning the applicability and versatility of our sensorimotor control strategy, as well as the comprehensive explanation of how to adapt the entire system for new subjects and potential customizations for bilateral amputees, have been added to the 'Method' section of the updated manuscript (Lines 657-666, Page 30). The related discussion on the applicability and potential improvements needed to transition our soft robotic system into a real commercial neuroprosthetic has also been included in the 'Discussion' section of the revised manuscript (Lines 479-485, Page 23). This enhancement strengthens the foundation of our research and contributes to the development of next-generation neural prosthetics.

3. In the discussion line 448, the authors mention that this work establishes foundational knowledge of neural processing. However, it seems more like this work is an application to a previous model that has been adapted. Indeed, this model has already been published!

Response:

We acknowledge the reviewer's observation regarding the use of a previously published model in our current study. Indeed, our work builds on existing models by integrating them with entirely new experimental data, including neural activation signals, and further computational enhancements. These are applied in a self-sensory and activation bio-robotic system to validate the models and restore human-like sensorimotor performance, advancing our understanding of tactile neural processing in neuroprosthetic applications. Previous models only predict or calculate afferent tactile signals; we use these predictions as inputs for our sensorimotor control strategy. We are sorry for any confusion this might have caused to our readers.

While it is accurate that the foundational model was introduced in prior publications, the current study significantly extends these earlier models by implementing them in new experimental setups and adapting them with updated neural dynamic algorithms that better mimic human sensory feedback mechanisms. This adaptation is critical as it validates the model under new conditions, offering robust evidence of its applicability and effectiveness in simulating human-like tactile responses in prosthetic devices. Our

contributions specifically involve significant enhancements to the sensorimotor transduction functions and their validation through rigorous *in-vivo* experiments and the applications on bio-robots. These efforts provide deeper insights into the neural mechanisms underlying tactile sensing and motor control, which are essential for the development of advanced prosthetic hands.

In our discussion, we aimed to highlight these contributions, noting how they build upon and extend the foundational work to achieve new understandings and applications. We have modified this sentence to clarify this point further in our updated manuscript to ensure the continuity and dependency on prior work is accurately represented and acknowledged.

The updated 'Discussion' section now reads: “Compared to other published works on closed-loop biomimetic hands or prosthetics, this study enhances the previously developed multi-level neural dynamic model by integrating it with new experimental data on neural activation signals and applying updated sensorimotor controlling algorithms in a bio-robotic system, restoring the human-like hand performance. This approach improves the understanding of neural processing in prosthetic functionalities.”

4. The preliminary data from the other 5 subjects don't provide enough insight into the performance of the study as a whole. The authors mention that individualization is necessary, but then each individual should participate in the closed loop or human vs algorithm.

Response:

We thank the reviewer for his/her insights regarding the preliminary data and the necessity of individualized testing protocols. The preliminary data presented from the five additional subjects were indeed aiming to illustrate the generalizability and adaptability of our model, rather than providing a comprehensive statistical analysis across varied individual conditions.

To address this valid concern, we carried out the further detailed studies involving each individual in a closed-loop setup and direct comparisons against the algorithm among different subjects. Individualization is central to our research, as each participant can present unique neuromechanical dynamics that might affect the performance of the prosthetic control strategies proposed, where each subject's data will directly contribute to refining and personalizing the control algorithms. This approach ensured that the model not only supports average predictions but is also robust enough to adapt to individual-specific variations, which is crucial for real-world applications.

The sensorimotor transduction functions for the other five subjects were also derived from additional experimental results. These functions are included in the supplementary

material (Tables S11-20). The gender and age information of the subjects is also included in the updated supplementary material (Table S21). Similar sensorimotor controlling strategies were observed across all six subjects. Further discussions on the similarities and slight differences in sensorimotor controlling strategies are included, this could significantly enhance the applicability of our summarized sensorimotor controlling strategies across diverse subjects. These relevant discussions have been added to the ‘Discussion’ section of the updated manuscript (Lines 401-406, Page 20).

This additional work not only deepens the findings but also substantiates the model’s adaptability and effectiveness across diverse subjects, thereby affirming its robustness and practical utility in real-world applications.

5. What are the takeaways from figures 3 and 4? Is there any insight that can be gained by looking at the firing rates and spike latency from the images for the different objects? While these distribution results can be helpful, a subset of these would likely be enough in the main text, with the rest in the supplementary.

Response:

Thank you for your comments and suggestions regarding Figures 3 and 4 and their presentation in the manuscript. These figures are crucial as they provide quantitative insights into the neuromorphic afferent tactile signals elicited under different grasping conditions, in order to help illustrate the model’s responsiveness to object features such as curvature and size.

Recommendation on Data Presentation:

As suggested, to streamline the main text, we have retained the distribution of the spiking rate and the first spike latency of the neuromorphic tactile signals observed when touching cylinders. The tactile information and diagrams related to spheres have been moved to the supplementary materials. This approach ensures that the main text remains focused on pivotal results, while still providing comprehensive data for in-depth examination by interested readers.

Takeaways from Figures 3 and 4:

Figure 3 illustrates the firing rates of neuromorphic tactile signals under cylindrical and spherical grasping conditions. It demonstrates that the tactile system exhibits higher firing rates for objects with smaller diameters, suggesting a sensitive response to curvature which may aid in fine object manipulation.

Figure 4 shows the distribution of first spike latencies across different tactile sensing elements when grasping cylindrical and spherical objects. Shorter latencies for smaller objects indicate a rapid initiation of neural responses, which is critical for timely and accurate sensorimotor control.

Insights from Firing Rates and Spike Latency:

The variations in firing rates and spike latencies across different object grasps provide insights into the tactile system's capability to differentiate object textures and shapes through temporal and rate coding strategies. This differentiation is essential for tasks requiring precise object manipulation and identification.

The distinct patterns observed in the firing rates and spike latencies help validate the effectiveness of the tactile sensor design and its integration with the neuromorphic model, showcasing the model's potential in replicating human-like tactile sensing in prosthetics.

We appreciate the suggestion to adjust the content distribution between the main manuscript and supplementary material and have implemented these changes to enhance the clarity and focus of our findings.

6. Caption for Fig. 5 is quite inadequate. (Previous critique also pointed out Fig 5 limitation). The discrimination accuracy (hit rate) of the artificial tactile system compared with human subject. Label each a, b, c, d and provide their key message. Is the Figure 5 results only for the one subject? This is confusing, since you mention additional 5 subjects.

Response:

Thank you for pointing out the need to enhance the caption for Figure 5 and clarify the subject data used. We acknowledge the confusion and have revised the caption to provide clearer information and label each panel accordingly:

Multiple Subjects: The results presented in **Figure 5 encompass data from multiple testing sessions involving all six subjects, not just one.** We have updated the experimental results of the in-vivo discrimination tests. The standard deviation of the hit rates for all six subjects has been added to Figure 5 in the revised manuscript. Additionally, a relevant description has been included in the caption of this figure (See the revised caption above or in the updated manuscript) to provide readers with a clear overview of our research and experimental outcomes. This approach was selected to demonstrate the robustness of the artificial tactile system across a diverse array of human sensory and motor responses. Each panel reflects a facet of these aggregated data, helping to illustrate both the general effectiveness and specific areas where the artificial system either aligns with or deviates from human performance.

Revised Figure 5 and its Caption:

Fig. R1: Comparative discrimination accuracy (hit rate) of the artificial tactile sensory system (ATSS) against multi-subject human performance benchmarks. (a) Hit rate based on cylinder grasping via rate coding displays the ATSS's capabilities for diameter identification through active touch, passive stimuli, and direct in vivo experiments, incorporating the standard deviation across six subjects to emphasize consistency of performance. (b) Hit rate based on cylinder grasping using the Victor-Purpura distance demonstrates the ATSS's precision in timing-based discrimination tasks, compared to human subjects over a variety of cylinder diameters. (c) Hit rate for spherical grasping by rate coding highlights the ATSS's proficiency in distinguishing spherical diameters, detailing active and passive interaction modes, supplemented by in vivo test data. (d) Hit rate for spherical grasping as determined by Victor-Purpura distance underscores the system's temporal resolution in identifying spheres, showcasing the system's effectiveness against the backdrop of human sensory and motor responses. The enhanced standard deviation representation for all panels underscores the robustness of the ATSS across diverse human experiences, illustrating a comprehensive view of the system's performance in alignment or deviation from the human subjects' tactile discrimination.

7. Fig. 6 - Unclear how the functions represent the dynamic relationship. Need more details.

Response:

Thank you for your comment requesting for further clarification on Figure 6. In response to your comment, we have made several key updates to both the figure and the accompanying descriptions to enhance understanding:

Revised Figure: We have updated Figure 6 to include clearer labelling of the mathematical functions, ensuring that each curve is distinctly marked to represent different model fits. This visual distinction helps in demonstrating how each function models the dynamic relationship between sensory input and motor output.

Revised Caption: The caption for Figure 6 has been revised to explicitly describe how these functions model the temporal changes and adaptive responses based on varying inputs. It now reads: "Figure 6: Dynamic Modelling of Sensory Input and Motor Output Relationships. This figure displays the mathematical functions derived from system identification techniques, quantifying the dynamic relationship between sensory inputs and motor outputs. Each curve represents a model fit, demonstrating how the system adapts to varying inputs and predicts motor responses, closely mimicking the adaptive responses observed in human neurophysiological processes."

Detailed Manuscript Explanation and Supplementary Codes: In the manuscript, we have expanded the section related to Figure 6 to include detailed information on how these functions were derived, the rationale behind selecting each function, and their significance in understanding neuroprosthetic control. We explained the mathematical basis of the functions and how they correlate with empirical data, providing a deeper insight into their application in modelling human-like neurodynamic. **Mathematical Basis:** We have uploaded the MATLAB codes illustrating how these transduction functions were derived using the 'System Identification' toolbox and have also provided sample input-output data in the supplementary material (Data S2). A detailed explanation describing the development and validation of these functions through both simulated and empirical data has been added to the 'Methods' section of the updated manuscript (Lines 564-586, Page 26-27).

Clarification on how these functions quantify changes over time in response to varying inputs, effectively modelling the adaptive responses observed in human neurophysiological processes. These updates were all include in the updated cation of Figure 6 presented above.

Revised Figure 6 and its Caption:

Fig. R2 Dynamic Modelling of Sensory Input and Motor Output Relationships. This figure displays the mathematical functions derived from system identification techniques, quantifying the dynamic relationship between sensory inputs and motor outputs. Each curve represents a model fit, demonstrating how the system adapts to varying inputs and predicts motor responses, closely mimicking the adaptive responses observed in human neurophysiological processes.

8. Fig. 7 - Need more details in the caption interpreting the results. The captioning could be improved. How do you explain the human/biological vs ATSS differences? You have done the statistics. Statements on the differences would be helpful.

Response:

Thank you for your comments and suggestions on Figure 7. We have now provided a more detailed caption to better interpret the results. We also have analysed the human/biological vs ATSS differences using statistical methods to ensure that our findings are robust and meaningful. Here are enhanced caption and further explanations in updated manuscript related to Figure 7:

Revised Caption for Figure 7:

Figure 7: Validation and performance comparison of the ATSS sensorimotor algorithm. (a) Active Grasping Phase: The left column illustrates the synchronous evolution of neural activation level of muscle synergy and contact pressure captured by the ATSS during an active grasping task. Biological and ATSS-generated 2nd order tactile afferent signals are juxtaposed, revealing the system's ability to emulate human tactile feedback patterns. (b) Reactive Grasping Phase: The right column presents the reactive phase where the ATSS adjusts to sudden contact, indicated by the dashed lines, with delayed neural activation and pressure adaptation. This response is compared with biological benchmarks, with the lower charts displaying the corresponding afferent tactile signals. These graphs collectively demonstrate the temporal accuracy and the neuromorphic efficacy of the ATSS, simulating humanlike sensorimotor functions as per the summarized transduction function. Statistical evaluations using Mann-Whitney U tests reveal significant differences in firing rates and Victor-Purpura distances ($p <$

0.03 and $p < 0.01$, respectively), demonstrating the enhanced capability of Victor-Purpura distances in discriminating tactile stimuli. This supports their potential utility in developing advanced tactile feedback mechanisms for prosthetic and robotic applications.

Interpreting Results and Explaining Differences:

The graphs in Figure 7 show that while the ATSS replicates human-like grasping performance to a large extent, there are noticeable differences in the neural activation levels and contact pressures between the ATSS and human subjects. These differences are quantitatively analysed through statistical tests, with the results indicating that:

Neural Activation: The ATSS tends to show delayed and sometimes heightened activation compared to human subjects, which could be attributed to the synthetic nature of neural signal processing within the ATSS.

Contact Pressure: ATSS generally exhibits higher contact pressures during initial contact but adjusts to stabilize similarly to human tactile feedback over time.

Statistical Analysis and Statements on Differences (Expanded Explanation in Manuscript):

In our analysis of the results shown in Figure 7, we conducted a detailed comparison of neural activation patterns between human subjects and the Artificial Tactile Sensory System (ATSS), employing Mann-Whitney U tests and Victor-Purpura distance metrics to quantify differences. These statistical methods revealed significant distinctions in how each system processes tactile information, highlighting areas for further enhancement in the ATSS.

The Mann-Whitney U tests, applied to neural firing rates and response latencies, showed significant differences with a p-value of less than 0.05. This indicates that the ATSS generally exhibits delayed response times and altered firing rate distributions compared to human subjects, suggesting that while the ATSS mimics human tactile responses, noticeable disparities in the speed and pattern of neural activation remain. Victor-Purpura distance metrics provided a nuanced analysis of the temporal dynamics of spiking activity. These metrics, which assess the cost of transforming one spike train into another, indicated greater distances between the ATSS and human subjects, with a p-value of less than 0.01. This result underscores that the timing and sequence of spikes in the ATSS are less synchronized with human neural patterns, emphasizing the system's current limitations in replicating exact human sensory processing.

These findings from both the Mann-Whitney U tests and Victor-Purpura distance metrics are crucial as they not only validate the existing capabilities of the ATSS but also highlight specific areas where improvements are needed. By pinpointing these discrepancies, future iterations of the ATSS can be fine-tuned to enhance its ability to closely replicate human neural dynamics, thereby improving its efficacy and realism for neuroprosthetic applications.

To address your feedback, we have included a detailed explanation of these differences above in the manuscript (Lines 340-350, Page 6), underpinning the statistical relevance and the potential implications for the design and development of more refined ATSS models. This discussion aims to provide readers with a clear understanding of where

the ATSS currently stands relative to human capabilities and the specific areas where improvements are targeted.

The adjustments in the caption and detailed results interpretation are intended to enhance understanding of the ATSS's capabilities and limitations compared to human tactile sensing. This will also address the previous critiques by clearly stating the differences observed and their potential impact on the system's performance in real-world applications.

9. Clarify what you mean by “converges into concise transduction functions.”

Response:

Thank you for asking for clarification on the phrase “converges into concise transduction functions.” This term refers to the process by which the complex interactions and data derived from our study are distilled into simplified but representative mathematical functions that accurately describe the dynamic relationships between sensory inputs and motor outputs in our biomechanical and neural models.

To address the reviewer's concern the phrase "converges into concise transduction functions" we have updated the abstract, introduction, or discussion section of the manuscript:

In the Introduction:

" The dynamic relationship between afferent tactile signals and motor neuron signals was encapsulated in 'transduction functions,' which offer the potential to restore sensorimotor performance in robotics or prosthetics. These 'concise transduction functions' represent a significant methodological advancement in this research, distilling complex sensorimotor interactions into simplified mathematical models. They capture the essential dynamics between afferent tactile signals and efferent motor responses, facilitating a deeper understanding of the underlying mechanisms crucial for neuroprosthetic development. " (Lines 84-94, Page 4).

This update aims to clearly integrate the explanation of how your study manages to refine and utilize transduction functions within the broader context of your research. They provide a clear link between the methodological innovations and their practical implications, ensuring that readers fully grasp the significance of these functions in advancing neuroprosthetic development.

10. *Neural dynamics term should be clearly defined and consistently used (it can mean different things in different contexts).*

Response:

Thank you for emphasizing the importance of clarity and consistency in our terminology. Following your feedback, we have carefully reviewed our manuscript and revised the term "neural dynamics" to ensure it is used precisely and appropriately. Here is a summary of the changes made to enhance clarity:

Revised Usage of Terms:

"Afferent tactile signals" – We replaced "neural dynamics" when it's referring to the cutaneous neural signals ‘

Location in Manuscript: Sections involving data analysis and results interpretation (Pages 2, 3, 4).

"Neural Firing Dynamics" - We replaced instances where "neural dynamics" referred specifically to changes in neural firing rates that are measured via electrophysiological techniques. This term now consistently describes the behaviour of firing rates over time, particularly in the context of processing sensory stimuli.

Location in Manuscript: Sections involving data analysis and results interpretation (Pages 12, 14, 19).

"Neural Response Dynamics" - Used when referring to the temporal aspects of neural responses to stimuli, such as how quickly neurons initiate, sustain, or change their activity in response to sensory input.

Location in Manuscript: Experimental methods section describing neural measurement techniques (Page 3, 22).

By defining and using these terms explicitly, we aim to accurately describe the processes studied and enhance the manuscript's readability and scientific precision.

11. *A minor issue but you may cite the first author et al Raspopovic's work Alternatively, Raspopovic and team...The same with Ortiz-Catalan and team,The same for lines 453-455*

Response:

Thank you for your attention to the details of our citation practices. We appreciate your suggestions for accurately acknowledging the contributions of Raspopovic and Ortiz-Catalan and their teams.

Revised Citation in updated manuscript:

Additionally, unlike other research developing tactile feedback systems, such as Raspopovic et al.'s practical sensory feedback applications in bidirectional hand prostheses ³⁶, Ortiz-Catalan et al.'s exploration of long-term bionic hand adaptation ³⁷, and Preatoni et al.'s ³⁸ investigation into multisensory perception, this research specifically focuses on tactile sensation and its direct translation into motor actions.

12. Please explain "Active and reactive grasping" (differentiate these earlier on, defining clearly).

Response:

Thank you for your feedback on the need to clearly define and differentiate "active and reactive grasping" early in our manuscript. We acknowledge the importance of these terms for understanding the experimental setup and results of our study.

The modifications in the manuscript:

Active Grasping:

Definition: Active grasping refers to the intentional manipulation of objects using precise motor commands that are initiated and controlled by the user. This type of grasping involves voluntary actions to reach, grip, and manipulate objects based on sensory input and cognitive planning.

Context of Use: In our experiments, active grasping is demonstrated when subjects or the ATSS initiate movements to interact with objects in a controlled environment, illustrating the system's ability to replicate human-like dexterity and intention-driven actions.

Reactive Grasping:

Definition: Reactive grasping occurs in response to unexpected changes or disturbances during the interaction with an object, such as adjusting grip due to slipping or sudden movement. This type of grasping is largely reflexive and involves rapid adjustments that are not pre-planned but are triggered by sensory feedback.

Context of Use: In the study, reactive grasping is assessed when subjects or the ATSS must modify their grip in response to unexpected stimuli, testing the system's reflexive capabilities and its ability to mimic human reactive adjustments.

These definitions have been added to the introductory sections of the Methods (Lines 630-633, Page 29) to ensure that readers are familiar with these concepts from the outset. By distinguishing between these types of grasping at the beginning, we aim to facilitate a better understanding of how each is evaluated and the relevance of each to the goals of our research in developing advanced prosthetic technologies.

13. Fig. 8 is interesting and novel for this reviewer; how were the model parameters (Laplace coefficients) obtained? How were active/reactive parameters tuned? This is mysterious. Also, it was found that the transduction functions of aS^2+bS+c and abS^3+cS^2+dS+e fairly represents.... How was this model determined? There is a high empiricism (or mystery) to this. A very detailed response is given in the rebuttal. But it is important that this modelling information is available in the supplementary information.

Response:

Thanks for the comments and asking the questions. We have now provided the details to clarify these questions (mystery) in the supplementary material.

Inclusion in Supplementary Information: We have included a detailed section in the supplementary materials that documents the full mathematical derivation of these functions, the algorithmic approach for parameter tuning, and the rationale behind the choice of each parameter and function form. Also, the MATLAB code for generating and optimizing transduction functions for both grasping types is included in the supplementary information (Data. S2 and S3). These aim to provide complete transparency and allow for reproducibility of our results. The added section in the updated Supplementary Information reads as follows:

Detailed Explanation for Modelling Techniques and Parameter Determination:

1. Obtaining Laplace Coefficients:

- Methodology: The Laplace coefficients were determined using a combination of system identification techniques and machine learning algorithms, utilizing extensive datasets from both simulated and experimental neural data.
- Process: These coefficients were optimized through a least square fitting procedure, iteratively adjusted to minimize the error between model predictions and observed data, enhancing model accuracy and stability.

2. Tuning Active/Reactive Parameters:

- Active Parameters: Tuned based on proactive interactions required in active grasping scenarios, using trial-and-error in controlled experimental settings to align closely with human performance metrics.
- Reactive Parameters: Adjusted for quicker responsiveness and higher sensitivity in reactive scenarios, involving dynamic simulations to handle sudden changes effectively.

3. Determination of Transduction Functions:

- Mathematical Formulation: We selected specific forms of transduction functions, aS^2+bS+c and abS^3+cS^2+dS+e , based on their historical success in similar biomechanical models and their ability to comprehensively represent the dynamics of sensory input conversion into motor outputs.

- **Empirical Validation:** These functions were validated against a subset of data not used in the training phase, ensuring they capture physiological processes accurately without overfitting.
- **Data-Driven Approach:** Empirical data collected from human subjects and the ATSS during active and reactive grasping tasks provided a rich dataset for system identification, used to extract Laplace coefficients that best represented the relationship between tactile signals and motor neuron activations.
- **System Identification and MATLAB Code:** The MATLAB code for generating and optimizing transduction functions for both grasping types is included in the supplementary information (Data. S2 and S3), enhancing the ability of other researchers to replicate and validate our findings.

Our approach to determining Laplace coefficients and tuning active/reactive parameters was meticulously data-driven, ensuring that our model not only accurately represents human sensorimotor control strategies but also aligns closely with biological dynamics. The supplementary MATLAB codes provide a foundation for other researchers to build upon our work, advancing the development of neuroprosthetic devices that closely mimic human tactile and motor functions.

14. The reviewer (and readers will too) appreciates including the Cuneate model information.

Response:

Thank you for acknowledging the inclusion of the Cuneate model in our research. We have added detailed information of the Cuneate model or 2nd tactile neuron to the modified manuscript (Lines 139-188, Page 6-9). The added information was also summarised as follows:

Detailed Description of the 2nd Order Tactile Neuron Model:

The 2nd order tactile neuron model is an integral component of our research, developed to simulate the neural dynamics of tactile information processing within the cuneate nucleus. This model utilizes a sophisticated approach to emulate the neural processing of afferent tactile signals from peripheral mechanoreceptors.

Model Foundations and Functionality:

- **Neural Dynamics Representation:** The afferent tactile signals are initially convolved with a postsynaptic potential (PSP) waveform, as described by Equation 5. This convolution is pivotal in simulating the neural activity that occurs following sensory stimulation. The PSP waveform is defined as:

$$PSP_i = \exp\left(\frac{t}{\tau_{decay}}\right) - \exp\left(\frac{t}{\tau_{rise}}\right)$$

The time decay $\tau_{decay} = 4\text{ms}$ and rise time $\tau_{rise} = 12.5\text{ms}$ determine the shape of the PSP_i kernel. The mapping from the afferent fibers to the cuneate neurons was defined based on neuroanatomical data.

- **Integration of Neural Dynamics:** The PSP signals from individual neurons are summed to form the total postsynaptic potential (PSP_{total}):

$$PSP_{total} = \sum_{i \in 36} PSP_i$$

This total postsynaptic potential represents the combined effect of neural activities and is crucial for the subsequent processing stages.

Cuneate Neuron Mapping and Simulation:

- **Neuroanatomical Data Mapping:** The mapping from the afferent fibers to the cuneate neurons is defined based on detailed neuroanatomical data, with an average divergence/convergence ratio of 1700/300. This ratio underlines the fast feed-forward encoding/decoding process characteristic of the 2nd neuron level.
- **Neural Encoding Process:** When 100 SAI (slowly adapting type I) mechanoreceptive units are activated, at least 567 cuneate neurons are recruited to process these signals, illustrating the model's capacity to mimic dense neural networks.
- **Dynamic Neural Computation:** The maximum PSP_{total} is selected and inputted into the Izhikevich model to compute the neural dynamics of the cuneate neurons using a winner-take-all algorithm. This method ensures that the most significant neural responses are highlighted, enhancing the model's realism and functionality.

Dual Neuron Types in the Model:

- **Excitatory and Inhibitory Neurons:** The model includes both excitatory neurons, which transmit sensory information, and inhibitory interneurons, which modulate and fine-tune the neural responses. This dual-neuron setup reflects the actual neurobiological processes within the cuneate nucleus, allowing for a biologically faithful simulation.

Somatotopic Organization:

- **Anatomical Accuracy:** The cuneate nucleus's somatotopic organization is meticulously replicated in our model, enabling precise spatial mapping of tactile information corresponding to different regions of the upper limb. This organization is critical for processing signals in a way that mirrors human sensory pathways, ensuring that tactile information is accurately relayed to higher brain centres for further interpretation and perception.

By employing this detailed and biologically inspired model above in the updated manuscript (Lines 340-350, Page 6), our study not only enhances understanding of the central processing of tactile information but also paves the way for developing more advanced neuroprosthetic systems that can effectively mimic human sensory processing and response mechanisms.

Responses to Reviewer #4

1. Please give information about the recording site of the reported microneurographic data of 1st order cutaneous receptors. I guess they are acquired from the median nerve, but please better explain the methods, as an example whether the recording site is at the level of the elbow or of the wrist (as both sites are reported in microneurographic literature).

Response:

Thank you for your inquiry regarding the recording site of the microneurographic data of 1st order cutaneous receptors used in our study. We understand the importance of detailing the methods used for collecting these critical data points.

The microneurographic recordings of 1st order cutaneous receptors were indeed acquired from the median nerve. Specifically, these recordings were taken at the position of the wrist. This site was chosen based on its accessibility and the reliability of signal acquisition, which are crucial for obtaining high-quality data representative of tactile sensory responses. We have thoroughly described the process of our microneurography tests in a previously published paper in the IEEE Transactions on Biomedical Engineering, which is referenced in our manuscript. This publication provides an extensive overview of our recording techniques and the experimental setup, ensuring that researchers and practitioners can replicate our methodology. To provide a clear and more straightforward description of our microneurography test, we have updated the manuscript to include a brief explanation of how we conducted and collected afferent signals from the median nerve of the wrist (Lines 117-120, Page 5).

Supplementary Material Inclusion:

To further aid in understanding and replicating our methods, we have included diagrams (Fig. S1 (a) and (b)) and additional details of how we conduct microneurography in the supplementary material of the manuscript (See Figure R3 here). These diagrams illustrate the positioning, the equipment used, and the specific techniques employed to capture neural signals from the median nerve at the wrist. We also updated the manuscript to mention that the detailed process of the microneurography and the relevant diagrams are given in supplementary information of this paper.

By providing this detailed response and ensuring that all relevant information is accessible in both the main manuscript and supplementary materials, we aim to clarify any uncertainties regarding our microneurography methods and the specifics of the recording site. This should enhance the reader's understanding and confidence in the validity and reliability of our data collection processes.

Fig. R3 The microneurography test and the experimental results. (a) Experimental setup for microneurography, the measurement was performed by inserting a tungsten electrode into the median nerve at the wrist. This technique captured single-afferent neural signals, which were essential for assessing the tactile sensitivity and neural response properties of the subjects. The tactile units' receptive fields were systematically stimulated using a Robotic Tactile Stimulator (RTS) to apply controlled sweeping motions across the fields at specified forces. (b) The stimulator is used to activate the cutaneous mechanoreceptor. (c) Example neural dynamics recorded from an SAI tactile unit. (d) Example neural dynamics recorded from a FAI tactile unit

2. I still do not understand how 2nd order recordings at Cuneate level have been performed: the authors now claim that the plots shown in Figure 7 (plot in the 3rd row from top) are human recordings, nevertheless to the best of my knowledge this would be the first time in the electrophysiological literature that Cuneate data are shown on human subjects, while published data so far are limited to animal subjects. Please clarify, as this would be a major breakthrough that should be properly reported and detailed in the manuscript.

Response:

Thank you for the opportunity to clarify the original contributions of our research depicted in Figure 7. We acknowledge your interest in the novelty of these findings. The data presented, particularly the plot in the third row from the top, are indeed a significant breakthrough of our study. To clarify, while these plots represent the activity patterns that would be expected in human Cuneate neurons, they are derived from a detailed computational model rather than direct electrophysiological recordings from human subjects. However, this is indeed the first time that cuneate data elicited by active touch has been derived and presented, representing one of the breakthroughs in our work.

Model-Based Simulation:

- **Simulation Basis:** The neural activity patterns shown are generated using a sophisticated computational model that simulates the responses of human Cuneate neurons. This model integrates extensive anatomical and physiological data derived from animal studies, which are currently the primary sources for such deep neural recordings, alongside extrapolations from human peripheral nerve recordings.
- **Data Integration:** The model incorporates both the typical neural response characteristics known from animal models and adjustments based on comparable human peripheral data to ensure that the simulated responses are as realistic as possible. The aim is to provide a predictive framework for what Cuneate nucleus activity might look like in humans, based on the best available data.

Justification for Approach:

- **Ethical and Technical Limitations:** Direct recordings from the human Cuneate nucleus present significant ethical and technical challenges. Due to these constraints, our approach leverages model-based simulations to explore these deep neural processes. This method allows us to hypothesize about the nature of human neural processing at this level without the direct use of invasive techniques.
- **Innovative Contribution:** Although direct human data would indeed represent a major breakthrough, our model-based approach also provides significant insights into potential human neural dynamics at the Cuneate level. These findings can help guide future research and potentially inform less invasive techniques for studying deep neural structures in humans.

Enhancements to Manuscript:

- **Detailed Reporting:** We have updated our manuscript to clearly state that these data are the result of computational simulations rather than direct recordings (Lines 162-188, Page 8-9). And we clarified this again in the caption of Figure 7. These clarifications are now included in both the figure caption and the methods section, ensuring that the basis for these data is transparent.
- **Highlights of Findings:** Additionally, we clarified that our multi-level numerical model was used to derive and present cuneate data elicited by active touch for the first time. Our simulated data significantly contribute to the field of neuroprosthetic, particularly in terms of designing devices that can effectively interface with human neural systems at various levels, including deep structures like the Cuneate nucleus (Lines 395-400, Page 19-20).

We hope this detailed explanation resolves any confusion regarding our methods and the nature of the data presented in Figure 7. We are committed to advancing our understanding of human neural dynamics through innovative and ethically responsible

methods, and we appreciate the opportunity to clarify the pioneering aspect of our research.

REVIEWERS' COMMENTS

Reviewer #3 (Remarks to the Author):

The paper has some methodological advances

30 integrating a finite

31 element hand model with a neural dynamic model, finely tuned using microneurography data,

And, some important novel conclusions:

human touch with

41 cutting-edge biomimetic control strategies reveals novel prospects in prosthetic design,

35 dynamic interplay between the afferent tactile signals and neural activation level of forearm muscles

36 converges into concise transduction functions

The ATSS developed in this study exhibited sensing performance comparable to humansubjects at the 2nd order cuneate neural signals stage, with differences in discrimination accuracy between

the human subjects and the ATSS below 15%.

The results are both good, and illuminative of the competitiveness of ATSS.

279 Fig. 4: Comparative discrimination accuracy (hit rate) of the artificial tactile sensory system (ATSS)

280 against multi-subject human performance benchmarks. (a) Hit

The following result really indicates the performance and utility of neuromorphic/spiking solution different sizes. This superior discriminatory power of Victor–Purpura distances, highlighted by their lower

312 p-value, demonstrates their effectiveness in mimicking human tactile sensing and suggests their potential

313 to enhance artificial tactile systems

Modeling and the results are the real strengths.

The ATSS performed active and reactive grasping under the control of the summarized transduction 330 functions.

Comparison between human and ATSS shows excellent results (Fig. 7).

Rebuttal

Thank you for responding to all the question in-depth, rigorously.

Query 1

- I still maintain that the 3 tasks are basic and while they may be the norm for the field, they do not do justice. You could, for example show a dynamic task, i.e. with sensing and loading there is an adaptation (eg. if you changed the load or the shape abruptly) so that a dynamical system would adjust the motor response based on changing sensory input.

Query 2: 1. Applicability to Other Subjects, Including Amputees

- This too is weak, i.e. no amputee studies were done, but at least you have data from several subjects.

Query 4: The preliminary data from the other 5 subjects don't provide enough insight ... each individual should participate in the closed loop or human vs algorithm. but then each individual should participate in the closed loop or human vs algorithm.

- This can be combined with Query 2 and show a dynamical adaptation.

Query 7: 7. Fig. 6 - Unclear how the functions represent the dynamic relationship. Need more details.

- Goes back to the comment above, and what's mentioned below (give the model parameters and the code for duplication or use of this important aspect of the paper).

Query 9: Clarify what you mean by "converges into concise transduction function..."

- See below; not adequately and clearly defined.

Query 12: Good to see that you have done a better job of defining active, reactive...

Query 14: The reviewer (and readers will too) appreciates including the Cuneate model information.

- The cuneate model is now very well described.

Minor Issues, suggest clarifications

You have the statement on Raspapovic/Otiz-Cartalan works, "However, the sensorimotor control strategy and neural coding mechanism have not been explicitly summarized and applied on robotics/prosthetics to restore human-like sensing and grasping performance, a crucial step for advancing prosthetic technology and rehabilitation."

- I don't think this is correct – you should be more specific for this rebuttal. E.g. model-based and closing the loop for grasps and objects.

Now you add "The dynamic relationship between afferent tactile signals and motor neuron signals was 85 encapsulated in 'transduction functions,"

- Please define transduction functions.

88 They capture the essential dynamics between afferent tactile signals and efferent motor responses,

- 89 facilitating a deeper understanding of the underlying mechanisms crucial for neuroprosthetic development

Thank you for providing

139 order tactile neuron model was developed based on 1st order neuron model (see Fig. 2).

The model details are helpful

162 The average divergence/convergence ratio of 1700/300 corresponding to the fast feed-forward encoding/decoding process of the 2nd neuron level 45

163 was adopted. Therefore, if 100 SAI units are activated,

164 there would be at least 567 cuneate neurons recruited to post-process these afferent tactile signals.

Some of this is introduction and could go into the Introductory part of the paper, not in results

181 cuneate nucleus before onward transmission. The cuneate nucleus serves a pivotal role in the somatosensory

182 pathway by receiving, processing, and encoding tactile information originating from peripheral

183 mechanoreceptors, including those present in the hand. Its primary function is to relay this tactile

184 information to higher-order brain centers, notably the somatosensory cortex. Within the cuneate nucleus,

221 the transduction function in terms of aS^2+bS+c

220 achieved the largest goodness of fit between the predicted and biological neural activation levels among the functions adopting the different numbers of pole/zeros below the 10th 221 order. Therefore, the transduction

- Where are the model parameters? Are they unique to each subject?

The same for

228 synergy were extracted through the system identification algorithm. The transduction function of

a

S^3+bS^2+cS+d

229 was summarized to represent the sensorimotor strategy of reactive grasping.

303 hit rates based on passive stimuli were lower than those under active touch. The MATLAB code for

304 calculating the Victor–Purpura distance is provided in Data S1 of the supplementary material.

- This is not as significant as the transfer function, dynamic model. Those parameters and the code will be useful to the readers.

These models are important and core of the paper: therefore, the model parameters and the code should be made available to the readers.

found that the transduction functions of aS^2+bS+c and aS^3+cS^2+dS+e

399 could fairly represent the dynamic

400 relationship between the afferent post-synaptic signals and the efferent neural activation level of muscle

401 synergy under active and reactive grasping, respectively.

The following statement in conclusion captures the contribution and could go in Abstract (or emphasize the sensation to motor actions where appropriate):

469 , this research focuses specifically on tactile sensation and its direct translation into motor actions.

Reviewer #4 (Remarks to the Author):

The revised paper addressed all comments proposed by the reviewer in previous interactions. I would recommend a professional editing of the figures, so to comply with the journal standards.

Best regards,

Calogero Maria Oddo

<https://www.santannapisa.it/en/calogero-maria-oddo>

Response to Reviewers' Comments

Paper title: Human tactile sensing and sensorimotor mechanism from afferent tactile signals to efferent motor control

Submitted to: Nature Communications

Manuscript number: NCOMMS-23-34899B

General Response

We thank the reviewers for their constructive suggestions which have greatly helped us to improve the manuscript. We respond to the individual points in detail below, indicating the corresponding changes that we have made to the manuscript.

Responses to Reviewer #3

1. I still maintain that the 3 tasks are basic and while they may be the norm for the field, they do not do justice. You could, for example show a dynamic task, i.e. with sensing and loading there is an adaptation (eg. if you changed the load or the shape abruptly) so that a dynamical system would adjust the motor response based on changing sensory input.

Response:

We appreciate the reviewer's comment and suggestion to include a dynamic task that demonstrates adaptation in response to changing sensory input. To address this concern, we have added an additional experiment involving reactive spherical grasping with varied pulling force on the grasped ball. It demonstrated that under the resultant reactive transduction function control, the grasping stability was maintained despite the variations in pulling force.

Additionally, we also demonstrated the grasping task without the control of our sensorimotor controlling strategy. In this scenario, human-like grasping stability could not be achieved, and the grasped object could easily be pulled out. This comparison highlights the effectiveness of our sensorimotor control strategy in maintaining stable grasping under dynamic conditions.

The video demonstrating these results has been added to the supplementary movie S2. These findings have been included in the revised manuscript (Line 341-344, Page 17). We believe this addition significantly strengthens our study by showcasing the adaptability and stability of the system in dynamic conditions.

2. Query 4: The preliminary data from the other 5 subjects don't provide enough insight ... each individual should participate in the closed loop or human vs algorithm. This can be combined with Query 2 and show a dynamical adaptation.

Response:

Thank you for the feedback regarding the preliminary data and the suggestion to have each individual participate in both the closed loop and human vs algorithm tasks, demonstrating dynamical adaptation.

To address this, we have expanded our study to include detailed experiments where each of the additional 5 subjects participated in the closed loop control performance test which is the human vs algorithm tasks. In these experiments, each subject's performance was analyzed to assess the effectiveness of the sensorimotor control strategy in dynamically adapting to different human subjects. The contact pressure of both hands of the 5 human subjects and the contact pressure of the biomimetic hand under the sensorimotor controlling strategies summarized based on their individual in-vivo measurements (efferent signals shown in Fig.S2) have been added to Fig.S3 in the supplementary material.

The results indicate that in the closed-loop condition, the performance of the biomimetic hand under the sensorimotor controlling strategies extracted from all the subjects showed significant improvements in grasping stability compared to the cases without sensorimotor control. These findings demonstrate the superior performance of the closed-loop control system in achieving human-like dynamical adaptation to different human subjects.

We have integrated these results with the adaptation experiments discussed in Query 2. The comprehensive data and analysis from these experiments are now included in the revised manuscript (Line 337-340, Page 17), with detailed results presented in Figures. S10-11. This addition provides a deeper insight into the system's ability to adapt dynamically under varying conditions.

3. Query 7: Fig. 6 - Unclear how the functions represent the dynamic relationship. Need more details. Goes back to the comment above, and what's mentioned below (give the model parameters and the code for duplication or use of this important aspect of the paper).

Response:

Thanks for your feedback regarding Figure 6 (now Figure 5 in the revised manuscript) and the need for more details on how the functions represent the dynamic relationship. To address this, we have added a detailed explanations in the revised manuscript that clarify the dynamic relationship depicted in Figure 6. Specifically, we have included a comprehensive description of the model parameters and the underlying functions used to represent the dynamic interactions in the caption of Figure 6.

Updated caption of Figure 6 (now Figure 5 of the updated manuscript) with detailed explanations has been given as: The transduction functions in this figure represent the dynamic relationship between afferent tactile signals and efferent motor responses, quantifying how sensory input (S) influences motor output over time and modelling the adaptive responses observed in human neurophysiological processes. The functions were derived using the ‘System Identification’ toolbox in MATLAB®, based on input-output data from our experiments. For active grasping, the function is $\frac{a}{S^2+bS+c}$, and for reactive grasping, the function is $\frac{a}{bS^3+cS^2+dS+e}$, where $a, b, c, d,$ and e are parameters optimized to fit the observed data. These functions capture the nonlinear and complex dynamics between sensory inputs and motor outputs. The development and validation process, detailed in the ‘Methods’ section, involved comparing predictions with empirical data from simulated and in-vivo experiments. The MATLAB code and corresponding data are provided in the supplementary material (Data S2), demonstrating the accuracy and applicability of the functions in modelling human-like sensorimotor performance.

Detailed Manuscript Explanation and Supplementary Codes: We have provided a comprehensive explanation of the mathematical basis of the functions and their correlation with empirical data, offering deeper insights into their application in modelling human-like neurodynamic. The MATLAB codes illustrating the derivation of these transduction functions using the ‘System Identification’ toolbox in MATLAB®, along with sample input-output data, have been uploaded in the supplementary material (Data S2). Additionally, a detailed explanation of the development and validation of these functions through both simulated and empirical data has been added to the ‘Methods’ section of the updated manuscript (Lines 581-591, Page 28). This explanation clarifies how these functions quantify changes over time in response to varying inputs, effectively modelling the adaptive responses observed in human neurophysiological processes.

We believe that these additions will significantly enhance the clarity and reproducibility of our findings.

*4. Query 9: Clarify what you mean by “converges into concise transduction function...”
- See below; not adequately and clearly defined.*

Response:

We appreciate the reviewer's request for clarification on the term “converges into concise transduction function”. In the updated manuscript, we have provided a detailed explanation (Line 81-86, Page 4) as follows:

The phrase "converges into concise transduction functions" refers to the process by which the complex dynamic relationship between afferent tactile signals and the neural activation levels of forearm muscles is distilled into simplified mathematical models. For active and reactive grasping, the relationship is captured by transduction functions. These functions effectively encapsulate the essential dynamics of sensorimotor interactions, allowing for an efficient representation of how sensory inputs translate into motor outputs, thereby facilitating the development of human-like sensorimotor performance in robotics and prosthetics.

We suppose that the detailed explanation can provide a clear and concise definition of these transduction functions and their role in modelling human-like neurodynamic.

Responses to Reviewer #3 Minor Questions

1. You have the statement on Raspapovic/Otiz-Catalan works, “However, the sensorimotor control strategy and neural coding mechanism have not been explicitly summarized and applied on robotics/prosthetics to restore human-like sensing and grasping performance, a crucial step for advancing prosthetic technology and rehabilitation.”

- I don't think this is correct – you should be more specific for this rebuttal. E.g. model-based and closing the loop for grasps and objects.

Response:

Thanks for the comment. Regarding the statement on Raspapovic and Ortiz-Catalan's works, we have revised the manuscript to provide a more specific comparison. We acknowledge the significant advancements made by their research in bidirectional hand prostheses and long-term bionic hand adaptation. However, in comparison, our study specifically focuses on model-based sensorimotor control strategies and neural coding

mechanisms. These strategies explicitly summarize and apply closed-loop control for grasping different objects, thereby restoring human-like sensing and grasping performance. This detailed approach is a crucial step for advancing prosthetic technology and rehabilitation by providing more precise and adaptive control mechanisms. The revised text has been updated in the manuscript to accurately reflect these distinctions.

The revised text reads: While Raspapovic and Ortiz-Catalan's works have advanced bidirectional prostheses and long-term bionic hand adaptation, our study focuses on model-based sensorimotor control strategies and neural coding mechanisms. Specifically, we apply these strategies to closed-loop control for grasping different objects, restoring human-like sensing and grasping performance. This approach is crucial for advancing prosthetic technology and rehabilitation with more precise and adaptive control mechanisms

2. Now you add "The dynamic relationship between afferent tactile signals and motor neuron signals was encapsulated in 'transduction functions,'" - Please define transduction functions.

Response:

We appreciate the reviewer's request for a definition of transduction functions. In the updated manuscript, we have provided a clear and concise definition to enhance understanding.

Transduction functions are mathematical models that encapsulate the dynamic relationship between afferent tactile signals (sensory inputs) and motor neuron signals (motor outputs). These functions translate complex sensory input data into corresponding motor responses, enabling accurate representation and prediction of sensorimotor system behavior.

This definition has been included in the updated manuscript (Lines 324-327, Page 16) to ensure clarity.

3. Some of this is introduction and could go into the Introductory part of the paper, not in results: 'cuneate nucleus before onward transmission. The cuneate nucleus serves a pivotal role in the somatosensory pathway by receiving, processing, and encoding tactile information originating from peripheral mechanoreceptors, including those present in the hand. Its primary function is to relay this tactile, information to higher-

order brain centers, notably the somatosensory cortex. Within the cuneate nucleus, the transduction function in terms of aS^2+bS+c achieved the largest goodness of fit between the predicted and biological neural activation levels among the functions adopting the different numbers of pole/zeros below the 10th order.'

Response:

We appreciate the reviewer's suggestion to relocate some of the introductory information currently placed in the results section. In the updated manuscript, we have moved the following content to the introductory part:

“The cuneate nucleus serves a pivotal role in the somatosensory pathway by receiving, processing, and encoding tactile information originating from peripheral mechanoreceptors, including those present in the hand. Its primary function is to relay this tactile information to higher-order brain centers, notably the somatosensory cortex.” This relocation ensures that the background information appropriately contextualizes the results.

Furthermore, we have clarified that within the cuneate nucleus, the transduction function $\frac{a}{s^2+bs+c}$ achieved the largest goodness of fit between the predicted and biological neural activation levels among the functions adopting different numbers of poles/zeros below the 10th order. This detail remains in the results section to highlight its significance in our findings.

4. Where are the model parameters? Are they unique to each subject? The same for: ‘synergy were extracted through the system identification algorithm. The transduction function of S^3+bS^2+cS+d was summarized to represent the sensorimotor strategy of reactive grasping. hit rates based on passive stimuli were lower than those under active touch. The MATLAB code for calculating the Victor–Purpura distance is provided in Data S1 of the supplementary material.’ This is not as significant as the transfer function, dynamic model. Those parameters and the code will be useful to the readers. These models are important and core of the paper: therefore, the model parameters and the code should be made available to the readers. ‘found that the transduction functions of aS^2+bS+c and aS^3+bS^2+cS+d could fairly represent the dynamic relationship between the afferent post-synaptic signals and the efferent neural activation level of muscle synergy under active and reactive grasping, respectively.’

Response:

We appreciate the reviewer's emphasis on the importance of making model parameters and code available to readers. In response, we have provided the necessary details and

ensured that they are easily accessible. The manuscript has also been updated accordingly (Line 581-591, Page 28).

The model parameters are unique to each subject but follow a consistent methodology for derivation. The parameters for the transduction functions $\frac{a}{s^2+bs+c}$ for active grasping and $\frac{a}{bs^3+cs^2+ds+e}$ for reactive grasping are specified in the supplementary tables. These parameters were extracted through the system identification algorithm, tailored to fit the observed data for each subject. All these parameters are presented in Tables S1-S20, enabling other researchers to replicate our control strategies on their own robotic platforms.

Furthermore, the MATLAB code used for calculating the Victor–Purpura distance and deriving the transduction functions is provided in the supplementary material (Data S1). This includes scripts for both the transfer function and dynamic model, allowing readers to replicate and build upon our work.

We have updated the manuscript to explicitly state the availability of these model parameters and the associated code. This ensures that readers can fully understand and utilize the core models presented in our study.

5. The following statement in conclusion captures the contribution and could go in Abstract (or emphasize the sensation to motor actions where appropriate): ‘this research focuses specifically on tactile sensation and its direct translation into motor actions.’

Response:

Thanks for your suggestion to highlight the specific contribution of our research in the Abstract. In response, we have incorporated the key statement from the conclusion to emphasize the direct translation of tactile sensation into motor actions.

Revised Abstract:

In tactile sensing, decoding the journey from afferent tactile signals to efferent motor commands is a significant challenge primarily due to the difficulty in capturing population-level afferent nerve signals during active touch. This study integrates a finite element hand model with a neural dynamic model using microneurography data to predict neural responses based on contact biomechanics and membrane transduction dynamics. This research focuses specifically on tactile sensation and its direct translation into motor actions. Evaluations of muscle synergy during in vivo

experiments revealed transduction functions linking tactile signals and muscle activation. These functions suggest similar sensorimotor strategies for grasping influenced by object size and weight. The decoded transduction mechanism was validated by restoring human-like sensorimotor performance on a tendon-driven biomimetic hand. This research advances our understanding of translating tactile sensation into motor actions, offering new insights into prosthetic design, robotics, and the development of next-generation prosthetics with neuromorphic tactile feedback.

By incorporating this statement, we highlight the key contribution of our research, ensuring it is prominently featured in the Abstract.

Responses to Reviewer #4

1. The revised paper addressed all comments proposed by the reviewer in previous interactions. I would recommend a professional editing of the figures, so to comply with the journal standards.

Response:

Thank you for the comment. All the figures have been revised by the authors in the updated manuscript to ensure they comply with journal standards and provide a clear presentation of our research.